# Anthropogenic fingerprints in daily precipitation revealed by deep learning

Yoo-Geun Ham[1,11 ✉], Jeong-Hwan Kim[1,11], Seung-Ki Min[2,3 ✉], Daehyun Kim[4], Tim Li[5,6], Axel Timmermann[7,8] & Malte F. Stuecker[9,10]

According to twenty-first century climate-model projections, greenhouse warming will intensify rainfall variability and extremes across the globe[1–4]. However, verifying this prediction using observations has remained a substantial challenge owing to large natural rainfall fluctuations at regional scales[3,4]. Here we show that deep learning successfully detects the emerging climate-change signals in daily precipitation fields during the observed record. We trained a convolutional neural network (CNN)[5] with daily precipitation fields and annual global mean surface air temperature data obtained from an ensemble of present-day and future climate-model simulations[6]. After applying the algorithm to the observational record, we found that the daily precipitation data represented an excellent predictor for the observed planetary warming, as they showed a clear deviation from natural variability since the mid-2010s. Furthermore, we analysed the deep-learning model with an explainable framework and observed that the precipitation variability of the weather timescale (period less than 10 days) over the tropical eastern Pacific and mid-latitude storm-track regions was most sensitive to anthropogenic warming. Our results highlight that, although the long-term shifts in annual mean precipitation remain indiscernible from the natural background variability, the impact of global warming on daily hydrological fluctuations has already emerged.

Changes in precipitation substantially affect societies and ecosystems[1]. Therefore, it is of the utmost importance to determine whether anthropogenic changes in precipitation are detectable. At the planetary scale, global-climate-model simulations show that globally averaged precipitation will increase by approximately 1–3% per degree of warming[7–9]. This change is not spatially homogeneous. Wet regions are projected to have the largest future increase, which is sometimes referred to as the 'wet-gets-wetter'[10] or 'wettest-gets-wetter' response. Moreover, areas that will experience greater ocean warming are also projected to show a mean intensification of rainfall ('warmer-gets-wetter')[11], which may further influence large-scale atmospheric circulation. In accord with the theory, the intensity of extreme daily precipitation events is projected to increase at the rate of about 7% K$^{-1}$ following the Clausius–Clapeyron relation in many parts of the world[2,12], whereas higher rates of increase have been observed regionally[13]. However, owing to the wide range of spatiotemporal scales of precipitation variability, an unequivocal fingerprint of human influence in precipitation has not yet been established from observational records[7,14].

Previous detection and attribution (D&A) studies[12,15,16] have identified anthropogenic influences on preprocessed precipitation statistics, such as the annual maxima of daily precipitation over land areas and the seasonal or zonal averages of global[17,18] and Arctic precipitation[19]. Although using spatial/temporal averages is beneficial for detection because it lowers the uncertainty related to natural internal variability, it is uncertain to what extent detection results based on these smoothed fields can be applied to hydrometeorological weather events that affect our daily lives[2,20].

Determining whether and to what degree greenhouse-gas-induced warming has altered daily precipitation in the observational record remains elusive for two reasons. First, daily precipitation amounts exhibit large internal variabilities associated with non-anthropogenic weather noise, which hinders climate-change-signal detection[8,21–23]. Second, conventional D&A methods assume a fixed spatial pattern of the climate-change signal (that is, fingerprint pattern)[24–26], which may not be sufficient to capture changes in higher-moment statistics such as variance. Therefore, efforts to detect climate-change signals imprinted in daily precipitation have thus far been unsuccessful (Extended Data Fig. 1). In this study, we overcome these two issues by combining large-ensemble climate-model simulations with a deep-learning algorithm and show that deep learning

[1]Department of Oceanography, Chonnam National University, Gwangju, South Korea. [2]Division of Environmental Science and Engineering, Pohang University of Science and Technology, Pohang, South Korea. [3]Institute for Convergence Research and Education in Advanced Technology, Yonsei University, Incheon, South Korea. [4]Department of Atmospheric Sciences, University of Washington, Seattle, WA, USA. [5]Department of Atmospheric Sciences, School of Ocean and Earth Science and Technology, University of Hawai'i at Mānoa, Honolulu, HI, USA. [6]Key Laboratory of Meteorological Disaster, Ministry of Education/Joint International Research Laboratory of Climate and Environmental Change/Collaborative Innovation Center on Forecast and Evaluation of Meteorological Disasters, Nanjing University of Information Science and Technology, Nanjing, China. [7]Center for Climate Physics, Institute for Basic Science, Busan, South Korea. [8]Pusan National University, Busan, South Korea. [9]Department of Oceanography, School of Ocean and Earth Science and Technology, University of Hawai'i at Mānoa, Honolulu, HI, USA. [10]International Pacific Research Center, School of Ocean and Earth Science and Technology, University of Hawai'i at Mānoa, Honolulu, HI, USA. [11]These authors contributed equally: Yoo-Geun Ham, Jeong-Hwan Kim. ✉e-mail: ygham@jnu.ac.kr; skmin@postech.ac.kr

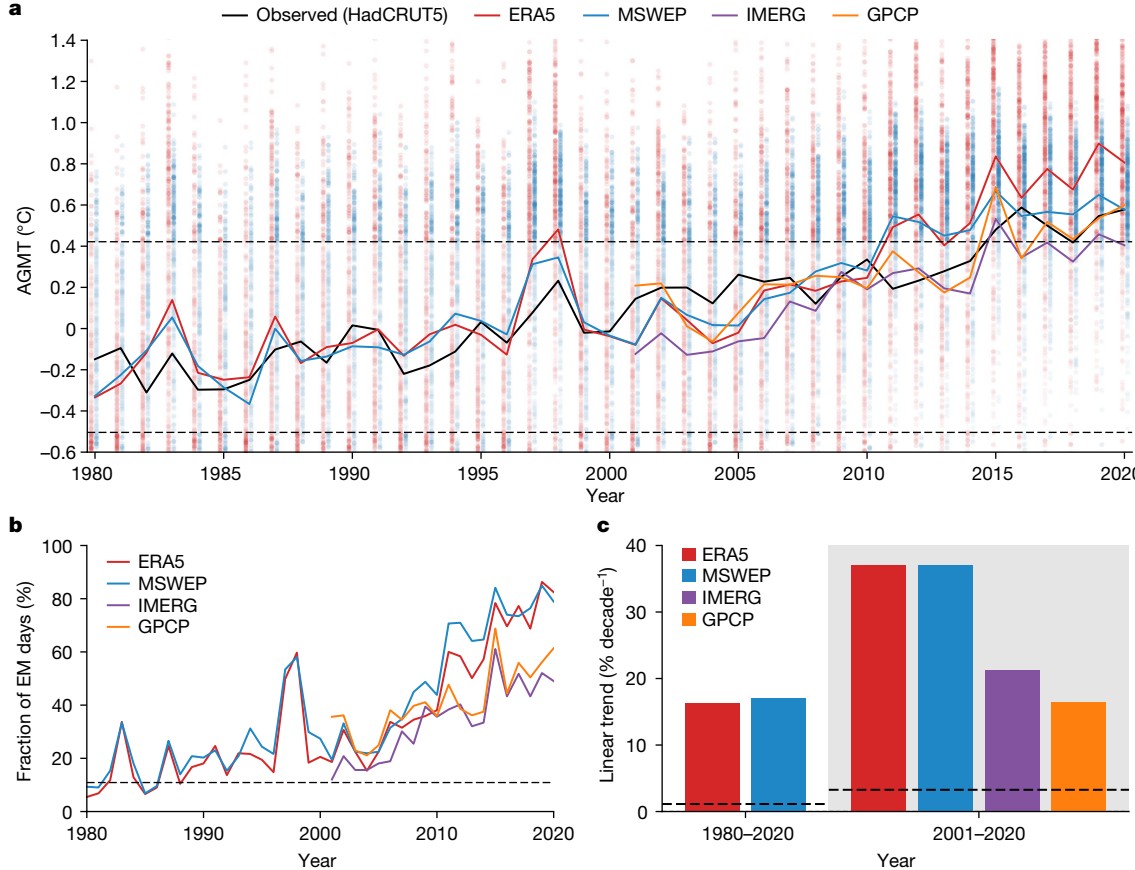

**Fig. 1 | Climate-change detection using deep learning. a**, Time series of the observed AGMT anomaly from 1980 to 2020 (black line) and the annual average of the estimated AGMT obtained by using daily precipitation fields from the MSWEP (blue line), IMERG (purple line) and GPCP (orange line) observations and ERA5 reanalysis (red line) as inputs in the DD model, whose temporal correlation with the observed AGMT is 0.74, 0.80, 0.76 and 0.85 during 2001–2020, respectively. The corresponding coloured dots denote the daily estimated AGMT using the MSWEP and ERA5 precipitation data. The dashed black horizontal lines denote a 95% confidence range of internal variability of the AGMT estimates, defined as the 2.5th–97.5th percentile of the daily estimated AGMT obtained from historical CESM2 LE simulations during 1850–1950. Observed and modelled anomalies are relative to 1980–2010 climatology. **b**, Fractional number of EM days within a corresponding year from 1980 to 2020 for which the estimated AGMT is greater than the upper bound of the 95% confidence range. Dashed line denotes an upper bound of the 95% confidence range of internal variability of fractional EM days, which is 10.9%. **c**, Linear trend of the number of EM days during 1980–2020 in ERA5 and MSWEP and during 2001–2020 in ERA5, MSWEP, IMERG and GPCP. The dashed lines denote upper bounds of the 95% confidence level based on the bootstrap method estimated using the historical CESM2 LE simulations (Methods).

can detect statistically significant climate-change signals in daily precipitation fields.

## Detection results from a deep-learning model

Our deep-learning model for D&A is based on a CNN, which is a widely used deep-learning technique for pattern recognition[5]. The algorithm takes global maps of daily precipitation anomalies (deviations from the long-term daily climatology) as an input variable and outputs an annual global mean 2 m air temperature (AGMT), which is a key climate-change metric[24] (see Methods and Extended Data Fig. 2 for the detailed model structure). To build a deep network that can detect the climate-change signal amidst large internal variability, we trained our deep-learning model with pairs of daily precipitation maps and the AGMT simulated by 80 members of the CESM2 Large Ensemble (LE)[6], which was forced from 1850 to 2100 with estimates of historical forcings and the SSP3-7.0 greenhouse-gas emission scenario (Methods). Being applied as a detection algorithm, the deep-learning model will be referred to as the deep detection (DD) model.

The convolutional process embedded in the CNN is able to capture local features in the global domain[5], making it suitable for detecting regional pattern changes associated with global warming. Also, with its translation-invariant feature, the DD model can extract common change patterns owing to the global warming in both the model simulations and the observations despite their systematic differences[27]. This feature contrasts with the existing D&A techniques, including the revised linear-regression-based approaches[26,28] and the feedforward neural networks[29], which detect climate-change signals based on a global stationary fingerprint pattern.

After being trained using the CESM2 LE data, the DD model was applied to satellite precipitation observations (Integrated Multi-satellite Retrievals for GPM (IMERG)[30] and Global Precipitation Climatology Project (GPCP)[31]), the gauge-satellite-reanalysis merged data (Multi-Source Weighted-Ensemble Precipitation (MSWEP)[32]) and precipitation data from a modern reanalysis product (ERA5) that assimilated ground-based radar and satellite data[33] (Methods). We used several datasets in our detection analysis to account for the uncertainties associated with indirect estimates of precipitation.

With the historical record of observed precipitation data as input, our predicted AGMT from the DD model reproduces the observations very well, with larger increases during recent decades (Fig. 1a), suggesting the possible influence of global warming on recent daily precipitation fields. The Pearson correlation between the annually averaged observed and predicted AGMT from 1980 to 2020 was 0.88 for both

MSWEP and ERA5, with $P$ values of less than 0.001. Slightly lower correlations (0.74–0.85) were found for the latest 20 years (2001–2020). By contrast, the corresponding correlation coefficients obtained using the ridge regression method[24] were systematically lower (0.33 and 0.36 during 1980–2020 for MSWEP and ERA5, respectively) (Extended Data Fig. 3a). This demonstrates that the DD model recognizes the global-warming signal in spatiotemporal features in daily precipitation, which has—so far—not been possible with standard linear detection methods.

To measure the detectability of the observed AGMT variations associated with daily precipitation fields using the DD model, we defined the internal variability range of the daily estimated AGMT as the 2.5th–97.5th percentile values obtained from the CESM2 LE simulation during the historical period from 1850 to 1950 (dashed range in Fig. 1a). The detection results showed that, from the mid-2010s onward, the annual average of the DD-predicted AGMT exceeded the upper bound of internal variability, thereby indicating that greenhouse warming already altered daily precipitation fields.

The days with the estimated AGMT greater than the upper bound of internal variability, henceforth referred to as emergence (EM) days, increased continuously after 1980 (Fig. 1b). The EM days in recent years lie clearly above the 97.5th percentile (10.9%) of internally generated EM days estimated using the CESM2 LE (Methods). From the mid-2010s onward, the climate-change signal can be detected from daily precipitation maps in more than half of all days each year (that is, >50% of the fractional EM days), regardless of the input data type.

The strong positive linear trends of EM days were found for all precipitation datasets: 17.1% decade$^{-1}$ and 16.3% decade$^{-1}$ during 1980–2020 for MSWEP and ERA5, respectively, and 21.3% decade$^{-1}$ and 16.5% decade$^{-1}$ for IMERG and GPCP during 2001–2020, respectively (Fig. 1c). These trends also exceeded the internal variability ranges of the EM days trends (dashed line in Fig. 1c; Methods). The detection result remained largely unaffected by the choice of the climate-model simulations[34] used in the training of the DD models (Extended Data Fig. 4), demonstrating a generalization capability of the deep-learning model for climate-change detection. Unlike our DD-based results, the ridge regression exhibited almost no trend in the EM days during recent decades (Extended Data Fig. 3b,c), indicating that the signal is barely detectable with the linear approach.

## Precipitation timescales and hotspot regions

To identify the source of the climate-change signal, we repeatedly ran the DD model using the satellite and reanalysis precipitation products with each time using anomalies capturing a different timescale. For this task, the precipitation anomalies were decomposed into a linear trend and high-frequency (<10-day), submonthly (10–30-day), subseasonal (30–90-day), subyearly (90-day–1-year) and low-frequency (>1-year) variabilities using Lanczos filtering[35]. The detection results for these different timescales were then compared with those using precipitation anomalies retaining all timescales.

When the linear trend component of precipitation anomalies was given to the DD model, the estimated AGMT decreased in time for both the ERA5 and MSWEP datasets during 1980–2020 (Fig. 2a). During 2001–2020, the results with the IMERG and GPCP datasets disagree on the sign of temporal changes in the estimated AGMT (Fig. 2b), possibly because of the discrepancies in the trend between the precipitation datasets[36]. Clearly, the mean state changes in precipitation represented by the linear trend is not the primary source of the anthropogenic climate-change signal found in Fig. 1. The negative contribution of the linear trend component found in three of the four precipitation datasets may have been partly caused by the recent negative phase of the Interdecadal Pacific Oscillation (IPO) in the tropical Pacific during the early twenty-first century and its associated precipitation response[37,38]. The observed interdecadal trends in tropical rainfall were considerably different from the climate-projection results simulated by global climate models[39] (Extended Data Fig. 5a,b). The weak contribution of the linear trend component to climate-change detection aligns with the outcomes obtained through the ridge regression method using daily precipitation, in which fingerprint pattern is coherent in its signs with the spatial pattern of the climatological precipitation change (Extended Data Fig. 5c).

Among all the timescales considered, the high-frequency precipitation anomalies with periods shorter than 10 days were mostly responsible for the positive trend in the AGMT (Fig. 2a,b), whereas the other temporal scales were found to exert negligible contributions. This clearly demonstrates that the emerging climate-change signal in the observed daily precipitation fields is mostly included in the high-frequency weather components rather than the low-frequency components or changes in the long-term mean states. The dominant role of high-frequency precipitation anomalies in yielding a positive AGMT trend was found regardless of the precipitation input dataset used, whereas the predicted global-warming trends differ slightly between precipitation products. This inter-dataset difference is presumably because of the uncertainties associated with the retrieval algorithms[40] or the forecast model used in the production of the datasets[41], especially over the ocean, for which direct observations of precipitation is lacking[42].

Next, to identify the spatial locations at which the high-frequency precipitation anomalies showed notable changes in association with the climate-change signal, we used a machine-learning-explainable method called occlusion sensitivity[43]. This method quantifies the relative importance of the input fields in deriving the machine-learning prediction. The occlusion sensitivity of an input grid box was obtained as the difference between the DD-model-predicted AGMT obtained with the original input data and the corresponding value obtained after substituting the input data over the 7 × 7 grid boxes surrounding the target grid box with zero (Methods). For each grid box, the occlusion sensitivity was calculated for all days and its linear trend during 1980–2020 was obtained to measure its contribution to the global-warming-signal detection.

The linear trend of occlusion sensitivity (Fig. 2c) highlights several hotspots in which a strong positive trend appears: the northern tropical eastern Pacific, northern South America, north Pacific, north Atlantic and Southern Ocean. Therefore, our results suggest that the positive trend in the estimated AGMT from the DD model was mainly caused by changes in high-frequency precipitation anomalies over these hotspots. These hotspot regions appear distinctly when using a different patch size for occlusion sensitivity (Extended Data Fig. 6a) or using other explainable methods, such as Shapley additive explanations (SHAP)[44] or the integrated gradients[45] method (Extended Data Fig. 6b,c).

The same locations appear as hotspots even when unfiltered anomalies are used (Extended Data Fig. 7a). Also, note that the positive linear trend of the occlusion sensitivity over the hotspots is prominent for satellite precipitation products for a relatively short period (that is, 2001–2020) (Extended Data Fig. 7b,c), whereas those over the equatorial Atlantic and central Africa appear only with the ERA5 and MSWEP datasets.

## Physical interpretations

When the occlusion sensitivity is obtained separately for each high-frequency precipitation percentile over the hotspot regions (boxed areas in Fig. 2c), it is highest for the top and bottom percentiles and lowest at around the 55th–60th percentiles that correspond to values around zero (green lines in Fig. 3a,b). This V-shaped pattern indicates that the DD model generates higher AGMT values for strong high-frequency precipitation anomalies with either a positive or a negative sign over the eastern Pacific Intertropical Convergence Zone

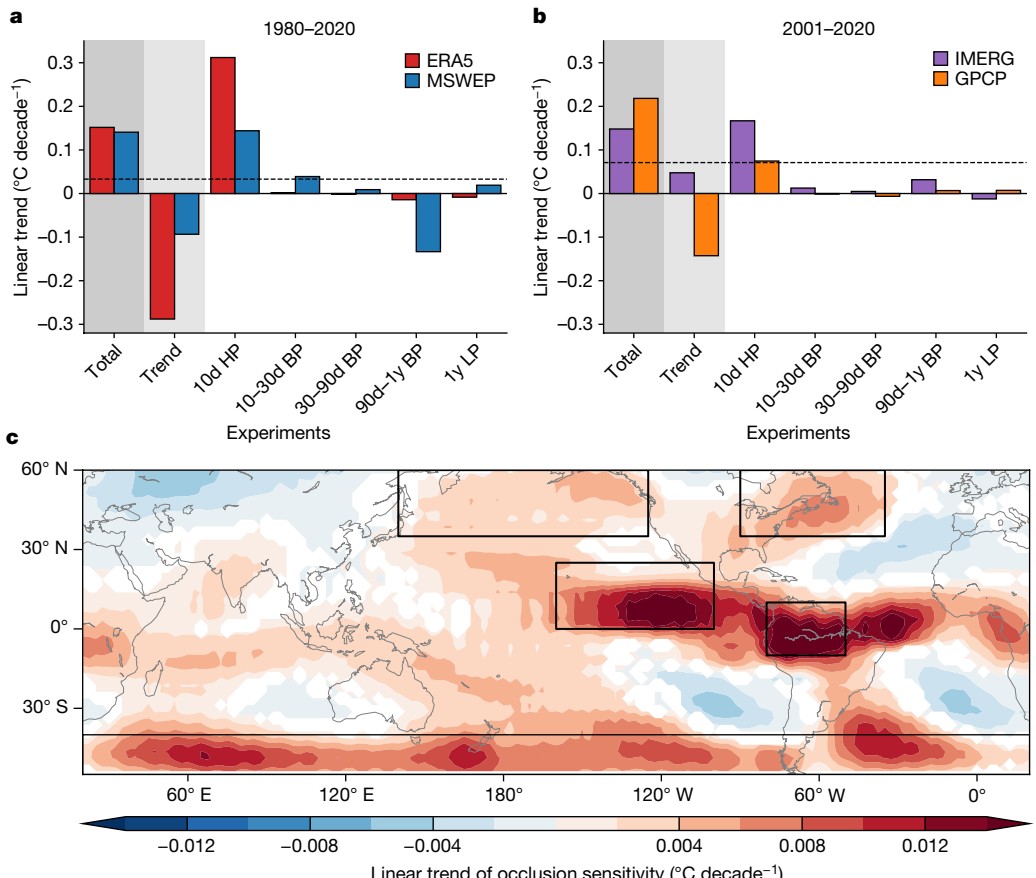

**Fig. 2 | Critical role of high-frequency precipitation variations in climate-change detection. a,b**, Linear trends of the estimated AGMT from the DD model for 1980–2020 using ERA5 reanalysis (red) or MSWEP (blue) (**a**) and 2001–2020 using IMERG (purple) or GPCP (orange) (**b**). Each case shows results from using unfiltered precipitation anomalies (denoted as 'Total'), linear trends of the precipitation anomalies ('Trend'), 10-day high-pass-filtered ('10d HP'), 10–30-day band-pass-filtered ('10–30d BP'), 30–90-day band-pass-filtered ('30–90d BP'), 90-day–1-year band-pass-filtered ('90d–1y BP') and 1-year low-pass-filtered ('1y LP') precipitation. The dashed black horizontal lines in panels **a** and **b** denote the upper bound of a 95% confidence range of internal variability of the estimated AGMT linear trend, obtained from historical CESM2 LE simulations during 1850–1950. **c**, Linear trend of the AGMT occlusion sensitivity for 10-day high-pass-filtered ERA5 and MSWEP precipitation anomalies from 1980 to 2020. Black boxes in panel **c** denote hotspot regions in which a strong positive trend appears. The shaded area indicates that the linear trend value exceeds the 95% confidence level, as determined by a *t*-test. The map was generated using the Basemap Toolkit (version 1.2.0; https://matplotlib.org/basemap/).

(ITCZ), northern South America and mid-latitude storm-track regions. Note that this nonlinear response of the AGMT to high-frequency precipitation anomalies cannot be accounted for in the ridge regression method because of its linear nature (black lines in Fig. 3a,b). Also, the probability density function (PDF) of the high-frequency precipitation anomalies over the hotspots showed a systematic shift towards the extreme percentiles in recent decades; for the top (>90th) and bottom (<10th) percentiles, the ratio of the PDFs for each decade to the reference PDFs for the whole period is smallest in the 1980s and greatest in the 2010s (Fig. 3c,d).

In synthesizing the results presented in Fig. 3, the DD model produces a robust increase in the estimated AGMT over recent decades, with more frequent extreme swings of high-frequency precipitation events over the hotspots. In other words, the DD model underscores the observed amplification of the high-frequency precipitation variability over the eastern Pacific ITCZ and mid-latitude storm-track regions as the global-warming signal. Although past climate-projection results have shown an increase in precipitation variability with warming[3,4,46], researchers have not assessed whether the projected changes can be detected in observations.

The substantial precipitation variability increases over the eastern Pacific ITCZ, northern South America and mid-latitude storm tracks were confirmed by the linear trend of the standard deviation of the observed high-frequency precipitation anomalies during 1980–2010 or 2001–2020 (Fig. 4a). In both satellite observations and the reanalysis products, the increases in the high-frequency precipitation variability in time were statistically significant, beyond the 95% range of internal variability estimated from historical CESM2 LE simulations during 1850–1950.

The recent robust increase in high-frequency variability over the eastern Pacific ITCZ and mid-latitude storm tracks, as represented by the shift of the high-frequency precipitation events from the moderate to the extreme percentiles, is confirmed by means of the spatial distribution of the difference in the high-frequency precipitation variability (Fig. 4b). The increase in the high-frequency precipitation variability during 2016–2020 relative to that during 2001–2005 is prominent over the eastern Pacific ITCZ and mid-latitude storm tracks. More notably, the spatial distribution of the increase in the high-frequency precipitation variability resembles that of the linear trend of the occlusion sensitivity (Fig. 2c). This clearly demonstrates that the global-warming signal was successfully detected from the daily precipitation through the increase in the extreme swings of the precipitation events on weather timescales.

The robust high-frequency precipitation variability increases over the eastern Pacific ITCZ and mid-latitude storm tracks can be physically understood using a simple moisture budget analysis[4] (Methods).

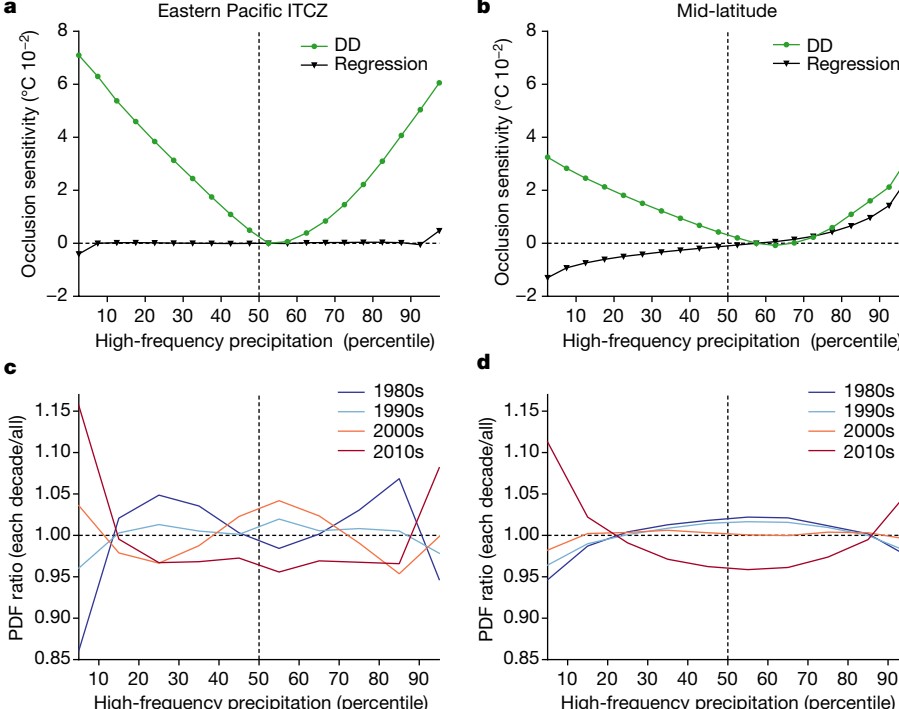

**Fig. 3 | Dominant regional precipitation characteristic changes detected by the deep-learning model. a,b,** The occlusion sensitivity with respect to the percentile of the high-frequency (that is, 10-day high-pass-filtered) precipitation anomalies in the DD model (green), as well as the ridge regression model (black) in the eastern Pacific ITCZ (**a**) and mid-latitude storm tracks (**b**). **c,d,** The ratios of the PDF of the high-frequency precipitation anomalies during the 1980s, 1990s, 2000s or 2010s to that during 1980–2020 in the eastern Pacific ITCZ (**c**) and mid-latitude storm tracks (**d**) (see Methods for the detailed procedure for calculating PDF values). The eastern Pacific ITCZ and mid-latitude storm tracks are defined as the boxed areas within 20° S–20° N and poleward of 30° S and 30° N, respectively (Fig. 2c).

The historical mean precipitation and high-frequency variability are both prominent over the eastern Pacific ITCZ, northern South America and mid-latitude storm tracks (Extended Data Fig. 8), which supports the 'wet-gets-more-variable' and 'variable-gets-more-variable' paradigms, respectively[4]. Even though the observed long-term trend can be obscured by the recent negative IPO event[37], the mean precipitation did increase slightly over the eastern Pacific ITCZ region (Extended Data Fig. 5a), at which the amplitude of the negative IPO-related tropical SST anomalies exhibited a local minimum[47]. Therefore, our conclusion does not invalidate 'warmer-gets-wetter' and its similar paradigm for the high-frequency variability (that is, so-called warmer-gets-more-variable model).

The degree of increase in high-frequency precipitation variability in the eastern Pacific ITCZ and mid-latitude storm tracks is much greater than the corresponding changes in climatological precipitation (Fig. 4c). The high-frequency variability trend ratio (that is, high-frequency precipitation variability trend divided by the variability during a reference period) is approximately three times greater than the climatology trend ratio (that is, mean precipitation trend divided by the climatological value during a reference period) over the eastern Pacific ITCZ and mid-latitude storm-track regions in MSWEP and ERA5. Therefore, our detection method enables one to overcome the limitations associated with linear methods, which have previously underestimated the detectable influence of global warming on precipitation data by focusing on the changes in the mean states and not the higher-order moments.

Our results are further evaluated using direct precipitation measurements. Although the hotspot regions identified in our study are mostly over the ocean, the one located in the Atlantic storm track covers the eastern USA, in which a relatively large number of stations provide daily rain-gauge data[48]. The results from the rain-gauge data are largely consistent with those from the satellite and reanalysis precipitation datasets, indicating a robust increase in the magnitude of high-frequency precipitation variability during recent decades in the eastern USA (Fig. 4d,e and Extended Data Fig. 9a,b). On the contrary, over the western USA, which is outside the Atlantic storm-track hotspot, the high-frequency precipitation variability does not show an organized trend pattern. Also, the change of high-frequency variability is greater than the mean precipitation change only over the eastern USA (Extended Data Fig. 9c,d). This rain-gauge-based analysis increases the robustness of our main findings.

Global warming has resulted in increased high-frequency precipitation variability over the tropical and mid-latitude regions, whereas the subtropical Atlantic and southeastern Pacific show a predominance of climatological drying instead (right bars in Fig. 4c); this is in accordance with the occlusion sensitivity over the corresponding regions, which indicated positive values in the bottom percentiles and negative values in the top percentiles (Extended Data Fig. 10). These results demonstrate that the unique convolutional process with the nonlinear response function in the deep-learning model allows for the detection of the dominant regional characteristic changes among various timescales. Note that the stronger positive AGMT response in the bottom extreme percentiles than the negative response in the top percentiles results in a net negative AGMT response to the decreased high-frequency variability over the subtropical Atlantic and southeastern Pacific (Fig. 4b). Consequently, this contributes to the negative occlusion sensitivity trend with high-frequency precipitation input (Fig. 2c).

Recent advances in deep learning have led to numerous innovative applications in climate science[29,49,50]. Deep learning is a useful method for revealing and categorizing patterns responsible for a target climate phenomenon at various spatiotemporal scales in an automated manner by compressing global information into an abstract level through non-parametric mapping. Through extracting robust regional fingerprints

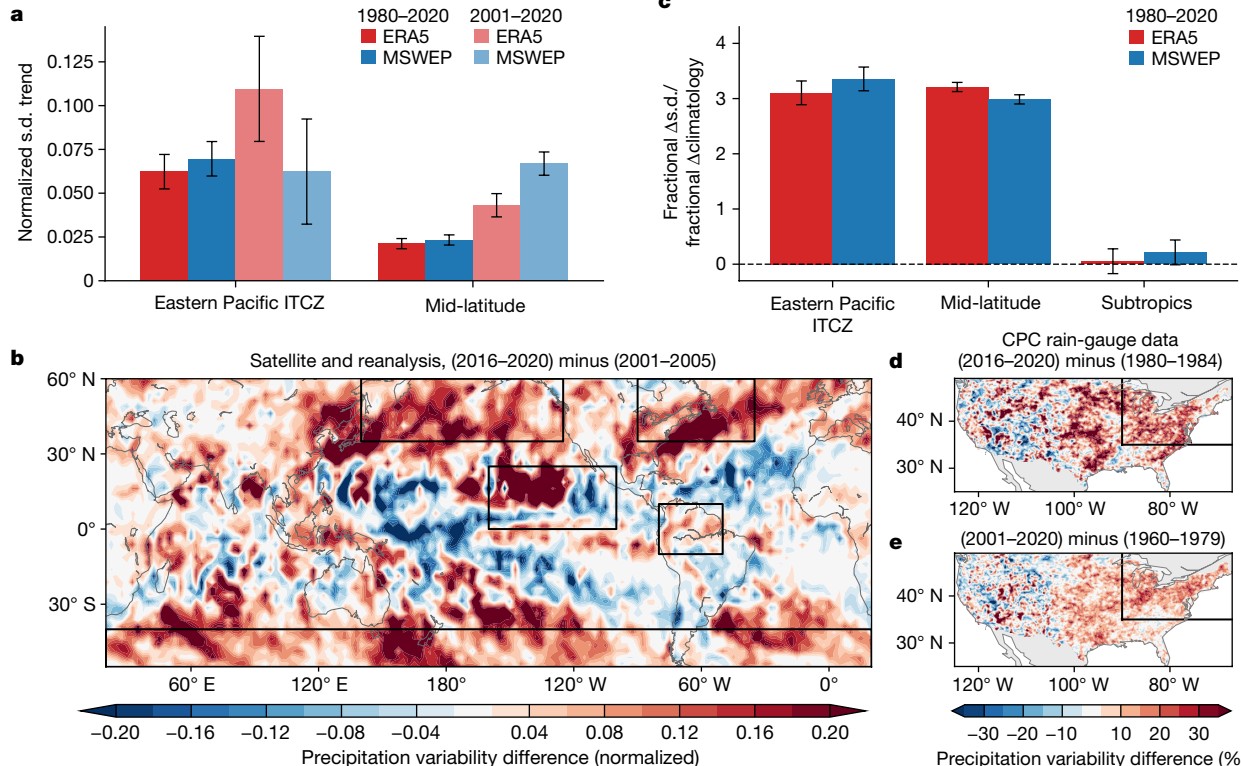

**Fig. 4 | Emergent precipitation variability amplification over the eastern Pacific ITCZ and mid-latitude storm-track regions caused by global warming. a**, Linear trend of the standard deviation (s.d.) of normalized ERA5 (red) and MSWEP (blue) high-frequency precipitation anomalies during 1980–2020 (darker bars) or 2001–2020 (lighter bars) over the eastern Pacific ITCZ (boxed area within 20° S–20° N) and mid-latitude storm-track regions (boxed area poleward of 30° S and 30° N). **b**, The difference in the high-frequency precipitation s.d. between 2016–2020 and 2001–2005 averaged across datasets (MSWEP, ERA5, GPCP and IMERG). **c**, Ratio of the linear trend of high-frequency precipitation variability during 1980–2020 to the linear trend

of precipitation climatology (each divided by the 1980–1984 mean) in the eastern Pacific ITCZ, mid-latitude storm tracks and subtropics using ERA5 (red) and MSWEP (blue). The error bars in panels **a** and **c** denote a 95% confidence interval of internal variability, obtained from historical CESM2 LE simulations during 1850–1950. **d,e**, Difference in the high-frequency precipitation s.d. between 2016–2020 and 1980–1984 (**d**) and between 2001–2020 and 1960–1979 (**e**) using National Oceanic and Atmospheric Administration Climate Prediction Center (CPC) daily rain-gauge data. The map was generated using the Basemap Toolkit (version 1.2.0; https://matplotlib.org/basemap/).

of global warming concealed in the complex probability distribution of precipitation, the deep-learning model has revealed that the observed increase in daily precipitation variability is an emergent anthropogenic signal despite a short period of precipitation datasets; however, the mean state changes remain virtually undetectable, as they are hindered by the large internal day-to-day variability. This confirms that the impact of global warming is ubiquitous and detectable, even in variables associated with high natural variance.

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

## Methods

### DD model for climate-change detection

The DD model, which refers to the CNN model for detecting climate-change signals embedded in daily precipitation anomalies, comprises an input layer, five convolution layers, two pooling layers, one fully connected layer and an output layer (Extended Data Fig. 2). The size of the convolution kernel, which extracts key features from the input to produce feature maps, is $3 \times 3$. Spatial pooling was performed after the first two convolution processes by using $2 \times 2$ max pooling with a stride of 2. L2 regularization was applied to minimize overfitting[24,51].

The DD model accepts gridded data of normalized daily precipitation anomalies as an input variable. These anomalies were determined by subtracting the daily climatology data for 1980–2010 and then normalizing them by dividing the longitudinally averaged standard deviation of the daily precipitation anomalies at the corresponding latitude during the same time period. The input variable has dimensions of $160 \times 55$ ($2.5° \times 2.5°$ resolution over 0°–400° E, 62.5° S–76.5° N). To properly consider the precipitation pattern around 360°(0°) E, the data were longitudinally extended by concatenating 0°–360° E and 360°–400° E. Through five convolutional and two max-pooling processes, the horizontal dimension of the feature map is reduced to $40 \times 14$. As the last convolutional layer uses 16 convolutional filters, the size of the dimension of the final feature map is 8,960 (that is, $40 \times 14 \times 16$). Then, each element of the final feature map is connected to the first dense layer with 32 neurons and, finally, the first dense layer is connected to the second dense layer with a single neuron to output a scalar value representing the AGMT anomaly of the corresponding year. The variability of the estimated annual mean AGMT anomaly was matched to the observed data to avoid the influence of systematic differences between the training and testing samples. Note that this post-processing did not affect the detection results, as both the test statistics (that is, internal variability of the estimated AGMT) and the detection metric (that is, AGMT on any specific day) were modified to the same degree.

We generated five ensemble members with different random initial weights and defined the ensemble-averaged AGMT as the final forecast. The Xavier initialization technique was applied to initialize weights and biases[52]. Tangent hyperbolic and sigmoid functions were used as the activation functions for the convolutional and fully connected layers, respectively. Adam optimization was applied as the gradient-descent method and the mean absolute error was applied as the loss function[53].

### Natural variability estimation

The natural variability ranges of the estimated AGMT and EM days are measured using a bootstrap method. First, the AGMT and the fractional EM days (that is, the number of EM days/365) are calculated for each year using the daily precipitation output of CESM2 historical ensemble simulations for 1850–1950. Then, the 97.5th percentile values of the 8,080 total cases (that is, 101 year × 80 ensemble members) are estimated, which corresponds to the upper bound of the 95% confidence range of the natural variability. The resulting 97.5th percentile values are 0.42 °C and 10.9% for AGMT and fractional EM days, respectively.

The natural variability range of the linear trends in the EM days (Fig. 1c), the estimated AGMT (Fig. 2a,b) and the precipitation variability (Fig. 4a,c) are also defined using a bootstrap method. We first sample 20-year segments from CESM2 LE simulations during 1850–1950 with a 10-year interval in the initial year of the segments. With nine values per ensemble member, a total of 720 (9 × 80) values of the 20-year segments are obtained. Similarly, a total of 960 samples of 41-year segments are constructed with a 5-year interval. Then, linear trends of the EM days, the estimated AGMT and the precipitation variability are calculated for each 20-year or 41-year segment. Finally, the upper and lower 2.5% percentile values are defined as the 95% two-tailed confidence interval of the natural variability.

### Occlusion sensitivity

Occlusion sensitivity is used to quantify the relative importance of each grid point when deriving an output variable[43]. The occlusion sensitivity $O(t, x, y)$ is a three-dimensional tensor incorporating time ($t$), longitude ($x$) and latitude ($y$) and is calculated using the following equation:

$$O(t, x, y) = \hat{y} - D[P(t, x, y) * Z(7, 7)].$$

Here $*$ is the horizontal convolution operator, $D$ and $\hat{y}$ denote the DD model and the estimated AGMT with the original input data $P(t, x, y)$, respectively, and $Z(7, 7)$ denotes $7 \times 7$ grid points occluding a mask with zero filling. A different patch size of $5 \times 5$ grid points is used for a sensitivity test. The occlusion sensitivity is plotted at the centred grid point of the corresponding grid box. To maintain the original size of the input map, the edge of the map is filled with zeros (that is, zero padding).

### Ridge regression method

The ridge regression method is used to estimate the coefficients of multiple regression models in which linearly independent variables are highly correlated. The loss function of ridge regression with $i$ samples and $j$ regression coefficients is defined by the following equation[24]:

$$\text{Loss} = \sum_{i=1}^{M} \left( y_i - \sum_{j=0}^{P} w_j x_{i,j} \right)^2 + \lambda \sum_{j=0}^{P} w_j^2,$$

in which $x_{i,j}$ and $y_i$ denote the input and label data, respectively. $w_j$ indicates the regression coefficient and $\lambda \sum_{j=0}^{P} w_j^2$ is the regularization term based on the sum of the squared regression coefficients (that is, the L2 norm). $P$ and $M$ denote the number of samples and the number of regression coefficients, respectively. The regularization suppresses overfitting by preventing the regression coefficient from becoming excessively large. $\lambda$ is a hyperparameter that determines the penalty intensity, which is set to 0.1 after several experiments to minimize the loss values for the validation dataset.

### Moisture budget equation for precipitation variability change

For timescales longer than a day, the zeroth-order balance in the moisture budget is found between precipitation ($P$) and vertical moisture advection[4]:

$$P_f \approx -\langle \omega \partial_p q \rangle_f, \tag{1}$$

in which $\omega$ and $q$ are the vertical pressure velocity and specific humidity, respectively. $\langle \cdot \rangle = \frac{1}{g} \int_{P_s}^{P_t} \cdot \, dp$ denotes the vertical integral throughout the troposphere. The subscript $f$ denotes variations at a specific timescale derived from the time filtering. Zhang et al.[4] suggested that the column-integrated vertical moisture advection can be reasonably approximated as the advection of the low-tropospheric mean moisture ($\overline{q}_l$) by mid-tropospheric vertical velocity anomalies ($\omega_{m_f}$), hence:

$$P_f \approx -\frac{\omega_{m_f} \overline{q}_l}{g}, \tag{2}$$

in which $g$ is the gravitational acceleration.

Equation (2) can be modified to denote the variability of precipitation and its change owing to the global warming as follows:

$$\sigma[P_f] \approx \frac{\sigma[(\omega_m)_f] \overline{q}_l}{g}; \tag{3}$$

$$\Delta\sigma[P_f] = \frac{1}{g}[\Delta\sigma[(\omega_m)_f]\overline{q}_{l0} + \sigma[(\omega_{m0})_f]\Delta\overline{q}_l], \tag{4}$$

in which $\sigma$ denotes standard deviation and $\Delta$ denotes the difference between the historical and future warming periods. The subscript 0 indicates the values from the historical period.

According to equation (4), well-known models for global precipitation change are similarly applicable for the high-frequency precipitation variability changes. The historical moisture climatology term (that is, $\bar{q}_{l0}$) on the right-hand side refers to the 'wet-gets-more-variable' paradigm and the historical precipitation variability term (that is, $\sigma[(\omega_{m0})_f]$) refers to the 'variable-gets-more-variable' paradigm. Given the strong coupling between the low-level moisture and the sea-surface temperature, the climatological moisture change term (that is, $\Delta\bar{q}_l$) presumably implies the 'warmer-gets-more-variable' paradigm.

### Satellite and reanalysis dataset

We analysed 21 years (2001–2020) of daily mean satellite-observed precipitation data from the IMERG version 6 (ref. 30) and the GPCP version 3.2 (ref. 31). Daily gauge-satellite-reanalysis merged precipitation was obtained from the MSWEP version 2.8 for the period from 1980 to 2020 (ref. 32). Daily reanalysis precipitation data obtained from ERA5, which spans 1980–2020, were also used[33]. Data were interpolated to a 2.5° × 2.5° horizontal grid. Domains over 0°–360° E and 61.25° S–76.25° N were used. Daily gauge-based precipitation at the horizontal resolution is 0.25° × 0.25° from National Oceanic and Atmospheric Administration (NOAA) CPC from 1960 to 2020 (ref. 48) was used for a domain over the continental USA (126.25°–67.25° W, 20°–49.5° N). The AGMT was obtained from HadCRUT5 data[54].

The PDF of the daily precipitation anomalies with respect to its percentile is calculated for each decade. After arranging daily precipitation anomalies over certain regions during the whole period (that is, 1980–2020) by their magnitudes, the values of precipitation for every tenth percentile are defined. The PDF of precipitation anomalies for each decade were calculated in the same way and then compared with the reference PDF value estimated by using the whole period for each percentile (Fig. 3).

### CESM2 LE simulations

To train the DD model, we used a climate model dataset from the CESM2 LE, which has state-of-the-art skills in simulating characteristics of the daily precipitation at various timescales[6]. All the ensemble members that provide daily precipitation output were used (that is, 80 ensemble members). With the aid of tens of realizations for historical and global-warming-scenario simulations, the total number of samples used in training our DD model is larger than what any other model simulation framework can provide, which is advantageous for training the deep-learning model. The simulations cover the period from 1850 to 2100, of which data from 1850 to 2014 were obtained from the historical simulations and the rest from the SSP3-7.0 scenario simulations. A domain over 0°–360° E and 61.25° S–76.25° N was used and the horizontal resolution was coarsened to 2.5° × 2.5°. The input data were prescribed in the form of a normalized anomaly; the modelled daily climatology from 1980 to 2010 was subtracted from the raw precipitation fields and then divided by longitudinally averaged standard deviation at the corresponding latitude during the same period.

Because the total number of samples was 7,329,200 days (80 members × 251 years × 365 days), which exceeded the limit of our computing resources, we subsampled the training and validation datasets by randomly selecting one year from each decade. Thus, the total number of days of training data was reduced to 730,000. For the validation dataset, we randomly selected a different year from each decade and then randomly selected 73 days from each selected year. The total number of days of validation data used was 146,000.

### Data availability

The data related to this study can be downloaded from: IMERG version 6, https://gpm.nasa.gov/data/imerg; ERA5, https://www.ecmwf.int/en/forecasts/datasets/reanalysis-datasets/era5; MSWEP version 2.8, http://www.gloh2o.org/mswep/; GPCP version 3.2, https://disc.gsfc.nasa.gov/datasets/GPCPDAY_3.2/summary; CESM2 LE, https://www.cesm.ucar.edu/projects/community-projects/LENS2/; CMIP6, https://esgf-node.llnl.gov/projects/cmip6/; HadCRUT5, https://www.metoffice.gov.uk/hadobs/hadcrut5/; CPC rain-gauge data, https://psl.noaa.gov/data/gridded/data.unified.daily.conus.html.

### Code availability

TensorFlow (https://www.tensorflow.org) libraries were used to formulate a climate-change-detection model using a CNN. The codes for generating the detection model and plotting the figures were downloaded from https://doi.org/10.5281/zenodo.8107114.

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

**Acknowledgements** This study was supported by Korea Environmental Industry & Technology Institute (KEITI) through 'Project for developing an observation-based GHG emissions geospatial information map', funded by Korea Ministry of Environment (MOE) (RS-2023-00232066). S.-K.M. was supported by the National Research Foundation of Korea (NRF) under grant no. NRF2021R1A2C3007366. T.L. was supported by NSFC grant 42088101 and NOAA grant NA18OAR4310298. A.T. was supported by the Institute for Basic Science (IBS), Republic of Korea, under grant IBS-R028-D1. M.F.S. was supported by NSF grant AGS-2141728 and the NOAA's Climate Program Office's Modeling, Analysis, Predictions, and Projections (MAPP) programme grant NA20OAR4310445. D.K. was supported by the Royal Research Foundation at the University of Washington, the NASA MAP programme (80NSSC21K1495), NOAA MAPP programme (NA21OAR4310343), NOAA CVP programme (NA22OAR4310608). This is SOEST publication 11687 and IPRC contribution 1602. We would also like to acknowledge the CESM2 Large Ensemble Project and supercomputing resources provided by the IBS Center for Climate Physics in South Korea and especially thank S.-S. Lee, N. Rosenbloom and J. Edwards for their important contributions to the CESM2 Large Ensemble Project.

**Author contributions** Y.-G.H. and J.-H.K. designed the study. Y.-G.H., S.-K.M., D.K. and J.-H.K. wrote the manuscript. J.-H.K. and Y.-G.H. performed the experiments and analyses. A.T. and M.F.S. contributed to the CESM2 Large Ensemble Project. All the authors discussed the study results and made substantial improvements to the manuscript.

**Competing interests** The authors declare no competing interests.

**Additional information**
**Correspondence and requests for materials** should be addressed to Yoo-Geun Ham or Seung-Ki Min.

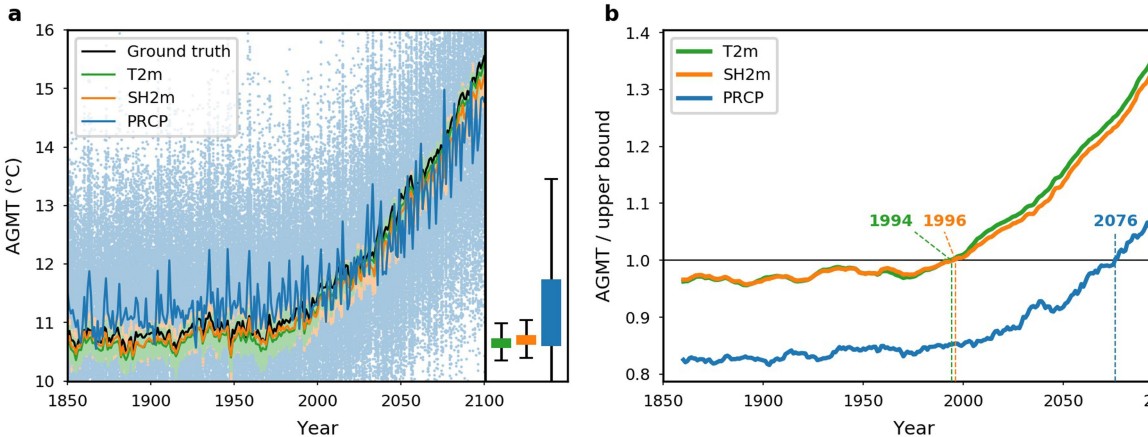

**Extended Data Fig. 1 | Climate-change detection using the ridge regression method in the CESM2 LE. a**, Time series of the simulated AGMT from 1850 to 2100 in the CESM2 LE (black line) and the annual average of the estimated daily AGMT by prescribing 2 m temperature (T2m, green line), 2 m specific humidity (SH2m, orange line) and precipitation (PRCP, blue line) in the ridge regression model[24]. Each dot denotes the estimated AGMT using daily input. The green, orange and blue bars on the right denote one standard deviation of estimated daily AGMT using T2m, SH2m and PRCP during the historical period (that is, 1850–1950), respectively. The black error bars denote the 2.5th–97.5th percentiles of the daily estimated AGMT in 1850–1950. **b**, Time series of the ratio of the annually averaged AGMT to the AGMT of the upper limit of test statistics (that is, 97.5th percentile of the daily estimated AGMT in 1850–1950). The first year that the ratio exceeds 1 for each case is indicated.

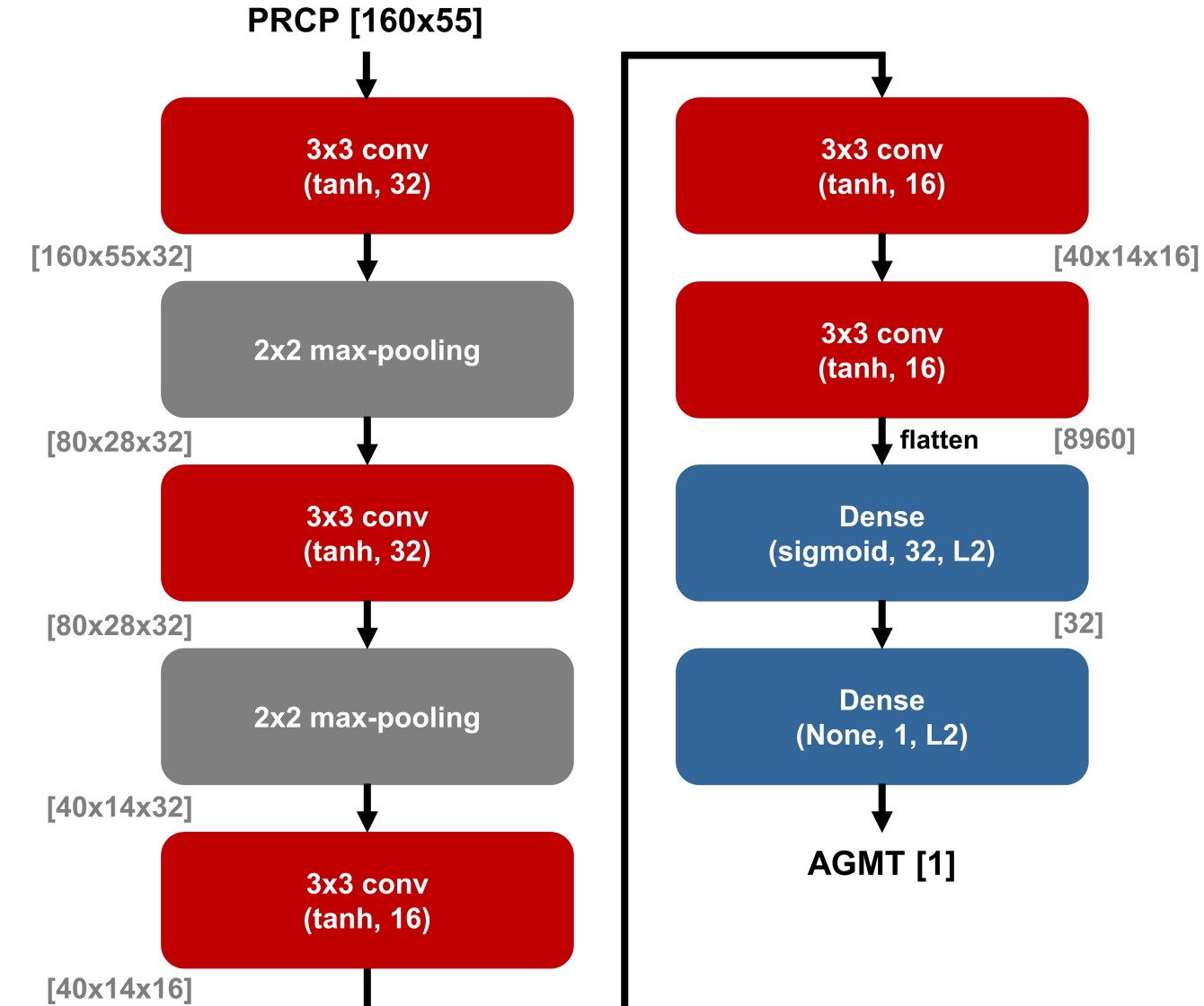

**Extended Data Fig. 2 | The architecture of the deep-learning model for climate-change detection.** The red, grey and blue boxes denote the convolutional layer, max-pooling layer and dense layer, respectively. The dimension of the product at each layer is denoted in the square brackets. The parentheses in the red boxes denote an activation function and the number of convolutional filters in the convolutional layer and the numbers in the parentheses in the blue boxes denote an activation function and the number of neurons in the dense layer.

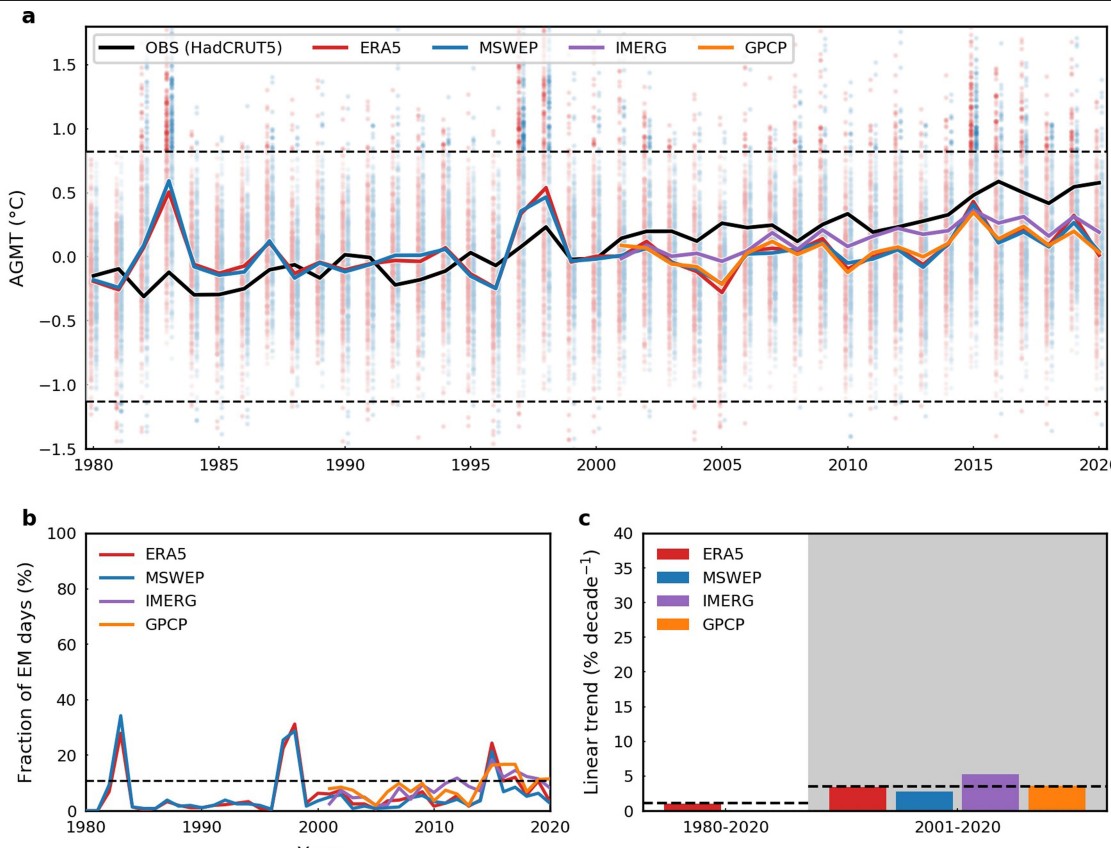

**Extended Data Fig. 3 | Climate-change detection using the ridge regression method. a**, Time series of the observed AGMT anomaly from 1980 to 2020 (black line) and the annual average of the estimated AGMT using daily precipitation fields from the MSWEP (blue line), IMERG (purple line), GPCP (orange line) observations and ERA5 reanalysis (red line) as inputs in the ridge regression model. The blue and red dots denote the daily estimated AGMT using the MSWEP and ERA5 precipitation data, respectively. The dashed black horizontal lines denote a 95% confidence range of internal variability of the AGMT estimates, defined as the 2.5th–97.5th percentile of the daily estimated AGMT obtained from historical CESM2 LE simulations during 1850–1950. **b**, Fractional number of EM days within a corresponding year from 1980 to 2020 for which the estimated AGMT is greater than the upper bound of the 95% confidence range. Dashed line denotes an upper bound of the 95% confidence range of internal variability of fractional EM days, which is 10.9%. **c**, Linear trend of the number of EM days during 1980–2020 in ERA5 and MSWEP and 2001–2020 in ERA5, MSWEP, IMERG and GPCP. The dashed lines denote the upper bounds of the 95% confidence level based on the bootstrap method estimated using the historical CESM2 LE simulations (see Methods).

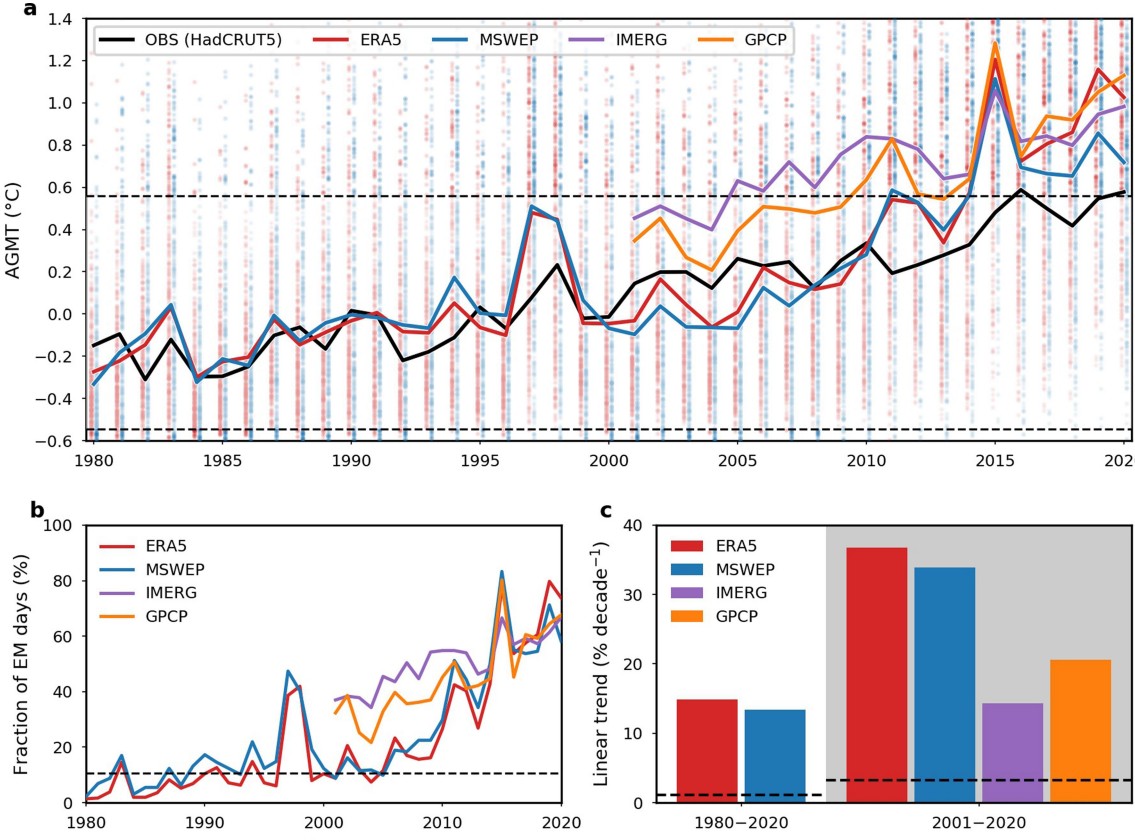

**Extended Data Fig. 4 | Deep-learning-based detection results using CMIP6 training dataset. a**, Time series of the observed AGMT anomaly from 1980 to 2020 (black line) and the annual average of the estimated AGMT obtained by using daily precipitation fields from the MSWEP (blue line), IMERG (purple line) and GPCP (orange line) observations and ERA5 reanalysis (red line) as inputs in the DD model trained with the historical + SSP3-7.0 simulations participated in CMIP6 models. The corresponding coloured dots denote the daily estimated AGMT using the MSWEP and ERA5 precipitation data. The dashed black horizontal lines denote a 95% confidence range of internal variability of the AGMT estimates, defined as the 2.5th–97.5th percentile of the daily estimated AGMT obtained from historical CMIP6 simulations during 1850–1950.
**b**, Fractional number of EM days within a corresponding year from 1980 to 2020 for which the estimated AGMT is greater than the upper bound of the 95%

confidence range. Dashed line denotes an upper bound of the 95% confidence range of internal variability of fractional EM days, which is 10.9%. **c**, Linear trend of the number of EM days during 1980–2020 in ERA5 and MSWEP and 2001–2020 in ERA5, MSWEP, IMERG and GPCP. The dashed lines denote the upper bounds of the 95% confidence level based on the bootstrap method estimated using the historical CMIP6 simulations. The first ensemble of 20 CMIP6 models, ACCESS-CM2, ACCESS-ESM1-5, CanESM5, CESM2, CESM2-WACCM, CMCC-CM2-SR5, EC-Earth3-AerChem, EC-Earth3, EC-Earth3-Veg, FGOALS-g3, GFDL-ESM4, INM-CM4-8, INM-CM5-0, IPSL-CM6A-LR, MIROC6, MPI-ESM1-2-HR, MPIP-ESM1-2-LR, MRI-ESM2-0, NorESM2-LM and NorESM2-MM, are used. The CMIP6 dataset for the DD model training includes historical simulations from 1850 to 2014 and future projections from 2015 to 2100 under the SSP3-7.0 scenario.

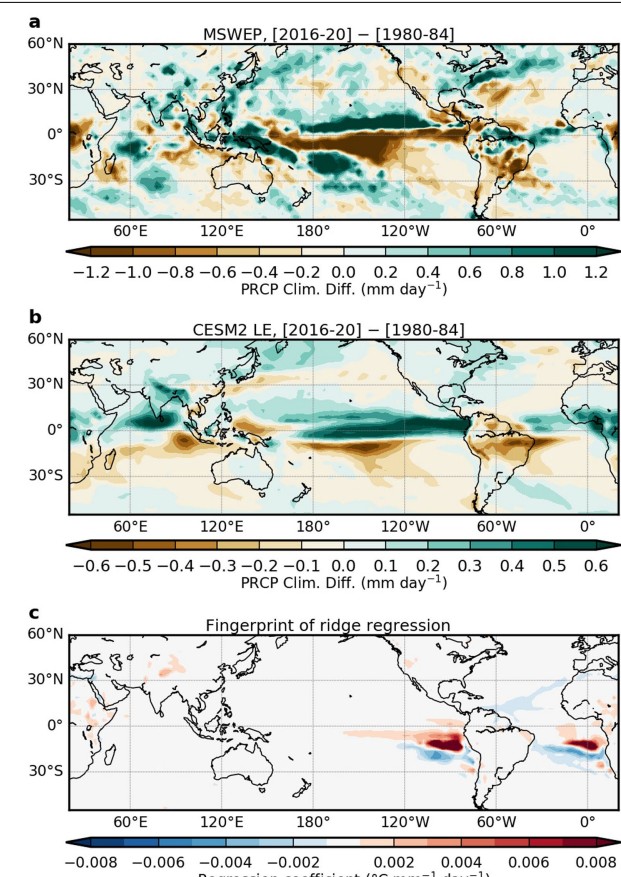

**Extended Data Fig. 5 | Changes in the climatological precipitation and the fingerprint pattern of the ridge regression method. a,b,** The observed difference of the climatological precipitation during 2016–2020 from that during 1980–1984 in MSWEP (**a**) and CESM2 LE (**b**) (unit: mm day⁻¹). **c,** The fingerprint pattern of the ridge regression model obtained by calculating the regression coefficients of the daily precipitation anomalies with respect to the AGMT during 1850–2100 in CESM2 LE (unit: °C mm⁻¹ day⁻¹). The map was generated using the Basemap Toolkit (version 1.2.0; https://matplotlib.org/basemap/).

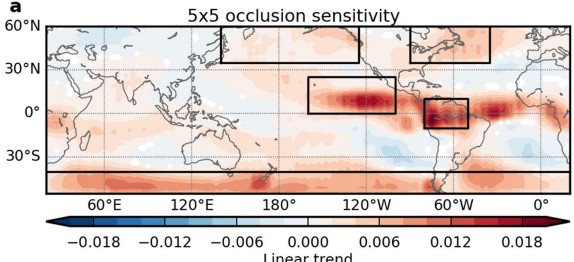

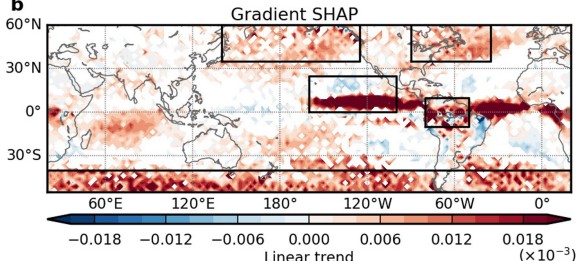

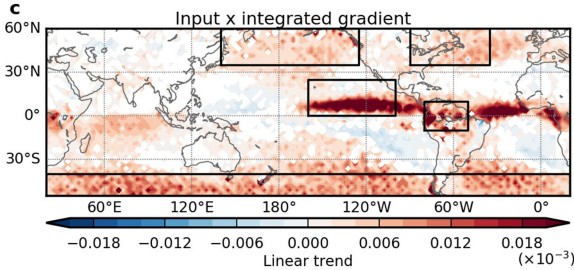

**Extended Data Fig. 6 | Hotspot regions revealed by other explainable methods. a–c**, Linear trend of the AGMT sensitivity (unit: °C decade⁻¹) for 10-day high-pass-filtered ERA5 and MSWEP precipitation anomalies during 1980–2020 measured by occlusion sensitivity with a patch size of 5 × 5 grid points (**a**) and SHAP (**b**) and Input × Integrated Gradients (**c**) methods. The SHAP value is estimated using a gradient explainer, in which the explainer is approximated by 1,000 randomly selected samples from the training dataset. The Input × Integrated Gradients is estimated by multiplying the original input by each of the alpha values (that is, 0, 0.2, 0.4, 0.6, 0.8 and 1.0), computing the gradient of the CNN model, integrating the obtained gradients (the integral is approximated by a Riemann sum) and multiplying by the original input. The shaded area indicates that the linear trend value exceeds the 95% confidence level, as determined by a *t*-test. The map was generated using the Basemap Toolkit (version 1.2.0; https://matplotlib.org/basemap/).

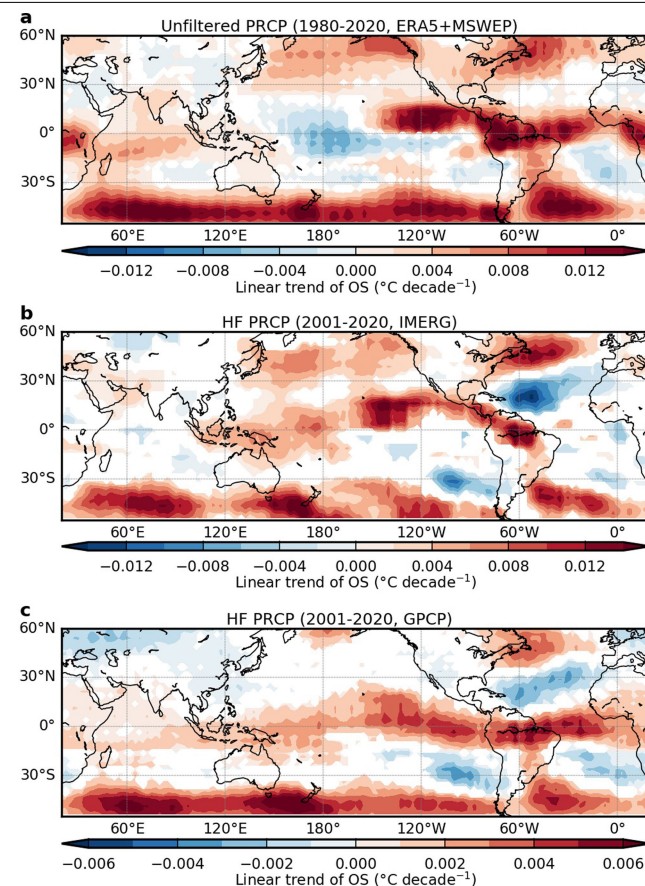

**Extended Data Fig. 7 | Occlusion sensitivity using the observed precipitation dataset. a–c**, Linear trend of the AGMT occlusion sensitivity (unit: °C decade$^{-1}$) using the unfiltered ERA5 and MSWEP precipitation anomalies from 1980 to 2020 (**a**) and that using 10-day high-pass-filtered precipitation from IMERG (**b**) and GPCP (**c**) satellite observations during 2001–2020. The shaded area indicates that the linear trend value exceeds the 95% confidence level, as determined by a *t*-test. The map was generated using the Basemap Toolkit (version 1.2.0; https://matplotlib.org/basemap/).

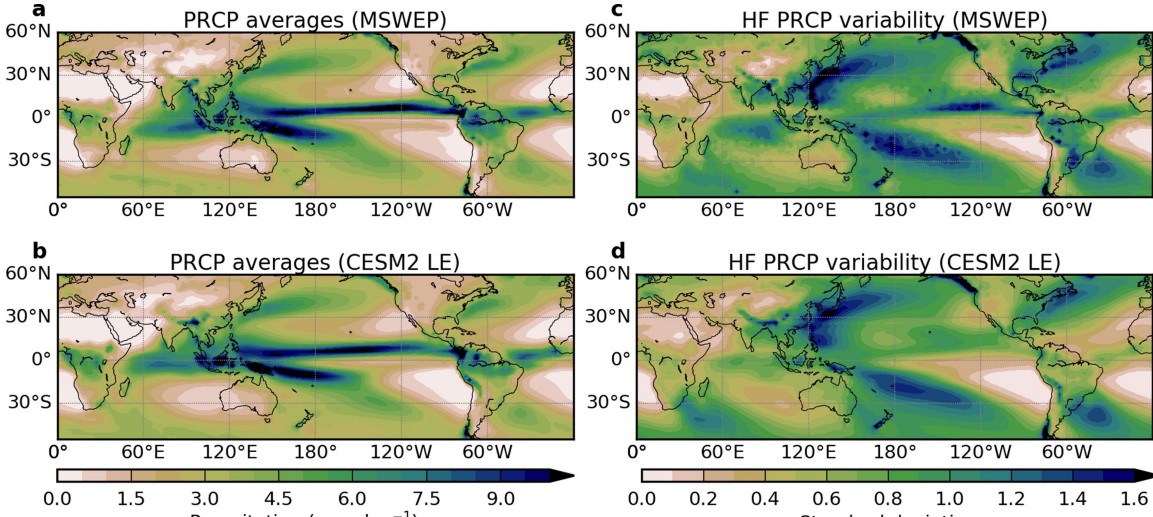

**Extended Data Fig. 8 | Spatial distribution of the historical precipitation and high-frequency precipitation variability climatology. a,b**, Climatological precipitation during 1980–2020 in MSWEP (**a**) and CESM2 LE (**b**). **c,d**, The climatological high-frequency (10-day high-pass-filtered) precipitation variability during 1980–2020 in MSWEP (**c**) observation and CESM2 LE (**d**). The map was generated using the Basemap Toolkit (version 1.2.0; https://matplotlib.org/basemap/).

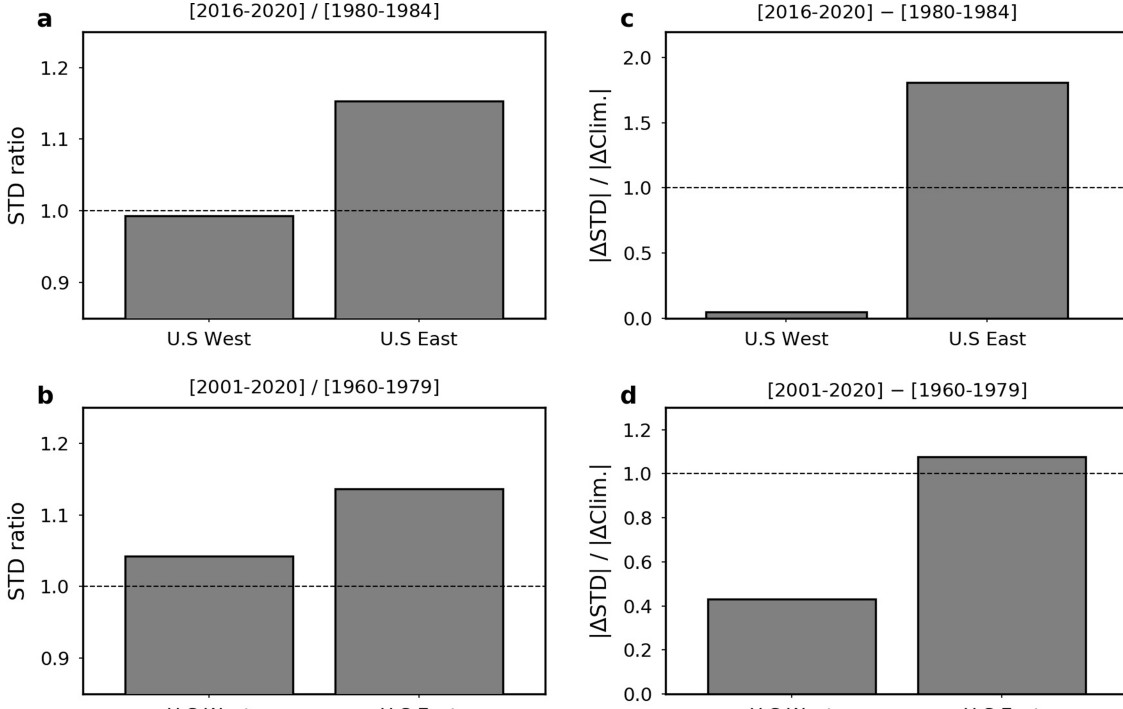

**Extended Data Fig. 9 | Evaluations using the rain-gauge data. a**, Ratio of the standard deviation (STD) of the high-frequency (that is, 10-day high-pass-filtered) precipitation anomalies during 2016–2020 to that during 1980–1984 in western USA (125°–110° W, 30°–50° N) and eastern USA (90°–55° W, 35°–50° N). **b**, Same as **a** but for the ratio of variability during 2001–2020 to that during 1960–1979. **c**, Percentage change in precipitation variability during 2016–2020 compared with 1980–1984 (that is, STD change divided by STD during 1980–1984) divided by the percent change in precipitation climatology in the western and eastern USA. **d**, Same as **c** but for the ratio between 2001–2020 to 1960–1979.

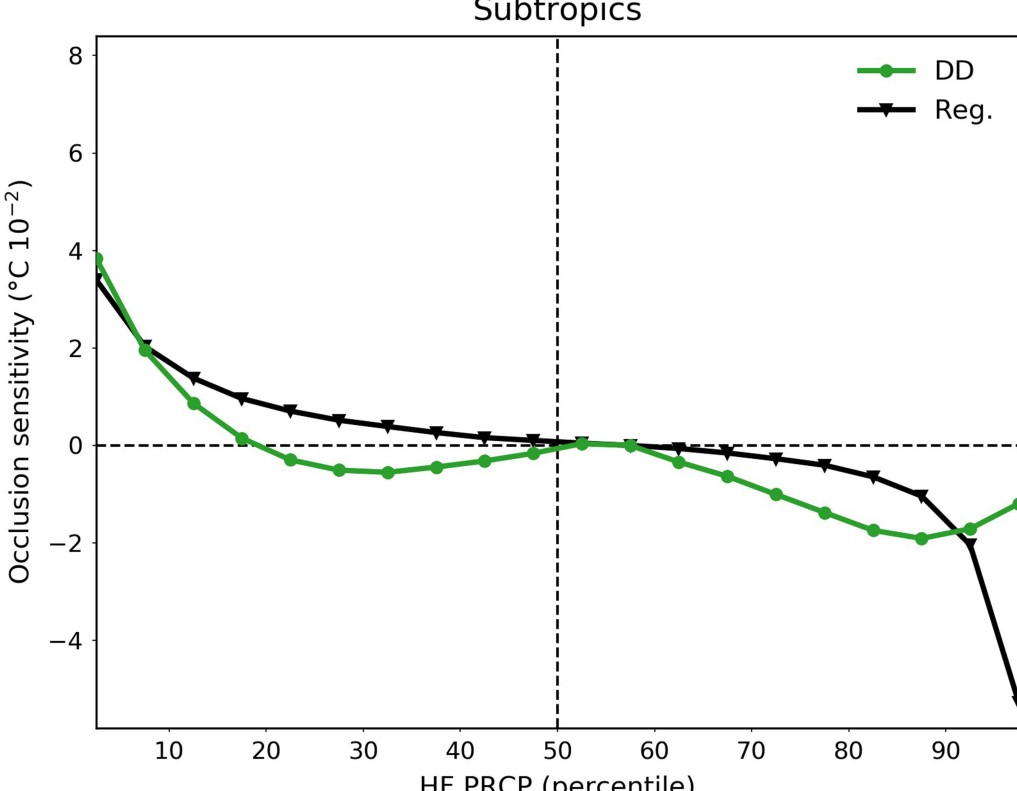

**Extended Data Fig. 10 | Subtropical drying captured in the DD model.** The occlusion sensitivity (°C 10⁻²) with respect to the percentile of the high-frequency (that is, 10-day high-pass-filtered) precipitation anomalies in the DD model (green) and the ridge regression model (black) in the subtropical Atlantic (40°–0° W, 25°–40° N) and southeastern Pacific (115°–75° W, 35°–20° S).