## [Peer Review File · Nature]

Manuscript Title: Anthropogenic fingerprints in daily precipitation revealed by deep learning

Redactions – unpublished data

Reviewer Comments & Author Rebuttals

Reviewer Reports on the Initial Version:

Referees' comments:

Referee #1 (Remarks to the Author):

Review "Anthropogenic fingerprints in daily precipitation revealed by deep learning" by Ham et al

Summary:

The authors train a deep learning model to predict annual global mean temperatures from daily precipitation maps, based on a large climate model ensemble. The authors then apply the deep learning model to daily precipitation maps (from a reanalysis and a satellite dataset), and show that both show an increasing trend, and that in recent years a sizeable fraction (about half of the days) has emerged from the natural climate variability, thus indicating detection of forced climate change in daily precipitation at almost global scale. The authors further probe the results and find that the detection stems largely from a number of hot spot regions and from the high-frequency time scale, thus indicating that changes in precipitation variability (that have long been predicted from theory) would be detectable at the global scale.

The results are, in my opinion, very interesting, and would add to and extend previous detection/attribution results of precipitation extremes to individual days (at the global scale). However, I also have a few concerns regarding (1) the observational datasets (which only provide indirect estimates of precipitation) and whether these datasets are "fit-for-purpose" for a detection/attribution study of daily precipitation, and the noise model of natural variability used for detection is problematic (only a 30-year period from ERA5), (2) the training dataset (consisting of only one climate model) and its generalisability to an out-of-distribution sample (the "observational" estimates), and (3) a more in-depth discussion, interpretation, and understanding of the results of the machine learning model applied to test datasets, and why in particular these hotspot regions stand out. In addition, I have a few comments/questions related to the methodology, outlined below.

Major comments:

(1) Observational datasets and detection from natural variability for very short reference period

The authors claim that "deep learning successfully detects the emerging climate change signals in daily precipitation fields during the observed record. [...] after applying the algorithm to the observational record, we found that the daily precipitation data represented an excellent predictor for the observed planetary warming [...]." (Abstract) However, the authors use a reanalysis dataset (ERA5) and a satellite-based precipitation record (IMERG) as "observational" input for the detection algorithm.

This is a bit concerning, because ERA5 does **not** assimilate any precipitation measurements from rain gauges (in contrast to what the authors claim on p. 4, l. 107), but only precipitation from

satellite datasets and from ground-based radar

(<https://confluence.ecmwf.int/display/CKB/ERA5%3A+data+documentation#heading-Observations>). Hence, ERA5-based daily precipitation estimates represent modelled quantities and cannot really be termed "observations". The satellite dataset IMERG is based on multiple different sensors that are combined over different time periods, and based on micro-wave and IR measurements that only represent indirect estimates of precipitation.

Moreover, because the identified "hot spot regions" (from which the climate signal is mostly detected) lie over oceans (without any gauge data available for calibration), it appears unclear to what extent the detected change in high-frequency characteristics can be really attributed to observed climate change, as opposed to modelled changes (in case of ERA5), or hypothetical sensor changes, etc. (in IMERG). However, the results appear pretty consistent between both datasets, and are indeed very interesting; so I think it would be important to make clear that the study does not use direct precipitation measurements. If possible, I would strongly encourage the authors to provide an evaluation of both datasets against rain gauge data.

In addition, I am wondering whether the 1950-1979 period (in ERA5) can really be used as a reliable baseline for "natural climate variability". In standard D&A, detection or attribution is typically assessed against long control simulations. It is thus a bit unclear if a 30-year period is long enough (to also cover low-frequency variability) - and whether just before the post-1980 trend period consists of a good natural variability baseline. The pre-1979 ERA5 baseline also appears slightly problematic because 1950-1978 ERA5 is still considered "experimental", and satellite data assimilation only (to my knowledge) only starts in 1979 (hence a potential breakpoint/different statistics). I think it would be appropriate to compare the obtained trends *in addition* against long control simulations from different large ensembles, or from CMIP6 simulations.

In this context, I believe that the emergence of a fraction of days (such as 50%) from natural variability (the here presented results) is identical to "detection" (which is defined differently in an IPCC context, in a defined statistical sense), and I would recommend to clarify the terminology in this regard.

(2) Training dataset (consisting of only one climate model) and its generalisability to an out-of-distribution sample

The authors train the deep learning model on a large climate model ensemble, CESM2-LE. However, detection is then assessed on (reanalysed) ERA5 and (satellite-based) IMERG data. It is well known that machine learning models do not generalise well outside of the training distribution; and it remains unclear how much the algorithm performance decreases due to the test datasets. In addition, heterogeneous training data (such as from other large ensembles, or from CMIP6) may potentially help the machine learning algorithms to generalise.

Hence, would it be an option to provide a careful evaluation of the generalisation capabilities by, for instance, comparing the algorithm's performance on (a) CESM2-LE, (b) different other climate model ensembles (and also test whether "detection" works robustly across different models), and (c) the actual test datasets? I believe such type of analysis would help to understand better to what extent the final detection results can be trusted and interpreted.

(3) In-depth discussion, interpretation, and understanding of the results

The authors provide an interesting and commendable analysis using interpretability tools, and a time scale separation, to understand which time scales and which regions are relevant for the signal detection. However, it is imO not so easy for a reader to relate those results to physical theory or to literature on precipitation change. The authors illustrate those hot spot regions and show that it appears to be high-frequency variability that leads to detectability. However, it

remains somewhat elusive as to **why** specifically those regions stick out.

For example, why is the eastern tropical Pacific and northern South America standing out (in Fig. 2c), and not -say- the West Pacific? Is this because of internal variability in testing datasets (and the testing period), or because of a higher Signal/Noise ratio in the CESM2-LE training dataset, which is propagated through the DD model?

How come that northern South America contributes in a strongly positive way to the AGMT increase (Fig. 2), but does not show such an increase in the CESM2-LE training dataset? (Extended Data Figure 8) How does the DD model "know" that it's supposed to look in northern South America, if there is no trend in the standard deviation in the training dataset?

Does the positive contribution of the eastern tropical Pacific at high frequencies mean that the CESM2-LE training and testing dataset show a comparable/physically consistent change in high-frequency precipitation statistics in this region? Can this be shown? (e.g. would Fig. 4b pattern for model ensemble members look similar?)

Are these hotspot regions also well-known as hotspots of high-frequency precipitation change in the literature? Or could the robustness of results further tested if, say, the same occlusion sensitivity tests would be applied to individual ensemble members of CESM2-LE, other large ensembles or CMIP6? If results are robust, I think that (broadly) the same patterns would be expected to emerge.

(4) Methodological comments/questions:

- Occlusion sensitivity and time-scale separation: Because the DD method is highly non-linear, I wonder whether such a "linear" separation of time scales, and propagation with the DD model; or interpretation for only one region via occlusion sensitivity, is actually possible? Can the regions of decreasing linear trends really be analysed/interpreted in isolation? (e.g., p. 8, l. 185-188) Or is it not the case that a highly nonlinear and complex deep learning model also shows interrelationships/interactions in the spatial structure (or at different time scales)?

- Ridge regression: The authors note that the hyper-parameter of ridge regression is set to a pre-specified value. I believe that the standard practice for a ridge regression model is cross-validation - have the authors conducted some tests before setting the hyper-parameter to this specific value?

- Precipitation characteristics detected by Occlusion sensitivity method (Fig. 3): How are the percentiles, shown in Fig. 3 and discussed in the text, actually defined? In particular, why are low percentiles so different across decades? Isn't a low percentile of daily precipitation simply a day without precipitation?

Minor comments:

p. 2, l. 34L. Slightly confusing imO to say "warming will, **on average**, intensify rainfall variability" (but I know what you mean). Maybe better to say something along "warming will, in general, [...]"

p. 2, l. 56. "unquivocal"

p. 3, l. 65. ImO the word "also" does not really fit here - before you refer to 1-3% per degree of warming increase of average precipitation, and then you say the intensity is "also" projected to increase by 7% K⁻¹. What is "also" increasing by 7% K⁻¹?

p. 3, l. 66/67 I am not entirely sure if "this process **is constrained**" is a good wording here - clearly, the CC relationship governs increases of precipitation extreme intensities to first order, but whether it "constrains" (i.e., limits) increases to 7% K⁻¹ in all possible cases remains unclear, as, for example, super CC-scaling has been observed in short-term high-intensity events (see e.g. Lenderink et al 2017 Journal of Climate). Maybe say something around "CC-scaling governs this process", or so?

p. 3, l. 71-73. The statement "conclusions are limited by the uncertainties associated with synoptic and climate noise, as they cannot be clearly separated from the signal using available precipitation observations" appears a partly contradictory to the previous sentence, where the authors refer to a number of studies that all/most found a discernible imprint of external forcing on zonal/extreme precipitation statistics. Is the intention to say that making a crystal-clear separation of GHG-/AER- and internal signals is difficult due to these uncertainties?

p. 3, l. 75: I am not entirely sure whether ref. 19 talks about *daily* precipitation, but I may have missed it.

p. 3, l. 74-80: A third reason of why D&A is difficult for daily precipitation may well be that there does not exist observational daily precipitation estimates over the ocean (except for indirect satellite observations).

p. 5, l. 122-124: It appears a bit odd imO to present linear trends over % changes per decades in emergence days, because the phenomenon of emergence must be clearly non-linear over time.

p. 10, l- 242: It doesn't look like that the DD model's Occlusion sensitivity is indeed *monotonically* decreasing as a function of percentiles...

p. 10, l. 254: please clarify whether "the mean state changes remain virtually undetectable" refers to mean state changes of *daily precipitation* from daily variability, as many papers have looked at zonal and/or extreme statistics, which are indeed detectable

p. 13, l. 378/379: The dimensions and resolution can't match. 2.5° horizontal resolution would amount to 144 grid cells (not 160), and 60°S-75°N would correspond to 54 grid cells at 2.5° horizontal resolution.

p. 14, l. 403: The main text refers to a box of 15° x 15° for occlusion sensitivity, but if it's 7 x 7 grid cells, this would correspond to a 17.5° x 17.5° occlusion sensitivity box.

p. 16, l. 434 typo: "nomalized"

References:

Lenderink, G., Barbero, R., Loriaux, J.M. and Fowler, H.J., 2017. Super-Clausius–Clapeyron scaling of extreme hourly convective precipitation and its relation to large-scale atmospheric conditions. *Journal of Climate*, 30(15), pp.6037-6052.

Referee #2 (Remarks to the Author):

I like the aim of this paper: the authors make a clear case for deep learning in detecting the signals of climate change in noisy daily precipitation anomalies. I also very much appreciated the emphasis on moving away from traditional "fingerprinting" methods, which assume the climate response can be captured by a stationary spatial pattern. I think the use of a CNN is well-motivated, novel, and deserving of publication. I have three primary comments: two that are perhaps easily addressed, but one that may pose a major problem.

First, and perhaps most concerning: I am worried about the authors' use of ERA5 reanalysis data as "observed" precipitation. Reanalysis assimilates other meteorological variables but not precipitation, which is non-Gaussian and consequently requires more advanced and non-linear assimilation technique: precipitation is merely a byproduct, and numerous studies (ie Alexander et al <https://iopscience.iop.org/article/10.1088/1748-9326/ab79e2/meta>) have called the use of precipitation reanalysis into question. It's particularly striking to me that the correlations in Figure 1 are much weaker when IMERG data is used. Satellite datasets such as IMERG themselves may contain artifacts due to orbital drift, changing constellations, and other effects. I would like to see much more discussion of observational uncertainties in precipitation- while AGMT estimates are fairly robust across datasets (please specify which one you used), you are trying to infer global mean warming from very noisy and inconsistent high-dimensional datasets. It would be useful, for example, to see the differences in Figure 2a recalculated over the same time period. Are the major

discrepancies between ERA5 and IMERG trends due to the different time periods they cover or due to differences in the “observational” products themselves?

Second, I would have liked to see more physical discussion of the results. I like the use of occlusion sensitivity to identify “hot spots” very much, and I appreciate the differences between the linear trend in precipitation (in which temperature changes are not detectable) and HF precipitation. However the result that global warming is undetectable in low-frequency changes but very apparent in high-frequency precipitation needs a clearer tie-in to the “wet-get-wetter”/ “warmer-get-wetter” explanation in the introduction. I am a little unconvinced by the explanation that the observed temperature changes are not detectable in the observed precipitation trends because of the IPO phase; the observed anomalously cool conditions in the eastern Pacific clearly would affect precipitation trends but also strongly affect temperature trends as well. If the argument is that the real world experienced a specific pattern of temperature change that de-linked global warming and regional long-term precipitation trends, that should be explicitly stated. Additionally, while it is obviously far beyond the scope of this paper to attribute changes to any given forced response, it is likely worth at least mentioning that greenhouse gases, aerosols, and natural forcings likely affected the pattern of warming and the sensitivity of particular regions to the global temperature change.

Finally, because Nature is a journal with an extremely broad readership, some clearer explanation of the methodology is in order, particularly the role of the CESM LENS in training the CNN and the resulting temperature. This is a justifiable method, but it needs to be justified! Especially since Figure 4 and the surrounding discussion make it seem like you can get a pretty decent “traditional” fingerprint by just looking at the spatial pattern of changes in daily precipitation variance. I understand that the convolution process detects features using a 3x3 filter, but I am not clear exactly what that filter looks like at each step. I also understand the role of the pooling layers in downsampling, but I am still very confused about how we then end up with a scalar quantity that carries the units of temperature. Some more user-friendly explanation of this process, perhaps by analogy with computer vision, might be helpful here.

Minor comments

L59 was -> is

L62 response

L62 will also -> are also projected to

L65 or more, some studies show/project super-CC scaling due to increased moist ascent (for example <https://www.pnas.org/doi/pdf/10.1073/pnas.1800357115>)

L76 is “excessive” really the right word? Large is probably fine

L95 You say seven million pairs here, but in the Methods you say you subsample a lower number, so it might not be accurate or necessary to say this here

L 103 This seems like a little bit of a strawman argument- fingerprint studies don’t assume that the effects of climate change occur globally on a single day, rather that there is a spatial response pattern to external forcing that is stationary with time, and thus will become more strongly detectable in the observations with stronger forcing or longer time scales.

L106 Need to justify the use of reanalysis (see major point 1 above)

Fig 1a Which dataset is the observed AGMT from? This needs to be clearly stated and cited.

Fig 1a Dotted lines = dashed lines?

Fig 1a I am confused about what natural variability means here. Clearly the internal variability in the annual mean global temperature is much lower than this. Is it the random variation in the detectability of AGMT in precipitation observations due to things like different spatial patterns of warming? Or is it just the difference between daily temperatures and annual mean temperatures? Please clarify.

Fig 1b This is very confusing – I had to read the caption several times- and I’m not sure it’s telling

us anything Fig 1a isn't already?

Fig 1c the sudden leap in the number of emerged days following the reference period 1950-1979 is jarring- is this authentic, or an artifact of the training period?

L141-142 some focus on why the datasets disagree (time period? Or something else) would be useful (see major point 1)

L152 don't think "simulated" is the right word here- do you mean observed global warming is not detectable using only daily trends?

Fig 2b I'm not sure this is conveying any information not already in 2a.

Fig 2c What do the boxes mean? It says in text but not figure caption.

L187 Does this mean that the HF precipitation anomalies in these regions are changing a lot in a way and the DD model doesn't expect large changes, or that the DD model expects large changes that aren't showing up in the observations?

L197 How do you do occlusion sensitivity with ridge regression?

L205 This doesn't say anything about anthropogenic warming, just warming (I agree it's anthropogenic, but you're just doing detection here, not attribution)

Figure 4a Would be useful to show a map of linear trends in the variance or std of daily precipitation, perhaps in the supplementary material.

Author Rebuttals to Initial Comments:

Referee #1 (Remarks to the Author):

Summary:

The authors train a deep learning model to predict annual global mean temperatures from daily precipitation maps, based on a large climate model ensemble. The authors then apply the deep learning model to daily precipitation maps (from a reanalysis and a satellite dataset), and show that both show an increasing trend, and that in recent years a sizeable fraction (about half of the days) has emerged from the natural climate variability, thus indicating detection of forced climate change in daily precipitation at almost global scale. The authors further probe the results and find that the detection stems largely from a number of hot spot regions and from the high-frequency time scale, thus indicating that changes in precipitation variability (that have long been predicted from theory) would be detectable at the global scale. The results are, in my opinion, very interesting, and would add to and extend previous detection/attribution results of precipitation extremes to individual days (at the global scale). However, I also have a few concerns regarding (1) the observational datasets (which only provide indirect estimates of precipitation) and whether these datasets are "fit-for-purpose" for a detection/attribution study of daily precipitation, and the noise model of natural variability used for detection is problematic (only a 30-year period from ERA5), (2) the training dataset (consisting of only one climate model) and its generalisability to an out-of-distribution sample (the "observational" estimates), and (3) a more in-depth discussion, interpretation, and understanding of the results of the machine learning model applied to test datasets, and why in particular these hotspot regions stand out. In addition, I have a few comments/questions related to the methodology, outlined below.

: Thank you for your insightful comments concerning our manuscript. Those comments are all very valuable and helpful for improving our manuscript. We believe that our revised manuscript and new analysis further support our main arguments and clarify the points of concern raised by the reviewer.

Major comments

*(1) Observational datasets and detection from natural variability for very short reference period: The authors claim that "deep learning successfully detects the emerging climate change signals in daily precipitation fields during the observed record. [...] after applying the algorithm to the observational record, we found that the daily precipitation data represented an excellent predictor for the observed planetary warming [...]." (Abstract) However, the authors use a reanalysis dataset (ERA5) and a satellite-based precipitation record (IMERG) as "observational" input for the detection algorithm. This is a bit concerning, because ERA5 does *not* assimilate any precipitation measurements from rain gauges (in contrast to what the authors claim on p. 4, l. 107), but only precipitation from satellite datasets and from ground-based radar (<https://confluence.ecmwf.int/display/CKB/ERA5%3A+data+documentation#heading-Observations>). Hence, ERA5-based daily precipitation estimates represent modelled quantities and cannot really be termed "observations". The satellite dataset IMERG is based on multiple different sensors that are combined over different time periods, and based on micro-wave and*

IR measurements that only represent indirect estimates of precipitation. Moreover, because the identified "hot spot regions" (from which the climate signal is mostly detected) lie over oceans (without any gauge data available for calibration), it appears unclear to what extent the detected change in high-frequency characteristics can be really attributed to observed climate change, as opposed to modelled changes (in case of ERA5), or hypothetical sensor changes, etc. (in IMERG). However, the results appear pretty consistent between both datasets, and are indeed very interesting; so I think it would be important to make clear that the study does not use direct precipitation measurements. If possible, I would strongly encourage the authors to provide an evaluation of both datasets against rain gauge data.

: We agree that evaluating the precipitation datasets used in our study against direct precipitation measurements is necessary considering their critical role on our detection results. As the reviewer mentioned, the precipitation measurements from rain gauges are currently most reliable observations among all available datasets. Although the hot spot regions identified in our study are mostly over the ocean, the one located in the Atlantic storm track covers the eastern U.S. (Fig. 2c in the main manuscript), in which relatively sufficient number of rain gauge data is available (the CPC daily rain gauge data, <https://psl.noaa.gov/data/gridded/data.unified.daily.conus.html>).

The results from the rain gauge data are consistent with those from the reanalysis and satellite precipitation datasets: a robust increase in the magnitude of high-frequency (HF) precipitation variability during recent decades in the eastern U.S. (Figure A-1, A-2a, and A-2b). The difference map between [2001–2020] and [1960–1979] shows the robust variability increase after 2000s particularly over the northeast U.S (Roque-Malo and Kumar, 2017). On the contrary, over the western U.S., which is outside the Atlantic storm track hot spot, the HF precipitation variability does not show any trend in its magnitude. In addition, the HF variability change ratio is greater than the mean precipitation change ratio only over eastern U.S. (Figure A-2c and A-2d). This demonstrates that our main finding is supported by rain gauge measurements.

Next, even though the number of rain gauge data is relatively few, we checked the HF variability change in other hot spot regions using the GPCC global daily rain gauge data (<https://climatedataguide.ucar.edu/climate-data/gpcc-global-precipitation-climatology-centre>). The regions with an increase in the HF precipitation variability during the recent period reasonably match the hot spot regions (Figure A-3). Over the south America (SA), the increase in the HF precipitation variability is prominent over the northern and southern part, which are within the hot spot regions over the Amazon and Southern Ocean. Interestingly, there is a few rain gauge data over the Aleutian Islands within the north Pacific hot spot, in which the HF precipitation variability is significantly increased.

We think the new results based on the rain gauge data lend confidence on the main findings of our study. Figure A-1 and A-2 are included as main Figure 4d-4e and Extended Fig. 9 in the revised manuscript with the descriptions below. We have also noted that our detection and analysis is based on indirect estimates of daily precipitation in line 111–113 of the revised manuscript.

Line 273–284: Our results are further evaluated using direct precipitation measurements. Although the hot spot regions identified in our study are mostly over the ocean, the one located

in the Atlantic storm track covers the eastern U.S., where relatively large number of stations provide daily rain gauge data. The results from the rain gauge data are largely consistent with those from the satellite and reanalysis precipitation datasets, indicating a robust increase in the magnitude of HF precipitation variability during recent decades in the eastern U.S. (Fig. 4d, 4e, and Extended Data Fig. 9a and 9b). On the contrary, over the western U.S., which is outside the Atlantic storm track hot spot, the HF precipitation variability does not show an organized trend pattern. In addition, the greater change of HF variability than the mean precipitation change is only observed over eastern U.S. (Extended Data Fig. 9c and 9d). This rain gauge-based analysis increases robustness of our main findings.

Figure A-1. The difference in the standard deviation of the high-frequency (i.e., 10-day high-pass filtered) precipitation anomalies between (a) 2016–2020 from 1980–1984, (b) 2001–2020 from 1960–1979 calculated using CPC rain gauge data. Black box denotes the hot spot regions defined based on the occlusion sensitivity analysis in Fig. 2c.

Figure A-2. (a) The ratio of the standard deviation (std) of the high-frequency (i.e., 10-day high-pass filtered) precipitation anomalies during 2016–2020 to that during 1980–1984 in western U.S. (125°–110°W, 30°–50°N) and eastern U.S. (90°–55°W, 35°–50°N). (b) Same as (a), but the ratio of that 2001–2020 from 1960–1979. (c) Percentage change in precipitation variability during 2016–2020 compared to 1980–1984 (i.e., std change divided by std during 1980–1984) divided by the percent change in

precipitation climatology in the western and eastern U.S.. (d) Same as (c), but for the ratio between 2001–2020 from 1960–1979.

Figure A-3. The HF precipitation standard deviation change during 2015–2019 from 1983–1987 using GPCC daily precipitation data. The black box denotes the hot spot regions defined in Figure 2c of the main manuscript.

As well as the validation with the direct measurements, we tried to increase the credibility of our detection results by utilizing independent observations: MSWEP version 2.8 which merges satellite, gauge, and reanalysis data (Beck et al., 2017, 2019) and GPCP version 3.2, which merges satellite and gauge data (https://disc.gsfc.nasa.gov/datasets/GPCPDAY_3.2/summary). The AGMT estimates using two additional precipitation datasets exhibit consistent results to those using ERA5 or IMERG dataset; the annual-mean of the estimated AGMT is above the 95 % confidence range of the natural variability (Figure A-4a), and the number of the emergence day significantly increases in time (Figure A-4b). As the results using two additional precipitation datasets strengthen our main findings, we add those results in Figure 1 of the revised manuscript.

Figure A-4. (a) Time series of the observed annual global mean 2 m air temperature (AGMT) anomaly from 1980 to 2020 (black line) and the annual average of the estimated AGMT using daily precipitation fields from the MSWEP_v2.8 gauge-satellite-reanalysis merged observation (blue), GPCP_v3 satellite observation (orange), IMERG_v6 multi-satellites observation (purple) and ERA5 reanalysis (red) as inputs in the deep detection (DD) model. (b) Fractional number of emergence (EM) days within a corresponding year from 1980 to 2020 for which the estimated AGMT is greater than the upper bound of the two-tailed 95% confidence range of the natural variability. Note that the natural variability range is obtained from the historical simulation during 1850–1950 of the CESM2 LE (Please see the below responses for details of the estimating natural variability).

*In addition, I am wondering whether the 1950-1979 period (in ERA5) can really be used as a reliable baseline for "natural climate variability". In standard D&A, detection or attribution is typically assessed against long control simulations. It is thus a bit unclear if a 30-year period is long enough (to also cover low-frequency variability) - and whether just before the post-1980 trend period consists of a good natural variability baseline. The pre-1979 ERA5 baseline also appears slightly problematic because 1950-1978 ERA5 is still considered "experimental", and satellite data assimilation only (to my knowledge) only starts in 1979 (hence a potential breakpoint/different statistics). I think it would be appropriate to compare the obtained trends *in addition* against long control simulations from different large ensembles, or from CMIP6 simulations.*

: We fully understand the reviewer’s concern that the 1950–1979 period may be too short to set a reliable baseline of the nature variability. Therefore, by following the standard D&A

procedure, we obtain the confidence range of the natural variability from the long-term simulations. For this purpose, the CESM2 LE is the most suitable dataset as it provides tens of ensemble members to assess the natural variability. By utilizing the earlier period of the historical simulations (i.e., 1850–1950, following Sippel et al. (2020)), total 8,080-year-long integrations (i.e., 101 years \times 80 ensemble members) is obtained to set a range of the natural variability. Firstly, AGMT is estimated using daily output of 8,080-year of integrations, then, the 97.5th and 2.5th percentile value is set as the upper and lower bound of the natural variability range (i.e., with 95 % confidence level). The resulting upper and lower bound is 0.43 °C and -0.49 °C, respectively. As a result, our main conclusion is not affected by the changes in the natural variability range; the detection result is still rigorous (Figure A-4a) with the long-term simulation-based natural variability range.

We also calculated the range of the natural variability with the historical simulations of the CMIP6 output during 1850–1950 (see Table A-1 below for CMIP6 model details) with total 2,020 years of integrations (101 year \times 20 climate models). The upper bound of the natural variability from the CMIP6 ensemble is 0.56 °C, which is about 30 % greater than the one from CESM2 LE. Nonetheless, the detection results are still rigorous as the annual average of the estimated AGMT is increased about 1.1 °C in ERA5, and 0.9 °C in MSWEP from [1980–1984] to [2016–2020]. This shows that our detection result is robust to different types and periods of datasets to estimate the natural variability.

In this context, I believe that the emergence of a fraction of days (such as 50%) from natural variability (the here presented results) is identical to "detection" (which is defined differently in an IPCC context, in a defined statistical sense), and I would recommend to clarify the terminology in this regard.

: Thank you for pointing this out. We agree that our definition of detection was somewhat unclear. The emergence of a fraction of days needs to be compared with a certain threshold based on natural variability to have a similar meaning to the conventional signal detection. The issue here is to define the threshold that properly represents the natural variability in a scientific basis. For this purpose, we defined the natural variability range of the EM days using the bootstrap method. Firstly, the AGMT is estimated using daily output of CESM2 historical ensemble simulations for 1850–1950, then, the fractional EM days is calculated for each year (i.e., number of EM days/365 days). The 97.5th percentile value of the total 8,080 cases (i.e., 101 year \times 80 ensemble members), which can be considered as the upper bound of the 95 %. The upper bound of the 95 %, and 99 % confidence range of the natural variability of fractional EM days is 10.9 %, and 18.1 % respectively. Based on this calculation, 50 % of fractional EM days is clearly outside of the natural variability range, therefore, we can conclude that the global warming signal embedded in daily precipitation is clearly emerged. We added the natural variability range of the fractional EM days in line 137–139 of the revised manuscript.

In addition, we defined the natural variability range of the degree of the increase in the EM days using the bootstrap method. To test the significance of 20-year linear trend of the fractional EM days, a linear trend of the fractional EM days for 20-year segments is repeatedly calculated from CESM2 ensemble simulations for 1850–1950. We have moved 20-year windows with a 10-year interval, which provides 9 values of 20-year linear trend per each ensemble member and total 720 samples of 20-year linear trend of fractional EM days. The

estimated upper bound of the 95 % confidence range of the linear trend of fractional EM days for 20 years is about 3.27 %/decade. When similar calculations were performed for 41-year trends (with a 5-year interval, total 960 samples), we obtained 1.13 %/decade as the upper bound of the 95 % confidence range. As the linear trend of EM days during both 1980–2020 and 2001–2020 is much larger than its estimated natural variability range (black dashed line in Figure A-5a), we can argue that the observed increase in the EM days during recent periods, which indicates an emergence of the global warming signal in daily precipitation events, is statistically significant, and hence robust to the extent that such increase would have never occurred before 1950. On the other hand, the trend of the observed fractional EM day using the ridge regression is within the natural variability (Figure A-5b), which supports our notion that the deep learning provides a methodological framework to detect the global warming signal in the daily precipitation. We demonstrated this point in line 140–144 of the revised manuscript as follows, and the natural variability range of the linear EM-day trend is denoted in main Fig. 1c.

Line 140–144: The strong positive linear trends of EM days were found for all precipitation datasets: 16.3, and 17.1 % decade⁻¹ during 1980-2020 for the MSWEP and ERA5, respectively, and 21.3, and 16.5 % decade⁻¹ for IMERG and GPCP during 2001–2020, respectively (Fig. 1c). These trends also exceeded the internal variability ranges of the EM days trends (dashed line in Fig. 1c. see Methods and Materials).

Figure A-5. Linear trend of the number of fractional EM days during 1980–2020 in MSWEP and ERA5, and 2001–2020 in MSWEP, GPCP, IMERG, and ERA5 using (a) DD model, and (b) ridge regression model. The dashed line denotes a 95% confidence level of the natural variability of the EM days estimated using the 1850-1950 period of the historical simulations in CESM2 LE.

(2) *Training dataset (consisting of only one climate model) and its generalisability to an out-of-distribution sample* : The authors train the deep learning model on a large climate model ensemble, CESM2-LE. However, detection is then assessed on (reanalysed) ERA5 and (satellite-based) IMERG data. It is well known that machine learning models do not generalise well outside of the training distribution; and it remains unclear how much the algorithm performance decreases due to the test datasets. In addition, heterogenous training data (such as from other large ensembles, or from CMIP6) may potentially help the machine learning algorithms to generalise. Hence, would it be an option to provide a careful evaluation of the generalisation capabilities by, for instance, comparing the algorithm's performance on (a) CESM2-LE, (b) different other climate model ensembles (and also test whether "detection" works robustly across different models), and (c) the actual test datasets? I believe such type of analysis would help to understand better to what extent the final detection results can be trusted and interpreted.

: Thank you for the constructive suggestions. We first test the sensitivity of the detection results by using heterogeneous *training* dataset. That is, rather than CESM2 LE, CMIP6 multi-model historical + SSP3-7.0 simulations are utilized to train the deep learning model (Table A-1 for the list of CMIP6 models, now included as Extended Data Table 2). Total number of integration year is 5,020 (251 years \times 20 models). The deep learning model trained using CMIP6 simulations successfully detected the global warming signal in daily precipitation (Figure A-6); more than half of the estimated AGMT values after 2000s is above the natural variability range, while most values is within the natural variability range during the 1980s. The linear trend in the number of fractional EM days obtained from the CMIP6-trained CNN model is also comparable to that from the CESM2-trained CNN model. This indicates that the deep learning model results are largely insensitive to the different training datasets. This result is included in line 144–147 and Extended Data Figure 4 of the revised manuscript.

Next, we examine the sensitivity by using heterogeneous *testing* dataset (Figure A-7). With the deep learning model trained using CESM2 LE, the AGMT is estimated using daily precipitation of the multi models participated in CMIP6 simulations. Note that the CESM2 simulations submitted to CMIP6 is an independent simulation from those in CESM2 LE, therefore, it is worthwhile to be check for the testing even though the model is identical (Figure A-7d).

Intriguingly, an increasing trend in the estimated AGMT well mimics the true one even using the homogeneous testing samples; the annually-averaged AGMT estimations generally exhibit an increasing trend in time for most CMIP6 models with few exceptions. For example, the testing results using the EC-Earth3 are almost identical even though training samples are heterogeneous to the testing samples. The linear trend of the fractional EM days is above the natural variability range for all CMIP6 models except for one model, which clearly indicates that the detection works robustly across different models. We believe that these sensitivity tests support the generalization of our detection results using deep learning.

CMIP ID	Institution	Scenario	Ensemble
ACCESS-CM2	Commonwealth Scientific and Industrial Research Organisation, Australian Research Council Centre of Excellence for Climate System Science	Historical + SSP3-7.0	r1i1p1f1
ACCESS-ESM1-5	Commonwealth Scientific and Industrial Research Organisation	Historical + SSP3-7.0	r1i1p1f1
CanESM5	Canadian Centre for Climate Modeling and Analysis	Historical + SSP3-7.0	r1i1p1f1
CESM2	National Center for Atmospheric Research	Historical + SSP3-7.0	r1i1p1f1
CESM2-WACCM		Historical + SSP3-7.0	r1i1p1f1
CMCC-CM2-SR5	Fondazione Centro Euro-Mediterraneo sui Cambiamenti Climatici	Historical + SSP3-7.0	r1i1p1f1
EC-Earth3-AerChem	EC-Earth consortium	Historical + SSP3-7.0	r1i1p1f1
EC-Earth3		Historical + SSP3-7.0	r1i1p1f1
EC-Earth3-Veg		Historical + SSP3-7.0	r1i1p1f1
FGOALS-g3	Chinese Academy of Sciences	Historical + SSP3-7.0	r1i1p1f1
GFDL-ESM4	National Oceanic and Atmospheric Administration, Geophysical Fluid Dynamics Laboratory	Historical + SSP3-7.0	r1i1p1f1
INM-CM4-8	Institute for Numerical Mathematics, Russian Academy of Science	Historical + SSP3-7.0	r1i1p1f1
INM-CM5-0		Historical + SSP3-7.0	r1i1p1f1
IPSL-CM6A-LR	Institut Pierre Simon Laplace	Historical + SSP3-7.0	r1i1p1f1
MIROC6	Japan Agency for Marine-Earth Science and Technology, Atmosphere and Ocean Research Institute, National Institute for Environmental Studies, and RIKEN Center for Computational Science	Historical + SSP3-7.0	r1i1p1f1
MPI-ESM1-2-HR	Max Planck Institute for Meteorology	Historical + SSP3-7.0	r1i1p1f1
MPI-ESM1-2-LR		Historical + SSP3-7.0	r1i1p1f1
MRI-ESM2-0	Meteorological Research Institute	Historical + SSP3-7.0	r1i1p1f1
NorESM2-LM	Center for International Climate and Environmental Research, Norwegian Meteorological Institute, Nansen Environmental and Remote Sensing Center, Norwegian Institute for Air Research, University of Bergen, University of Oslo, and Uni Research	Historical + SSP3-7.0	r1i1p1f1
NorESM2-MM		Historical + SSP3-7.0	r1i1p1f1

Table A-1. Description of the climate models participated in Coupled Model Intercomparison Project phase 6 used for the sensitivity experiments.

Figure A-6. Same as Figure A-4, but with the deep learning model trained with CMIP6 historical + SSP3-7.0 dataset.

Redactions – unpublished data

(3) *In-depth discussion, interpretation, and understanding of the results*
*The authors provide an interesting and commendable analysis using interpretability tools, and a time scale separation, to understand which time scales and which regions are relevant for the signal detection. However, it is not so easy for a reader to relate those results to physical theory or to literature on precipitation change. The authors illustrate those hot spot regions and show that it appears to be high-frequency variability that leads to detectability. However, it remains somewhat elusive as to *why* specifically those regions stick out. For example, why is the eastern tropical Pacific and northern South America standing out (in Fig. 2c), and not -say- the West Pacific? Is this because of internal variability in testing datasets (and the testing period), or because of a higher Signal/Noise ratio in the CESM2-LE training dataset, which is propagated through the DD model?*

: Thank you for the thoughtful question. The signal-to-noise (STN) ratio of the change in the precipitation variability in CESM2 LE (i.e., ensemble-averaged precipitation variability trend divided by ensemble spread of precipitation variability trend) shows consistent results with the occlusion sensitivity (Figure A-8a). The STN ratio exhibits systematically larger values over the hot spot regions, such as the eastern Pacific ITCZ, and mid-latitude storm tracks, while those over the equatorial western Pacific is small. The spatial distribution of the STN ratio is mainly determined by those of the signal component (Figure A-8b). The noise component exhibits larger values over the tropical western Pacific and the equatorial central-eastern Pacific (Figure A-8c), indicating the changes in the HF variability over those regions are quite uncertain. On the contrary, noise components over the eastern Pacific ITCZ and the mid-latitudes is relatively small, confirming the high confidence in the increase in the HF variability due to the global warming, therefore, increase the in-situ STN ratio.

Figure A-8. (a) Signal-to-noise ratio (ensemble-averaged precipitation variability trend divided by ensemble spread of precipitation variability trend), (b) signal component, and (c) noise component of the high-frequency precipitation variability trend during 1980–2020 in CESM2 LE.

To understand the prominent variability increases of the high-frequency precipitation anomalies over the hot spot regions in details, we followed the framework in Zhang et al. (2021) which introduced a simple moisture budget analysis that the precipitation is dominated by vertical moisture advection as follows.

$$P_f \approx -\langle \omega \partial_p q \rangle_f \approx -\left(\frac{\omega_m q_l}{g}\right)_f \approx -\frac{\omega_m \bar{q}}{g} \quad (1)$$

where P , ω , and ω_m is precipitation, three-dimensional vertical velocity, and vertical velocity at mid-troposphere, respectively. $\langle \cdot \rangle = \frac{1}{g} \int_{p_s}^{p_t} dp$ denotes the vertical integral throughout the troposphere. q , q_l , and \bar{q}_l is three-dimensional moisture, low-tropospheric moisture anomaly, and climatological mean low-level moisture, respectively. The subscript f denotes variations at a specific time scale derived from the time filtering.

Eq. (1) can be modified to denote the variability and its change due to the global warming as follows.

$$\sigma[P_f] \approx \sigma\left[-\frac{(\omega_m)_f \bar{q}}{g}\right] = \frac{\sigma[-(\omega_m)_f] \bar{q}}{g} \quad (2)$$

$$\Delta\sigma[P_f] = \frac{\bar{q}_{l0}}{g} (\sigma[-(\omega_m)_{m1}]_f - \sigma[-(\omega_m)_{m0}]_f) + \frac{\sigma[-(\omega_m)_f]}{g} (\bar{q}_{l1} - \bar{q}_{l0}) \quad (3)$$

where σ denotes variability and Δ denotes the change due to the global warming. The subscript 0, and 1 denotes a historical, and future period, respectively. First term on the right side of Eq. (3) is referred to as the dynamic contribution as it depends on the change in vertical motion, and the second term is referred to as the thermodynamic contribution as it depends on the moisture change.

According to Eq. (3), well-known paradigms for global precipitation change is similarly applicable for the high-frequency precipitation variability changes. The historical moisture climatology term (\bar{q} will be referred as *HisC*) on the right-hand side refers the “wet-gets-more variable” paradigm, and the historical precipitation variability term ($\sigma[-(\omega_m)_f]$ will be referred as *HisV*) refers “variable-gets-more variable” paradigm. Given the strong coupling between the low-level moisture and the sea surface temperature, the climatological moisture change term ($\Delta\bar{q}$, will be referred as *Cchange*) presumably implies that the “warmer-gets-more variable”.

The regions with greater *HisC* (Figure A-9) are overlapped to hot spot regions (as in Fig. 2c of the main manuscript) to a large extent; *HisC* tends to be greater over the eastern Pacific ITCZ, northern South America, mid-latitude storm tracks than other regions. However, as *HisC* over the western Pacific is great as those in eastern Pacific ITCZ, it cannot explain why the equatorial western Pacific did not stand out as the hot spot region. This implies that the spatial distribution of *HisV* and *Cchange* might contribute to cause the differences between the western and eastern Pacific.

HisV exhibits a spatial coherence with the precipitation variability change, and shows a positive gradient over the equatorial Pacific (Figure A-10a and A-10b). Even though *Cchange* exhibited less coherences between the model simulation and the observations, relatively stronger

increases over the off-equatorial central-eastern Pacific than that over the equatorial western Pacific remain consistent (Figure A-10c and A-10d). This indicates that the reason why the western Pacific did not stand out as a hot spot region is explainable to some extent.

Aforementioned results consistently indicate that the hot spot regions shown in Figure 2c of the main manuscript are physically interpretable to some extent. As a reviewer suggested, to increase reader’s understanding, we provided physical interpretations in the revised manuscript why several specific regions stand out as hot spot regions in line 250–261 of the revised manuscript. In addition, the details of the moisture budget equation and its derivations are given in Methods and Materials section.

Line 250–261: The robust HF precipitation variability increases over the eastern Pacific ITCZ and mid-latitude storm tracks can be physically understood using a simple moisture budget analysis (see Methods and Materials). The mean precipitation and HF variability are both prominent over the eastern Pacific ITCZ, northern South America, mid-latitude storm tracks (Extended Data Fig. 8), which supports the “wet-gets-more variable”, and “variable-gets-more variable” paradigms, respectively. Even though the observed long-term trend can be contaminated by the recent negative IPO event, the mean precipitation is slightly increased over the eastern Pacific ITCZ region (Extended Data Fig. 5), where the amplitude of the negative IPO-related tropical SST anomalies exhibited a local minima. Therefore, our conclusion does not invalidate “warmer-gets-wetter”, and its similar paradigm for the HF variability (i.e., so-called “warmer-gets-more variable” paradigm).

Figure A-9. Climatological precipitation during the historical period (i.e., 1980–2020) in (a) MSWEP and (b) CESM2 LE.

Figure A-10. The high-frequency (10 day high-pass filtered) precipitation variability during the historical period (i.e., 1980–2020) in (a) MSWEP, and (b) CESM2 LE. Difference in the precipitation climatology during 2016–2020 from that during 1980–1984 in (c) MSWEP, and (d) CESM2 LE.

How come that northern South America contributes in a strongly positive way to the AGMT increase (Fig. 2), but does not show such an increase in the CESM2-LE training dataset? (Extended Data Figure 8) How does the DD model "know" that it's supposed to look in northern South America, if there is no trend in the standard deviation in the training dataset?

: Thank you for the good points. As a reviewer pointed out, the DD model only can capture the global warming signal embedded in the training samples. In this respect, the high-frequency (HF) precipitation variability increase should appear in the training samples over the hot spot regions.

To examine the detailed HF variability change over the northern South America (SA) (75–60°W, 10°S–10°N), the annually-averaged HF variability in the CESM2 LE simulations from 1850 to 2100 is calculated (Figure A-11). Interestingly, it does not show a monotonic change; the daily variability of HF precipitation remained similar from 1850 to 1920, and decreased from 1920 to 1960, then increased afterwards. This implies that the variability change over the northern SA would be dependent on the degree of the global warming. Once the trend is calculated using the period corresponding to the period for the satellite-based observations (i.e., 2001–2020), the increased HF variability is evident over the northern SA (Figure A-12). The HF variability increase over the northern SA, defined by those in 2071–2100 relative to 1976–2005, is also shown in multi-model analysis (Pendergrass et al. 2017). That means, our testing period (i.e., 1980–2020) mostly lies in the period with the increasing trend, and the DD model can recognize the increase in the HF variability over the northern SA as a global warming signal. To avoid the confusion, we matched the period for the trend of the HF variability in Extended Data Fig. 7 to the observed testing period (i.e., 2001–2020).

Figure A-11. Time-series of the annually averaged daily HF variability over northern South America (75–60°W, 10°S–10°N) from 1850 to 2100 in the CESM2 LE simulations.

Figure A-12. The difference in the HF precipitation variability during 2016–2020 from that during 2001–2005 in CESM2 LE.

Does the positive contribution of the eastern tropical Pacific at high frequencies mean that the CESM2-LE training and testing dataset show a comparable/physically consistent change in high-frequency precipitation statistics in this region? Can this be shown? (e.g. would Fig. 4b pattern for model ensemble members look similar?)

: The difference in the precipitation variability between two periods (i.e., [2016–2020] minus [2001–2005]) is compared between the CESM2 LE (Figure A-12) and the observations (Figure A-13). The increases in the precipitation variability are largely consistent over the hot spot regions in both CESM2 LE and the observations; the increase in the HF precipitation variability is prominent over the eastern Pacific ITCZ, northern South America, mid-latitude storm tracks. The only systematic difference is that the increase in the HF variability over the tropical Pacific

is further extended to the west in CESM2 LE. This would be because the observed negative IPO-related variability change in 2000s is not simulated in CESM2 LE. The high spatial consistency between the observations and the CESM2 LE confirms to some extent that the changes in the HF precipitation variability is consistent between the model simulations and the observations.

Figure A-13. The difference in the HF precipitation variability during 2016–2020 from that during 2001–2005 in the ERA5, MSWEP, GPCP, and IMERG.

Are these hotspot regions also well-known as hotspots of high-frequency precipitation change in the literature? Or could the robustness of results further tested if, say, the same occlusion sensitivity tests would be applied to individual ensemble members of CESM2-LE, other large ensembles or CMIP6? If results are robust, I think that (broadly) the same patterns would be expected to emerge.

: The detailed spatial distribution of hotspots of HF precipitation change has not been examined so far in the literature, while the general increase in the precipitation variability at shorter time scales is getting attention in recent studies (Pendergrass et al., 2017; Zhang et al., 2021). To examine the robustness of hotspot regions, the result is further evaluated by applying same occlusion sensitivity tests for CESM2 LE (Figure A-14a) and CMIP6 (Figure A-14b). In both CESM2 LE and CMIP6, the linear trend of the occlusion sensitivity tends to be high over the eastern Pacific ITCZ, north Pacific and Atlantic storm tracks, and southern Oceans along 50 °S. Pattern correlation between the linear trend of the occlusion sensitivity in CESM2 LE, and CMIP6 and that in the satellite/reanalysis (i.e., composite of MSWEP and ERA5) is 0.78, and 0.89, respectively. This is consistent with the strong coherence in the spatial distribution of the changes in HF precipitation variability between CESM2 LE and the observations, even though that in CESM2 LE is extended to the west compared to the satellite observations. This confirms the spatial coherences of hot spots with different testing dataset, demonstrating the robustness of the hotspot regions revealed by the occlusion sensitivity.

Figure A-14. The linear trend of the occlusion sensitivity in (a) CESM2 LE with 15 ensemble members and (b) 20 CMIP6 models during 1980–2020.

(4) Methodological comments/questions

- Occlusion sensitivity and time-scale separation: Because the DD method is highly non-linear, I wonder whether such a "linear" separation of time scales, and propagation with the DD model; or interpretation for only one region via occlusion sensitivity, is actually possible? Can the regions of decreasing linear trends really be analysed/interpreted in isolation? (e.g., p. 8, l. 185-188) Or is it not the case that a highly nonlinear and complex deep learning model also shows interrelationships/interactions in the spatial structure (or at different time scales)?

: Thank you for the insights. We agree that this is a very important point which should be validated. To examine whether the “linear” separation using the time-filtering holds to interpret the occlusion sensitivity (OS) results, we checked whether the summed linear trend of the OS by high-pass filtered anomalies and that by low-pass filtered anomalies is similar to that using unfiltered anomalies. In both ERA5 and MSWEP, the linear trend of the OS simulated by low-frequency components (i.e., period > 1 month) exhibited negative values over the equatorial central-eastern Pacific (Figure A-15a and A-15b). This might be due to the difference in the long-term tropical changes between the recent observations and model projections (Kosaka and Xie, 2013). In addition, the values over the northern mid-latitude storm tracks, Southern Ocean, and northern South America are opposite between two precipitation products, indicating the uncertainties in the contribution of low-frequency components on climate change detection results.

By adding the linear trend of the OS by low-frequency component (i.e., period > 1 month) and that by high-frequency component (i.e., period < 10 days, Figure A-15c and A-15d), the hot spots which is highlighted with high-frequency components are still prominent, while the negative value over the equatorial central Pacific is emerged due to the contribution of the low-frequency component (Figure A-16a and A-16b). Importantly, the spatial distribution of the summed value is quite similar to that by the unfiltered anomalies (Figure A-16c and A-16d), confirming that the linear separation of time scales to pin down the source of the successful detection is valid to some extent.

The decreasing linear trend of the OS over the subtropics mainly appears in the high-frequency, and it is cancelled by the low-frequency components (Figure A-15). This indicates that the climatological drying represented in the low-frequency component is properly captured as the global warming signal. As direct evidence, the AGMT response to the given precipitation input exhibited a negative relationship (Figure A-17a), which confirms that DD model captures the climatological drying as a global warming signal over the subtropics.

As the DD model is trained to capture the long-term trend changes over the subtropics, the decreasing OS over the subtropics attributed to the high-frequency component is likely to be a residual led by the variance change. Over the subtropics, the occurrence of the extreme precipitation events tends to be decreased in time (Figure A-17b); the PDF of the precipitation events at the extreme percentile (i.e., > 90 percentile, or < 10 percentile) is smallest in 2010s. According to the AGMT response shown in Figure A-17a, the reduction in the occurrence of the negative extremes contribute to decrease the AGMT, while that of the positive extremes contribute to increase the AGMT. That means that the AGMT responses to changes in two precipitation extreme occurrences are cancelled by each other. However, there is a residual in the time-averaged AGMT as the degree of AGMT change to the same given extreme event occurrence difference is much sensitive for the negative extremes (Figure A-17a). That is, the degree of the AGMT decrease by the less frequent occurrence of the negative extreme events is higher than the AGMT increase due to the less frequent positive extreme events. This indicates that the OS trend over the subtropics is decreased through the rectification of HF precipitation variability change, therefore, interpretable to some extent. This is briefly mentioned in line 285–299 of the revised manuscript.

Line 285–299: While the tropical and mid-latitude fingerprints in daily precipitation are dominated by HF variability change, those over the subtropical Atlantic and southeastern Pacific are overwhelmed by the climatological drying; this is in accordance with the occlusion sensitivity over the corresponding regions, which indicated positive values in the bottom percentiles and negative values in the top percentiles (Extended Data Fig. 10). ... Note that, due to the stronger positive sensitivity for the negative extreme percentile compared to the negative sensitivity for the positive extreme percentiles, the reduction in HF variance over the subtropical Atlantic (as shown in Fig. 4b) results in a pronounced reduction in AGMT estimation at negative extreme percentiles than the increase in AGMT at positive extreme percentiles. Consequently, this contributes to the negative occlusion sensitivity trend with HF precipitation input (Fig. 2c).

Figure A-15. Linear trend of the occlusion sensitivity using low-pass filtered (period > 1 month, upper panels) or high-pass filtered (period < 10 days, lower panels) precipitation anomalies for ERA5 (left), and MSWEP (right) during 1980-2020.

Figure A-16. Summed linear trend of the occlusion sensitivity by low-pass filtered anomalies (i.e., period > 1 month) and high-pass filtered anomalies (i.e., period < 10 days) for (a) ERA5, and (b) MSWEP during 1980–2020. Linear trend of the occlusion sensitivity using unfiltered precipitation anomalies for (c) ERA5, and (d) MSWEP during 1980–2020.

Figure A-17. (a) The occlusion sensitivity ($^{\circ}\text{C}/100$) with respect to the percentile of the high frequency (i.e., 10 day high-pass filtered) precipitation anomalies in the DD model (green) as well as the ridge regression model (black) in the subtropical Atlantic ($40\text{--}0^{\circ}\text{W}$, $20\text{--}35^{\circ}\text{N}$) and southeastern Pacific ($110\text{--}80^{\circ}\text{W}$, $35\text{--}20^{\circ}\text{S}$). (b) The ratios of the probability density function of the high-frequency (HF) precipitation anomalies during the 1980s, 1990s, 2000s, or 2010s to that during 1980-2020 in the corresponding regions.

- Ridge regression: The authors note that the hyper-parameter of ridge regression is set to a pre-specified value. I believe that the standard practice for a ridge regression model is cross-validation - have the authors conducted some tests before setting the hyper-parameter to this specific value?

: We conducted a sensitivity experiment by defining the value of hyperparameter λ of ridge regression differently as 0.01, 0.05, 0.1, and 0.5 using a training and validation dataset. As a result, the validation loss was the smallest when 0.1 was used as the λ value. Therefore, we set the λ value of ridge regression to 0.1 in this study. Note that the test dataset (i.e., the observed dataset) was not used in this sensitivity experiment. This point is mentioned in the revised manuscript as follows. “ λ is a hyperparameter that determines the penalty intensity, which is set to 0.1 after several experimentations to have smallest loss values for the validation dataset.”

- Precipitation characteristics detected by Occlusion sensitivity method (Fig. 3): How are the percentiles, shown in Fig. 3 and discussed in the text, actually defined? In particular, why are low percentiles so different across decades? Isn't a low percentile of daily precipitation simply a day without precipitation?

: Thank you for pointing this out. The detailed procedure to define the probability density function (PDF) ratio with respect to the percentile is as follows. (1) We flattened the dimensions of all-season HF precipitation anomalies in each hot spot region: three-dimensional array with [time, latitude, longitude] is converted to a one-dimensional array with [time \times latitude \times

longitude]. Here, HF precipitation anomalies include the cases without precipitation (i.e., days whose daily precipitation amount is zero). (2) We obtained percentile values and a PDF of HF precipitation anomalies for the entire period (i.e., from 1980 to 2020). (3) Next, PDFs of HF precipitation anomalies for each decade (i.e., the 1980s, 1990s, 2000s, and 2010s) were calculated based on the reference value for each percentile by using the whole period. (4) Finally, the PDF ratio was obtained by calculating the ratio of PDFs for each decade to that of the entire period.

To examine whether the relatively large difference in the lowest percentile is due to the changes in days without precipitation, we calculated the PDF ratio by excluding the days without precipitation (i.e., days whose daily precipitation amount is 0) (Figure A-18). It is found that the differences in the occurrence ratio between decades is still largest in the lowest percentile, indicating that the large decadal difference in the lowest percentile is not due to the changes in the number of days without precipitation. This implies the precipitation amount is decreasing for the lowest percentile rather than the number of days without precipitation is increasing.

To clarify the definition of the percentile, the procedure to calculate the PDF ratio is mentioned in Methods and Materials section as follows.

Line 560–566: The probability density function (PDF) of the daily precipitation anomalies with respect to its percentile is calculated for each decade. After arranging daily precipitation anomalies over certain regions during the whole period (i.e., 1980–2020) by their magnitudes, the values of precipitation for every 10th percentile are defined. The PDF of precipitation anomalies for each decade were calculated in the same way and then compared with the reference PDF value estimated by using the whole period for each percentile (Fig. 3).

Figure A-18. (a), and (b) denote the ratios of the probability density function of the HF precipitation anomalies during the 1980s, 1990s, 2000s, or 2010s to that during 1980–2020 in the eastern Pacific ITCZ, and mid-latitude storm tracks, respectively. Note that the cases without precipitation (i.e., days whose daily precipitation amount is 0) is excluded.

Minor comments:

*p. 2, l. 34L. Slightly confusing to say "warming will, *on average*, intensify rainfall variability" (but I know what you mean). Maybe better to say something along "warming will, in general, [...]"*

: Thank you for the suggestion. To reflect the reviewer's suggestion, the sentence of "greenhouse warming, on average, will intensify rainfall variability and extremes" is corrected to "greenhouse warming will intensify rainfall variability and extremes across the globe".

p. 2, l. 56. "unquivocal"

: It is corrected to "unequivocal".

p. 3, l. 65. The word "also" does not really fit here - before you refer to 1-3% per degree of warming increase of average precipitation, and then you say the intensity is "also" projected to increase by 7% K⁻¹. What is "also" increasing by 7% K⁻¹?

: Sorry for the confusion. The word "also" is deleted to avoid confusion.

*p. 3, l. 66/67 I am not entirely sure if "this process *is constrained*" is a good wording here - clearly, the CC relationship governs increases of precipitation extreme intensities to first order, but whether it "constrains" (i.e., limits) increases to 7% K⁻¹ in all possible cases remains unclear, as, for example, super CC-scaling has been observed in short-term high-intensity events (see e.g. Lenderink et al 2017 Journal of Climate). Maybe say something around "CC-scaling governs this process", or so?*

: Thank you for pointing this out. We agree that super-CC scaling should be mentioned for reviewing the global warming impact on extreme precipitation. The corresponding sentence is modified to indicate the super-CC scaling as follows. "... the intensity of extreme daily precipitation events is projected to increase at the rate of about 7 % K⁻¹ following the Clausius-Clapeyron (CC) relation in many parts of the world, while higher rates of increase have been observed regionally."

p. 3, l. 71-73. The statement "conclusions are limited by the uncertainties associated with synoptic and climate noise, as they cannot be clearly separated from the signal using available precipitation observations" appears a partly contradictory to the previous sentence, where the authors refer to a number of studies that all/most found a discernible imprint of external forcing on zonal/extreme precipitation statistics. Is the intention to say that making a crystal-clear separation of GHG- /AER- and internal signals is difficult due to these uncertainties?

: Sorry for the possible confusion. We intended to say that the detection of the precipitation imprints was only possible with pre-processed statistics (i.e., annual maxima, or seasonal/zonal averages) which suppresses the variability of uninterested temporal and spatial scales. While using spatial/temporal averages is beneficial for the detection because it lowers the uncertainty related to the natural internal variability, it is uncertain to what extent the detection results based on these smoothed fields can be applied to hydro-meteorological weather events that impact our daily lives. We modified the corresponding paragraph to clarify our main point.

*p. 3, l. 75: I am not entirely sure whether ref. 19 talks about *daily* precipitation, but I may have missed it.*

: Thank you for pointing this out. We found that ref. 19 is about the monthly-averaged anomalies. As it is not well matched to the given texts, ref. 19 is deleted in the revised manuscript.

p. 3, l. 74-80: A third reason of why D&A is difficult for daily precipitation may well be that there does not exist observational daily precipitation estimates over the ocean (except for indirect satellite observations).

: We noted that point in line 181–183 of the revised manuscript as follows “... due to the uncertainties ..., especially over the ocean where direct observations of precipitation is lacking.”

p. 5, l. 122-124: It appears a bit odd to present linear trends over % changes per decades in emergence days, because the phenomenon of emergence must be clearly non-linear over time.

: As a reviewer noted, the number of emergence (EM) days did not exhibit linear increase, but showed faster increase after 2000 (Figure A-19). To consider this feature, we calculated a quadratic trend in the EM days during 1980–2020 (dashed lines in Figure A-19). We found that the quadratic trend line fits greatly for the EM days during 1980–2020. The quadratic trend coefficient for the ERA5, and MSWEP is 0.042, and 0.043 %/year², respectively. As the 97.5th percentile of the natural variability of the quadratic trend coefficient is only 0.003 %/year², the quadratic trend in the EM days in ERA5 and MSWEP is also beyond the 95 % confidence range of the internal variability. This confirms that the main conclusion is still rigorous with the quadratic trend.

Even though the quadratic trend line fits better than the linear trend line for the fractional EM days during 1980–2020, we kept the original definition (i.e., linear trend coefficient) in Figure 1c, as a unit of the quadratic trend coefficient (i.e., %/year²) is hard to interpret physically, therefore, it might cause a confusion for the readers.

Figure A-19. Fractional number of emergence (EM) days within a corresponding year from 1980 to 2020. The red, and blue dashed line is the quadratic trend line for the EM days of ERA5, and MSWEP, respectively, and the black dashed line denotes that quadratic trend line of 97.5th percentile of the natural variability estimated by CESM2 LE.

*p. 10, l. 242: It doesn't look like that the DD model's Occlusion sensitivity is indeed *monotonically* decreasing as a function of percentiles...*

: Thank you for pointing this out. We deleted 'monotonically' from the revised manuscript.

*p. 10, l. 254: please clarify whether "the mean state changes remain virtually undetectable" refers to mean state changes of *daily precipitation* from daily variability, as many papers have looked at zonal and/or extreme statistics, which are indeed detectable*

: Sorry for the possible confusion. Detection using zonally- or seasonally-averaged precipitation was possible by reducing the natural variability through a zonal or seasonal averages. With the daily precipitation without any pre-processing, the mean precipitation change in the precipitation is not detectable. We clarify this point by revising the corresponding sentence as follows "The mean states changes remain virtually undetectable as it is hindered by the large internal day-to-day variability"

p. 13, l. 378/379: The dimensions and resolution can't match. 2.5° horizontal resolution would amount to 144 grid cells (not 160), and 60°S-75°N would correspond to 54 grid cells at 2.5° horizontal resolution.

: Sorry for the confusion. As the reviewer pointed out, the latitude range is 61.25°S–76.25°N with 55 grid cells, not 60°S–75°N. Meanwhile, it is correctly denoted that 160 longitudinal grid cells are used; we extended the gridded data to 40° by concatenating 0°–360°E and 0°–40°E (i.e., 144+16=160 grid points). This is to properly consider the precipitation signal around 0°E; for example, 357.5°E and 0°E are geographically close, but they are the furthest apart in the gridded map. This point is mentioned in the revised manuscript as follows: "The longitudinally extended data by concatenating 0°–360°E and 0°–40°E is to properly consider the precipitation pattern over around 0°E."

p. 14, l. 403: The main text refers to a box of 15° x 15° for occlusion sensitivity, but if it's 7 x 7 grid cells, this would correspond to a 17.5° x 17.5° occlusion sensitivity box.

: Thank you for pointing this out. It has a 17.5° × 17.5° size as a reviewer noted. To avoid the confusion, we only mentioned a number of grid points for the occlusion sensitivity.

p. 16, l. 434 typo: "nomalized"

: Corrected.

References

- Beck, H. E., Van Dijk, A. I., Levizzani, V., Schellekens, J., Miralles, D. G., Martens, B., & De Roo, A. (2017). MSWEP: 3-hourly 0.25 global gridded precipitation (1979–2015) by merging gauge, satellite, and reanalysis data. *Hydrology and Earth System Sciences*, 21(1), 589-615.
- Beck, H. E., Wood, E. F., Pan, M., Fisher, C. K., Miralles, D. G., Van Dijk, A. I., ... & Adler, R. F. (2019). MSWEP V2 global 3-hourly 0.1 precipitation: methodology and quantitative assessment. *Bulletin of the American Meteorological Society*, 100(3), 473-500.

- Holloway, C. E., & Neelin, J. D. (2009). Moisture vertical structure, column water vapor, and tropical deep convection. *Journal of the atmospheric sciences*, 66(6), 1665-1683.
- Kosaka, Y., & Xie, S. P. (2013). Recent global-warming hiatus tied to equatorial Pacific surface cooling. *nature*, 501(7467), 403-407.
- Pendergrass, A. G., Knutti, R., Lehner, F., Deser, C., & Sanderson, B. M. (2017). Precipitation variability increases in a warmer climate. *Scientific reports*, 7(1), 1-9.
- Roque-Malo, S., & Kumar, P. (2017). Patterns of change in high frequency precipitation variability over North America. *Scientific reports*, 7(1), 1-12.
- Sippel, S., Meinshausen, N., Fischer, E. M., Székely, E., & Knutti, R. (2020). Climate change now detectable from any single day of weather at global scale. *Nature climate change*, 10(1), 35-41.
- Zhang, W., Furtado, K., Wu, P., Zhou, T., Chadwick, R., Marzin, C., ... & Sexton, D. (2021). Increasing precipitation variability on daily-to-multiyear time scales in a warmer world. *Science advances*, 7(31), eabf8021.

Referee #2 (Remarks to the Author):

I like the aim of this paper: the authors make a clear case for deep learning in detecting the signals of climate change in noisy daily precipitation anomalies. I also very much appreciated the emphasis on moving away from traditional “fingerprinting” methods, which assume the climate response can be captured by a stationary spatial pattern. I think the use of a CNN is well-motivated, novel, and deserving of publication. I have three primary comments: two that are perhaps easily addressed, but one that may pose a major problem.

: Thank you for the encouraging comments and instructive comments. We believe that our revised manuscript and new analysis further supports our main arguments and clarifies the points of concern raised by the reviewer.

Major comments :

First, and perhaps most concerning: I am worried about the authors’ use of ERA5 reanalysis data as “observed” precipitation. Reanalysis assimilates other meteorological variables but not precipitation, which is non-Gaussian and consequently requires more advanced and non-linear assimilation technique: precipitation is merely a byproduct, and numerous studies (ie Alexander et al <https://iopscience.iop.org/article/10.1088/1748-9326/ab79e2/meta>) have called the use of precipitation reanalysis into question. It’s particularly striking to me that the correlations in Figure 1 are much weaker when IMERG data is used. Satellite datasets such as IMERG themselves may contain artifacts due to orbital drift, changing constellations, and other effects. I would like to see much more discussion of observational uncertainties in precipitation- while AGMT estimates are fairly robust across datasets (please specify which one you used), you are trying to infer global mean warming from very noisy and inconsistent high-dimensional datasets. It would be useful, for example, to see the differences in Figure 2a recalculated over the same time period. Are the major discrepancies between ERA5 and IMERG trends due to the different time periods they cover or due to differences in the “observational” products themselves?

: We fully understand the reviewer’s concern that detection is performed by using the reanalysis (i.e., ERA5) and the satellite-based indirect observation (IMERG). We tried to increase the credibility of our detection results by utilizing 1) additional precipitation products which merged satellite, gauge, and reanalysis data, and 2) direct rain gauge data.

Firstly, the detection results are supplemented by using two additional precipitation products; MSWEP version 2.8 from 1980 to 2020 (Beck et al., 2017, 2019), and GPCP version 3.2, from 2001 to 2020 (https://disc.gsfc.nasa.gov/datasets/GPCPDAY_3.2/summary). The AGMT estimates using two additional precipitation datasets exhibit consistent results to those using ERA5 or IMERG dataset; the annual-mean of the estimated AGMT is above the 95 % confidence range of the natural variability after mid-2010s (Figure B-1a), and the number of the emergence day significantly increases in time (Figure B-1b). The temporal correlation coefficient between the observed AGMT and the annually-averaged estimated AGMT during 1980-2020 is nearly same for the reanalysis product (ERA5) and the merged product (i.e., MSWEP) as 0.97, implying that the results is robust across the different products for relatively long-term period. For the period during 2001–2020, the temporal correlation becomes slightly lower; the temporal correlation between the observed AGMT and the estimated AGMT using

ERA5, IMERG, MSWEP, and GPCP is 0.83, 0.80, 0.75, and 0.76, respectively. The reanalysis (i.e., ERA5) exhibited slightly higher temporal correlations than that of the merged products (i.e., MSWEP, IMERG, GPCP), which is possibly because both the reanalysis and the training data are commonly based on outputs from global climate models.

To confirm our main finding is rigorous for four different precipitation products, we compared the results in Figure 2 (Figure B-2). In both the ERA5 and that of MSWEP during 1980–2020, the high-frequency (HF) component (i.e., period < 10 days) is responsible for the increase in the predicted AGMT in time (Figure B-2a). The importance of the HF component is still rigorous in the IMERG and GPCP data during 2001–2020 (Figure B-2b), which confirms our main finding is rigorous in four different precipitation products.

The AGMT estimation using the trend component exhibits the strong decrease in time for ERA5, and relatively weaker decrease in MSWEP, and GPCP, and the weak increase in IMERG. Shiogama et al. (2022) emphasized the differences between monthly precipitation trends during 1980–2014 between GPCP, MSWEP, and GSWP over some tropical regions where the density of rain-gauge observations is generally low. In the areas with few rain gauges, precipitation is estimated by combinations of spatial interpolations, satellite measurements, sounding observations and reanalyses, which have many methodological degrees of freedom, leading to discrepancies in the long-term trends among precipitation datasets. The systematic difference in the trend pattern/amplitude between the ERA5 and satellite precipitation data is also known (Nogueira, 2020).

To understand the opposite trend of the AGMT estimation in IMERG to the others, Figure B-3 shows the linear trend of the precipitation anomalies during 2001–2020 for IMERG and GPCP. Interestingly, positive linear trend is shown over the eastern ITCZ, mid-latitude storm tracks, and southern Ocean, which is matched to hot spot regions to some extent. In details, the amplitude of the positive linear trend over the hot spot region is generally larger in IMERG, except for the eastern ITCZ. On the other hand, the regions with positive linear trend in GPCP is too narrow over the north Atlantic, and not clear over the north Pacific and southern Ocean. The relatively high spatial coherence of the linear trend pattern with the hot spot region in IMERG might be responsible for the positive AGMT trend. While those discussions might be useful to understand the differences in the estimated AGMT trend using trend components between precipitation products, however, it is out of our main scope that the HF is the most crucial component for the successful detection. Therefore, we only briefly noted the cause of the different estimated AGMT trend in the revised manuscript as follows.

Line 159–163: When the linear trend component of precipitation anomalies was given to the DD model, the estimated AGMT decreased in time for both ERA5 and MSWEP dataset during 1980–2020 (Fig. 2a). During 2001–2020, the results with IMERG and GPCP dataset disagree on the sign of temporal changes in the estimated AGMT (Fig. 2b), possibly due to the discrepancies in the trend between the precipitation datasets (Shiogama et al., 2022).

Even though it is consistent for all precipitation products that the HF component is responsible for the successful detection, the degree of the AGMT increase due to the HF component in GPCP is systematically weaker than that in MSWEP or IMERG, and that in ERA5 is greater than the others. The weaker AGMT increase using GPCP can be understood by the instrumental

differences. Rui and Yunfei (2005) pointed out that the microwave emission signals are saturated at higher rain intensity, so that an emission-based algorithm, such as GPCP, cannot be performed well for estimating heavy rain. This can cause the underestimation of the heavy rain rate in GPCP, and possibly, IMERG, which merges the products of various instruments including precipitation radar, estimates heavy rain more accurately. The stronger AGMT increase in ERA5 also might be related to the frequency of heavy rainfall events; ERA5 overestimates the frequencies in the higher bins of the precipitation histograms than that of GPCP (Hassler and Lauer, 2021). This implies that the different frequency of the heavy rainfall might be related to the different degree of the AGMT increase due to HF components. We noted the point in line 178–180 of the revised manuscript.

Line 178–180: The dominant role of HF precipitation anomalies in yielding positive AGMT trend was found regardless of the precipitation input dataset used, while the degrees of the AGMT increase slightly differ between precipitation products.

The hot spot regions identified using the occlusion sensitivity exhibited high spatial consistency between precipitation products (Figure B-4). This indicates that the detection results and the physical mechanisms of the successful detection are robust across different precipitation products, despite uncertainties in precipitation products derived from various measurements. As the results using two additional precipitation datasets strengthen our main findings, we add those results in Figure 1 of the revised manuscript.

Figure B-1. (a) Time series of the observed annual global mean 2 m air temperature (AGMT) anomaly from 1980 to 2020 (black line) and the annual average of the estimated AGMT using daily precipitation fields from the MSWEP_v2.8 gauge-satellite-reanalysis merged observation (blue), GPCP_v3.2 satellite observation (orange), IMERG_v6 multi-satellites observation (purple) and ERA5 reanalysis (red) as inputs in the deep detection (DD) model. (b) Fractional number of emergence (EM) days within a corresponding year from 1980 to 2020 for which the estimated AGMT is greater than the upper bound of the two-tailed 95% confidence range of the natural variability.

Figure B-2. Linear trend of the estimated AGMT from (a) 1980–2020 (unit: $^{\circ}\text{C/decade}$) using ERA5 reanalysis (red) or MSWEP (blue), and (b) 2001–2020 using IMERG (purple) or GPCP (orange) with the unfiltered precipitation anomalies (denoted as total), with linear trends of the precipitation anomalies (denoted as trend), 10-day high-pass filtered (denoted as 10d HP), 10–30 day band-pass filtered (10–30d BP), 30–90 day band-pass filtered (30–90d BP), 90 day–1 year band-pass filtered (90d–1y BP), and 1 year low-pass filtered (1y LP) precipitation.

Figure B-3. Linear trend of the precipitation anomalies during 2001–2020 in (a) IMERG and (b) GPCP.

Figure B-4. The linear trend of the occlusion sensitivity for high-frequency precipitation using (a) ERA5, (b) MSWEP during 1980–2020, and (c) IMERG, and (d) GPCP during 2001–2020.

However, all precipitation products we utilized are still obtained from an indirect measurement. The precipitation measurements from rain gauges are currently the most reliable observations among all available datasets. Although the hot spot regions identified in our study are mostly over the ocean, the one located in the Atlantic storm track covers the eastern U.S. (Fig. 2c in the main manuscript), in which relatively sufficient number of daily rain gauge data is available (the CPC daily rain gauge data, <https://psl.noaa.gov/data/gridded/data.unified.daily.conus.html>).

The results from the rain gauge data are consistent with those from the reanalysis and satellite precipitation datasets: a robust increase in the magnitude of high-frequency (HF) precipitation variability during recent decades in the eastern U.S. (Figure B-5, B-6a, and B-6b). The difference map between [2001–2020] and [1960–1979] shows the robust variability increase after 2000s particularly over the northeast U.S. (Roque-Malo and Kumar, 2017). On the contrary, over the western U.S., which is outside the Atlantic storm track hot spot, the HF precipitation variability does not show any trend in its magnitude. In addition, the HF variability change ratio is greater than the mean precipitation change ratio only over eastern U.S. (Figure B-6c and B-6d). This demonstrates that our main finding is supported by rain gauge measurements.

Next, even though the number of rain gauge data is relatively few, we checked the HF variability change in other hot spot regions using the GPCC global daily rain gauge data (<https://climatedataguide.ucar.edu/climate-data/gpcc-global-precipitation-climatology-centre>). The regions with an increase in the HF precipitation variability during the recent period reasonably match the hot spot regions (Figure B-7). Over the south America (SA), the increase in the HF precipitation variability is prominent over the northern and southern part, which are within the hot spot regions over the Amazon and Southern Ocean. Interestingly, there is a few

rain gauge data over the Aleutian Islands within the north Pacific hot spot, and the HF precipitation variability is significantly increased.

We think the new results based on the rain gauge data lend confidence on the main findings of our study. Figure B-5 and B-6 are included as main Figure 4 and Extended Fig. 9 in the revised manuscript with the description in line 273-284 as follows.

Line 273-284: Our results are further evaluated using direct precipitation measurements. Although the hot spot regions identified in our study are mostly over the ocean, the one located in the Atlantic storm track covers the eastern U.S., where a relatively large number of stations provide daily rain gauge data. The results from the rain gauge data are largely consistent with those from the satellite and reanalysis precipitation datasets, indicating a robust increase in the magnitude of HF precipitation variability during recent decades in the eastern U.S. (Fig. 4d, 4e, and Extended Data Fig. 9a and 9b). On the contrary, over the western U.S., which is outside the Atlantic storm track hot spot, the HF precipitation variability does not show an organized trend pattern. In addition, the change of HF variability is greater than the mean precipitation change only over the eastern U.S. (Extended Data Fig. 9c and 9d). This rain gauge-based analysis increases robustness of our main findings.

Figure B-5. The difference in the standard deviation of the high-frequency (i.e., 10-day high-pass filtered) precipitation anomalies between (a) 2016–2020 from 1980–1984, (b) 2001–2020 from 1960–1979 calculated using CPC rain gauge data. Black box denotes the hot spot regions defined based on the occlusion sensitivity analysis in Fig. 2c.

Figure B-6. (a) The ratio of the standard deviation (std) of the high-frequency (i.e., 10-day high-pass filtered) precipitation anomalies during 2016–2020 to that during 1980–1984 in western U.S. (125° – 110° W, 30° – 50° N) and eastern U.S. (90° – 55° W, 35° – 50° N). (b) Same as (a), but the ratio of that 2001–2020 from 1960–1979. (c) Percentage change in precipitation variability during 2016–2020 compared to 1980–1984 (i.e., std change divided by std during 1980–1984) divided by the percent change in precipitation climatology in the western and eastern U.S.. (d) Same as (c), but for the ratio between 2001–2020 from 1960–1979.

Figure B-7. The HF precipitation standard deviation change during 2015–2019 from 1983–1987 using GPCC daily precipitation data. The black box denotes the hot spot regions defined in Figure 2c of the main manuscript.

Second, I would have liked to see more physical discussion of the results. I like the use of occlusion sensitivity to identify “hot spots” very much, and I appreciate the differences between the linear trend in precipitation (in which temperature changes are not detectable) and HF precipitation. However the result that global warming is undetectable in low-frequency changes but very apparent in high-frequency precipitation needs a clearer tie-in to the “wet-get-wetter”/ “warmer-get-wetter” explanation in the introduction. I am a little unconvinced by the explanation that the observed temperature changes are not detectable in the observed precipitation trends because of the IPO phase; the observed anomalously cool conditions in the eastern Pacific clearly would affect precipitation trends but also strongly affect temperature trends as well. If the argument is that the real world experienced a specific pattern of temperature change that de-linked global warming and regional long-term precipitation trends, that should be explicitly stated.

: As a reviewer pointed out, high-frequency precipitation variability is expected to be tied to the low-frequency precipitation change (mostly projected onto the trend) to some extent. However, low-frequency precipitation change is not the only factor to affect the high-frequency variability change, and those are the reason why the high-frequency variability change is prominent over hot spot regions, even though its trend is elusive in the observations.

Zhang et al. (2021) introduced a simple moisture budget equation to explain the precipitation variability change as follows.

$$P_f \approx -\langle \omega \partial_p q \rangle_f \approx -\left(\frac{\omega_m q_l}{g} \right)_f \approx -\frac{\omega_m \bar{q}}{g} \quad (1)$$

where P , ω , and ω_m is a precipitation, three-dimensional vertical velocity, and vertical velocity at mid-troposphere, respectively. $\langle \cdot \rangle = \int_{P_s}^{P_t} dp$ denotes the vertical integral throughout the troposphere. q , q_l , and \bar{q}_l is a three-dimensional moisture, low-tropospheric moisture anomaly, and climatological mean low-level moisture, respectively. The subscript f denotes variation at a specific time scale derived from the time filtering.

Eq. (1) can be modified to denote the variability and its change due to the global warming as follows.

$$\sigma[P_f] \approx \sigma \left[-\frac{(\omega_m)_f \bar{q}}{g} \right] = \frac{\sigma[-(\omega_m)_f] \bar{q}}{g} \quad (2)$$

$$\Delta \sigma[P_f] = \frac{\bar{q}_{l0}}{g} (\sigma[-(\omega_m)_{f1}] - \sigma[-(\omega_m)_{f0}]) + \frac{\sigma[-(\omega_m)_f]}{g} (\bar{q}_{l1} - \bar{q}_{l0}) \quad (3)$$

where σ denotes variability and Δ denotes the change due to the global warming. The subscript 0, and 1 denotes a historical, and future period, respectively. First term on the right side of Eq. (3) is referred to as the dynamic contribution as it depends on the change in vertical motion, and the second term is referred to as the thermodynamic contribution as it depends on the moisture change.

According to Eq. (3), well-known paradigms for global precipitation change is similarly applicable for the high-frequency precipitation variability changes. The historical moisture climatology term (\bar{q} will be referred as *HisC*) on the right-hand side refers the “wet-gets-

more variable” paradigm, and the historical precipitation variability term ($\sigma[-(\omega_{m0})_f]$ will be referred as *HisV*) refers “variable-gets-more variable” paradigm. Given the strong coupling between the low-level moisture and the sea surface temperature, the climatological moisture change term ($\Delta\bar{q}$, will be referred as *Cchange*) presumably implies that the “warmer-gets-more variable”.

As mentioned in the main manuscript, *Cchange* in the observations is quite different from the model projection results as the observed long-term trend is contaminated by the negative IPO event (Figure B-8). However, even though the equatorial precipitation differences exhibited the negative value due to the negative IPO-related La Nina-like signal, the positive precipitation anomalies are exhibited along the eastern Pacific ITCZ. This is closely related to the detailed spatial distribution of the negative IPO-related SST anomalies; the amplitude of the negative eastern Pacific SST anomalies shows a local minimum over the off-equator, and weak but positive precipitation anomalies are exhibited (Figure B-8, and see Fig. 2 and Fig. 3 of Medhaug et al., 2017). This means that our detection result does not regulate “warmer-gets-wetter”, or its similar paradigm for the HF variability (i.e., so-called “warmer-gets-more variable” paradigm).

In addition, both *HisC* (Figure B-9a and B-9b) and *HisV* (Figure B-9c and B-9d) show high spatial coherence with our hot spot region with a strong similarity between the observations and the model projection. For example, over the eastern ITCZ, northern South America, mid-latitude storm tracks, both *HisC* and *HisV* are clearly greater than other regions in both model simulation and the observations. Note that the similarity between the *HisC* and high-frequency variability change denotes that the “wet-gets-wetter” mechanism still holds for the high-frequency anomalies (i.e., so-called a “wet-gets-more variable” paradigm).

We provided aforementioned physical interpretations in line 250–261 of the revised manuscript as follows. The details of the moisture budget equation and its derivations are given in Methods and Materials section.

Line 250–261: The robust HF precipitation variability increases over the eastern Pacific ITCZ and mid-latitude storm tracks can be physically understood using a simple moisture budget analysis (see Methods and Materials). The mean precipitation and HF variability are both prominent over the eastern Pacific ITCZ, northern South America, mid-latitude storm tracks (Extended Data Fig. 8), which supports the “wet-gets-more variable”, and “variable-gets-more variable” paradigms, respectively. Even though the observed long-term trend can be contaminated by the recent negative IPO event, the mean precipitation is slightly increased over the eastern Pacific ITCZ region (Extended Data Fig. 5), where the amplitude of the negative IPO-related tropical SST anomalies exhibited a local minima. Therefore, our conclusion does not invalidate “warmer-gets-wetter”, and its similar paradigm for the HF variability (i.e., so-called “warmer-gets-more variable” paradigm).

Figure B-8. Difference in the precipitation climatology in (a) MSWEP during 2016–2020 from 1980–1984, and (d) CESM2 LE during 2016–2020 from that during 1980–1984.

Figure B-9. Climatological precipitation during the historical period (i.e., 1980–2020) in (a) MSWEP observation and (b) CESM2 LE. The high-frequency (10 day high-pass filtered) precipitation variability climatology during the historical period (i.e., 1980–2020) in (c) MSWEP and (d) CESM2 LE.

Additionally, while it is obviously far beyond the scope of this paper to attribute changes to any given forced response, it is likely worth at least mentioning that greenhouse gases, aerosols, and natural forcings likely affected the pattern of warming and the sensitivity of particular regions to the global temperature change.

: As a reviewer suggested, it is worthwhile to check the attribution of greenhouse gases, aerosol, and natural forcing to the HF precipitation variability changes. Figure B-10 shows the linear trend (unit: $\text{mm day}^{-1} \text{ decade}^{-1}$) of the annual standard deviation of the HF precipitation during 1980–2020 in the historical simulation with all forcings (Hist-ALL), those only with greenhouse gas forcing (Hist-GHG), only with aerosol forcing (Hist-AER), and only with solar and volcanic forcings (Hist-NAT) of the CESM2 submitted to CMIP6. The overall HF variability increase over the hot spot regions shown in Hist-ALL is mostly attributed to the greenhouse gases (Hist-GHG), while those due to aerosol, solar, and volcanic forcings are ignorable (Figure B-10a). This feature is also quite clear in each hot spot region Figure B-10b-

10f). Even though this information would be useful for the attribution study, however, we haven't noted this point in the revised manuscript due to the limitation of the number of the Extended Data figures.

Redactions – unpublished data

Finally, because Nature is a journal with an extremely broad readership, some clearer explanation of the methodology is in order, particularly the role of the CESM LENS in training the CNN and the resulting temperature. This is a justifiable method, but it needs to be justified!

: Thank you for the comments. We selected CESM2 LE as a training dataset as 1) it has a sufficient number of samples, and 2) it provides state-of-the-art performances in simulating characteristics of the daily precipitation in various time-scales. As CESM2 LE has tens of ensemble members, the total number of training samples is quite enough to train the deep learning model. In addition, it shows an outstanding performance in simulating the spatial distribution and the intensity of the mean precipitation over land (Li et al., 2022), ocean (Li et al., 2020), and global extreme precipitation (Abdelmoaty et al. 2021). As a result, CESM2 exhibited a good performance in simulating the overall shape of probability density function (PDF) of the daily precipitation (Martinez-Villalobos et al., 2022). In this respect, CESM2 LE can be one of the best choices to train the model to seek the changes in the characteristics of the global daily precipitation fields. We briefly describe the advantages of using CESM2 LE in Methods and Materials section as follows.

Line 569–576: ...(CESM2-LE), which has state-of-the-art skills in simulating characteristics of the daily precipitation in various time-scales. ... With the aids of tens of realizations for historical and global warming scenario simulations, the total number of samples used in training our DD model is larger than any other model simulation frameworks can provide, which is advantageous for training the deep learning model.

Nevertheless, as a single model may contain a relatively higher degree of the model error than the samples from multiple numbers of models, it is worthwhile to check whether the main results in this study would be still rigorous in a multi-model framework (i.e., CMIP6 models). A historical + SSP3-7.0 simulations from 20 CMIP6 models are utilized to train the deep learning model, therefore, total number of integration year is 5,020 (251 years × 20 models).

The deep learning model trained using CMIP6 successfully detected the global warming signal in daily precipitation (Figure B-11); more than half of the estimated AGMT after the 2000s is above the natural variability range, while most values are within the natural variability range during the 1980s. The linear trend of EM day also exhibited a similar degree with the model trained with the CESM2 LE. This indicates that the deep learning model results are largely insensitive to the type of training dataset. We have added this point in Line 171-175 of the revised manuscript.

Figure B-11. The detection results using the deep learning model trained with CMIP6 historical + SSP3-7.0 dataset.

Especially since Figure 4 and the surrounding discussion make it seem like you can get a pretty decent “traditional” fingerprint by just looking at the spatial pattern of changes in daily precipitation variance.

: One might test a traditional fingerprint method to the precipitation variance averaged for a

certain time period (e.g., annual mean of precipitation variance), then would come to the similar conclusion. However, the natural variability range would be dependent on the period for the time-averaging, which means that the detection using a traditional method is sensitive to the way of pre-processing. Detection using the deep learning is free from this issue by putting the daily precipitation without any pre-processing. As a result, the detection can be done in a much more elegant way.

More importantly, the DD model highlights the statistical moment containing the global warming signal among all possible moments through the nonlinear fitting. For example, as the detection is done by using daily precipitation fields which contain all statistical moments, we can conclude that the variance change is the most pronounced global warming signal of all moments including the climatological changes.

In addition, the CNN can show a superior performance to the traditional methods due to the partial translation invariance feature of the CNN. In the first stage of the CNN, the convolution operation is performed on the input to give linear activations. In the second stage, the resultant activations are passed through a non-linear activation function such as Sigmoid, Tanh or Relu. The product after the second stage is called as a feature map. In the third stage, the max-pooling operation is performed to reduce the dimension by selecting the maximum value for small patches ($2 \text{ grid} \times 2 \text{ grid}$ in this study) of a feature map. For example, the result of the max-pooling process for $[1 \ 0 \ 0 \ 0]$, $[0 \ 1 \ 0 \ 0]$, $[0 \ 0 \ 1 \ 0]$, and $[0 \ 0 \ 0 \ 1]$ are all 1. This indicates that slight shifts or changes in the original input map does not affect the output (a.k.a., partial translation invariance).

This partial translation invariance in the CNN allows to focus on the regional features of the variability increase between the model simulation and the observation, while acquiescing the grid-by-grid differences. In this study, size of the patch for the max-pooling is $2 \text{ grid} \times 2 \text{ grid}$ ($5^\circ \times 5^\circ$) and we repeated the max-pooling for 2 times. As a result, a maximum 10° differences in the peak location of the variability increase between the model simulation and the observations can be still recognized as the consistent regional climate change signal.

As shown in Figure B-12, even though the increase in the high-frequency precipitation variability is consistently shown over the off-equatorial eastern Pacific in both observations and the model simulation, but the peak location of variability increase in the model simulation is shifted to the south roughly about 10° to the observed. Similarly, over the north Atlantic and Pacific, the variability increase is clearly shown in both observation and model simulation, while its peak location is different in a grid scale. In a traditional fingerprint method, the fingerprint pattern is projected onto the input map through a grid-by-grid multiplication, the slight differences in the variability increase between the model simulation and the observation would significantly degrade the detection result.

As aforementioned discussion would be useful for the readers to understand the systematic differences in the traditional fingerprint method and the Deep Detection method, we added the following sentences in the revised manuscript.

Line 103–106: In addition, with its translation invariant feature, the DD model can extract common change patterns due to the global warming in both the model simulations and the observations in spite of their systematic differences.

Figure B-12. Difference of the high-frequency precipitation variability during 2016–2020 from that during 2001–2005 in (a) MSWEP, GPCP, IMERG, and ERA5, and (b) CESM2 LE.

I understand that the convolution process detects features using a 3x3 filter, but I am not clear exactly what that filter looks like at each step. I also understand the role of the pooling layers in downsampling, but I am still very confused about how we then end up with a scalar quantity that carries the units of temperature. Some more user-friendly explanation of this process, perhaps by analogy with computer vision, might be helpful here.

: Figure B-13 shows the examples of the convolutional filter at 1st convolutional layer in deep detection (DD) model. The spatial distribution of the convolutional filter is normally extremely hard to be understood physically. That is why the deep learning model is called a black box, and the interpretable method, such as occlusion sensitivity, is often used to understand the results produced by deep learning model.

The details of the dimension reduction in the DD model is as follows. As a reviewer mentioned, the pooling process reduces the horizontal dimension of the input map, which was 160×55, to 40×16. As we utilized 16 convolutional filters in the last convolutional layer, the dimension of the output after the convolutional processes is 40x14x16. Then, all the grid points (total of 40×16×16=8,960) is connected to the 1st dense layer with 32 neurons, then, 1st dense layer is connected to 2nd dense layer with a single neuron to produce a final output (i.e., scalar value). That means, in addition to the pooling process, the dimension is further reduced by linking feature maps to the dense layer with few neurons. The dimension of the output for each layer in the DD model is provided in the square brackets of the Extended Data Fig. 2, and its detailed

description is given in the Method section of the revised manuscript as follows.

Line 452–458: Through five convolutional and two max-pooling processes, the horizontal dimension of the feature map is reduced to 40×14 . As the last convolutional layer uses 16 convolutional filters, the size of the dimension of the final feature map is 8,960 (i.e., $40 \times 14 \times 16$). Then, each element of the final feature map is connected to the 1st dense layer with 32 neurons, and lastly, 1st dense layer is connected to the 2nd dense layer with a single neuron to output a scalar value representing the AGMT anomaly of the corresponding year.

Figure B-13. The spatial distribution of 32 convolutional filters in the 1st convolutional layer.

Minor comments :

L59 was -> is

: Corrected

L62 response

: Corrected

L62 will also -> are also projected to

: Corrected

L65 or more, some studies show/project super-CC scaling due to increased moist ascent (for example <https://www.pnas.org/doi/pdf/10.1073/pnas.1800357115>)

: Thank you for pointing this out. We agree that super-CC scaling should be mentioned for reviewing the global warming impact on the extreme precipitation. The corresponding sentence

is modified to indicate the super-CC scaling and we included Nie et al. (2018) as a reference as follows. "... the intensity of extreme daily precipitation events is projected to increase at the rate of about 7 % K⁻¹ following the Clausius-Clapeyron (CC) relation in many parts of the world, while higher rates of increase have been observed regionally (Nie et al., 2018)."

L76 is "excessive" really the right word? Large is probably fine

: Corrected

L95 You say seven million pairs here, but in the Methods you say you subsample a lower number, so it might not be accurate or necessary to say this here

: Thank you for the suggestion. The phrase to specify the detailed number of samples is deleted from the main text, and moved to the Method section.

L 103 This seems like a little bit of a strawman argument- fingerprint studies don't assume that the effects of climate change occur globally on a single day, rather that there is a spatial response pattern to external forcing that is stationary with time, and thus will become more strongly detectable in the observations with stronger forcing or longer time scales.

: Thank you for pointing this out. The corresponding sentence is corrected as follows "This is in contrast to the existing D&A techniques that detect climate change signals based on a global stationary fingerprint pattern."

L106 Need to justify the use of reanalysis (see major point 1 above)

: We demonstrated the results obtained from reanalysis is supported by the rain gauge data and the satellite precipitation observations to some extent. Please see the response to major point 1.

Fig 1a Which dataset is the observed AGMT from? This needs to be clearly stated and cited.

: We utilized HadCRUT5 data to calculate AGMT. It is stated in the Method section with a reference.

Fig 1a Dotted lines = dashed lines?

: Corrected.

Fig 1a I am confused about what natural variability means here. Clearly the internal variability in the annual mean global temperature is much lower than this. Is it the random variation in the detectability of AGMT in precipitation observations due to things like different spatial patterns of warming? Or is it just the difference between daily temperatures and annual mean temperatures? Please clarify.

: Thank you for the helpful insights. We found that (1) the random variations in the detectability of AGMT in daily precipitation fields using the DD model and (2) the day-to-day variation of the GMT during the historical period contribute to the natural variability here. The differences in variations of the daily GMT and that of yearly GMT (i.e., AGMT) is ignorable. Factor (1) comes from the uncertainties in estimating the AGMT using the daily precipitation, while factor (2) comes from the daily variability of the GMT. To quantify the amplitude of estimated AGMT variation due to (1) and (2), Figure B-14 shows the standard deviation (STD) of the observed and estimated AGMT, daily GMT, and estimated daily AGMT. The STD of the estimated daily GMT is 0.55, and that of the observed daily GMT (0.29) or observed AGMT

(0.26) is about half. That is, factor (1) significantly enlarge the estimated AGMT variation. We clarify in the revised manuscript that the random variations in the detectability of AGMT in daily precipitation fields using the DD model contributes to the internal variability change as follows.

Line 126–130: To measure the detectability of the observed AGMT variations associated with daily precipitation fields using the DD model, we defined the internal variability range ...

Figure B-14. (a) Variance of the observed AGMT (left), and annual-mean estimated AGMT (right). (b) Variance of the observed daily GMT (left) and the estimated daily AGMT (right).

Fig 1b This is very confusing – I had to read the caption several times- and I’m not sure it’s telling us anything Fig 1a isn’t already?

: As a reviewer pointed out, all the information in Fig. 1b is already given in Fig. 1a. To avoid the possible confusion, we removed Fig. 1b.

Fig 1c the sudden leap in the number of emerged days following the reference period 1950-1979 is jarring- is this authentic, or an artifact of the training period?

: As the period for the training dataset spans from 1850 to 2100, and all of them is obtained from model simulations (i.e., CESM2 LE), the sudden leap shown in the observation cannot be caused by any artifact in the training period.

The sudden leap after 1979 in the observations is shown not only in the emergence days, but also in annual averages of the estimated AGMT (thick colored lines in Fig. 1a). In addition, interestingly, the sudden leap after 1980 is also evident for the true AGMT (thick black line in Fig. 1a, also see <https://www.metoffice.gov.uk/hadobs/hadcrut5/>), indicating almost no trend in AGMT during 1950–1978, which is due partly to the aerosol cooling offsetting greenhouse warming together with no strong volcanic eruptions (e.g., Gillett et al. 2021). As none of observations is newly injected as of 1980, this leap is less possible to be caused by the artifact by the different temporal coverage of the input observations to generate HadCRUT5. Instead, the sudden increase of the AGMT after 1979 is attributed to the phase shift of the Interdecadal Pacific Oscillation (IPO) to some extent (Meehl et al., 2016).

It is definitely worthwhile to note a possible reason for the sudden leap if we kept the original time-series in Fig. 1. However, as Fig. 1 is revised by cropping the results during 1950–1978 of using the preliminary version of the ERA5 (<https://cds.climate.copernicus.eu/cdsapp#!/dataset/reanalysis-era5-single-levels-preliminary-back-extension?tab=overview>), this is not an issue to be pointed out in new Fig. 1.

L141-142 some focus on why the datasets disagree (time period? Or something else) would be useful (see major point 1)

: Thank you for pointing this out. The systematic difference in the trend component in various precipitation products is likely due to the differences in the precipitation characteristics in each product. Shiogama et al. (2022) emphasized that the differences between monthly precipitation trends during 1980–2014 between GPCP, MSWEP, and GSWP are found in some tropical regions where the density of rain-gauge observations is generally low. In the areas with few rain gauges, precipitation is estimated by combinations of spatial interpolations, satellite measurements, sounding observations and reanalyses, which have many methodological degrees of freedom, leading to discrepancies in the long-term precipitation trends among precipitation datasets. In addition, the systematic difference in the trend pattern/amplitude between the ERA5 and satellite precipitation data is also known (Nogueira, 2020). Those points are briefly mentioned as follows.

Line 161–163: During 2001–2020, the results with IMERG and GPCP dataset disagree on the sign of temporal changes in the estimated AGMT (Fig. 2b), possibly due to the discrepancies in the trend between the precipitation datasets.

L152 don't think "simulated" is the right word here- do you mean observed global warming is not detectable using only daily trends?

: Thank you for the suggestion. The corresponding sentence is modified as a reviewer suggested.

Fig 2b I'm not sure this is conveying any information not already in 2a.

: Thank you for the suggestion. As all the information in Fig. 2b is already given in Fig. 2a, we replaced the original Fig. 2b to show the sensitivity experiments in the additional observations.

Fig 2c What do the boxes mean? It says in text but not figure caption.

: Sorry for the confusion. We mentioned in the figure caption that boxes denote hot spot regions in which a strong positive trend appears.

L187 Does this mean that the HF precipitation anomalies in these regions are changing a lot in a way and the DD model doesn't expect large changes, or that the DD model expects large changes that aren't showing up in the observations?

: Thank you for pointing this out. The HF precipitation variability in subtropics is expected to be decreased due to the global warming in both observations and the CESM2 LE (Figure B-12). To capture this feature properly, the occlusion sensitivity to the precipitation input should have exhibited an inverted V-shape pattern. Rather, the linear inverse relationship between the

occlusion sensitivity and the precipitation is shown (Figure B-15), indicating that the DD model captures the climatological drying as the most prominent global warming fingerprint over the subtropics. As the observations exhibited a climatological drying in recent decade as in the CESM2 simulations, the occlusion sensitivity is slightly increased in time with the low-frequency (i.e., period > 1 month) precipitation anomalies (Figure B-16).

As the DD model is trained to capture the long-term trend changes over the subtropics, the decreasing trend of the occlusion sensitivity over the subtropics attributed to the high-frequency component is likely to be a residual led by the variance change; According to the occlusion sensitivity in Figure B-15, the degree of the AGMT decrease by the less frequent occurrence of the negative extreme events is higher than the AGMT increase due to the less frequency positive extreme events. This indicates that the occlusion sensitivity trend over the subtropics is decreased through the rectification of HF precipitation variability change. This is mentioned in line 285–299 of the revised manuscript.

Line 285–299: While the tropical and mid-latitude fingerprints in daily precipitation are dominated by HF variability change, those over the subtropical Atlantic and southeastern Pacific are overwhelmed by the climatological drying; this is in accordance with the occlusion sensitivity over the corresponding regions, which indicated positive values in the bottom percentiles and negative values in the top percentiles (Extended Data Fig. 10). ... Note that, due to the stronger positive sensitivity for the negative extreme percentile compared to the negative sensitivity for the positive extreme percentiles, the reduction in HF variance over the subtropical Atlantic (as shown in Fig. 4b) results in a pronounced reduction in AGMT estimation at negative extreme percentiles than the increase in AGMT at positive extreme percentiles. Consequently, this contributes to the negative occlusion sensitivity trend with HF precipitation input (Fig. 2c).

Figure B-15. The occlusion sensitivity ($^{\circ}\text{C}/100$) with respect to the percentile of the high-frequency (i.e., 10 day high-pass filtered) precipitation anomalies in the DD model (black), and the ridge regression model (gray) in the subtropical Atlantic ($40^{\circ}\text{--}0^{\circ}\text{W}$, $20^{\circ}\text{--}35^{\circ}\text{N}$) and southeastern Pacific ($110^{\circ}\text{--}80^{\circ}\text{W}$, $35^{\circ}\text{--}20^{\circ}\text{S}$).

Figure B-16. Linear trend of the occlusion sensitivity using low-pass filtered (period > 1 month) precipitation anomalies for (a) ERA5, and (b) MSWEP during 1980–2020.

L197 How do you do occlusion sensitivity with ridge regression?

: The linear trend of the occlusion sensitivity using unfiltered precipitation anomalies with ridge regression exhibits positive values over the off-equatorial far-eastern Pacific, equatorial Atlantic Ocean, tropical western Indian Ocean (Figure B-17a), which is overlapped to the regions with the increased climatological precipitation to some extent (Figure B-17b). It also shows prominent negative values over the equatorial central Pacific and mid-South Africa, which is overlapped to the decreased climatological precipitation trend. This spatial coherence of the occlusion sensitivity trend and the climatological precipitation changes indicates that the ridge regression detects the changes associated with the climatological component. On the other hand, the linear trend of the occlusion sensitivity using high-frequency anomalies exhibited a noisy pattern (Figure B-18), and is not matched to the variability changes shown in the CESM2 LE and the observations (Figure B-12). This again confirmed that the high-frequency precipitation variability change is hardly detectable using the ridge regression.

Figure B-17. (a) The linear trend of the occlusion sensitivity of the unfiltered precipitation anomalies from ERA5 and MSWEP during 1980–2020 using the ridge regression. (b) Climatological precipitation during 2016–2020 subtracted by that during 1980–1984 in MSWEP and ERA5.

Figure B-18. The linear trend of the occlusion sensitivity of the high-frequency (i.e., period < 10 days) precipitation anomalies from ERA5 and MSWEP during 1980–2020 using the ridge regression.

L205 This doesn't say anything about anthropogenic warming, just warming (I agree it's anthropogenic, but you're just doing detection here, not attribution).

: We agree to the reviewer's opinion. The corresponding phrase is corrected from 'anthropogenic warming' to 'global warming'.

Figure 4a Would be useful to show a map of linear trends in the variance or std of daily

precipitation, perhaps in the supplementary material.

: Thank you for the suggestion. As a reviewer pointed out, Figure 4a is much closely linked to the variability change, rather than the extreme event occurrence frequency. To increase the consistency between panels in Figure 4, Figure 4b is changed to show the difference in the std of HF daily precipitation (also shown in Figure B-19). Note that the spatial patterns of the HF standard deviation change are quite similar to those of the extreme precipitation event change, therefore, the results is still rigorous with the new Figure 4b; the increase in the HF precipitation variability is shown in the eastern Pacific ITCZ, northern South America, mid-latitude storm tracks over both hemispheres.

Figure B-19. The difference in the HF precipitation variability during 2016–2020 from that during 2001–2005 averaged for the MSWEP, ERA5, GPCP, and IMERG datasets.

References

- Abdelmoaty, H. M., Papalexiou, S. M., Rajulapati, C. R., & AghaKouchak, A. (2021). Biases beyond the mean in CMIP6 extreme precipitation: A global investigation. *Earth's Future*, 9(10), e2021EF002196.
- Gillett, N. P., Kirchmeier-Young, M., Ribes, A. et al. (2021) Constraining human contributions to observed warming since the pre-industrial period. *Nature Climate Change*, 11, 207–212.
- Hassler, B., & Lauer, A. (2021). Comparison of reanalysis and observational precipitation datasets including ERA5 and WFDE5. *Atmosphere*, 12(11), 1462.
- Medhaug, I., Stolpe, M. B., Fischer, E. M., & Knutti, R. (2017). Reconciling controversies about the ‘global warming hiatus’. *Nature*, 545(7652), 41-47.
- Li, J. F., Xu, K. M., Richardson, M., Lee, W. L., Jiang, J. H., Yu, J. Y., ... & Liang, H. C. (2020). Annual and seasonal mean tropical and subtropical precipitation bias in CMIP5 and CMIP6 models. *Environmental Research Letters*, 15(12), 124068.
- Li, Z., Liu, T., Huang, Y., Peng, J., & Ling, Y. (2022). Evaluation of the CMIP6 precipitation simulations over global land. *Earth's Future*, 10(8), e2021EF002500.

- Martinez-Villalobos, C., Neelin, J. D., & Pendergrass, A. G. (2022). Metrics for Evaluating CMIP6 Representation of Daily Precipitation Probability Distributions. *Journal of Climate*, 1-79.
- Meehl, G. A., Hu, A., Santer, B. D., & Xie, S. P. (2016). Contribution of the Interdecadal Pacific Oscillation to twentieth-century global surface temperature trends. *Nature Climate Change*, 6(11), 1005-1008.
- Nogueira, M. (2020). Inter-comparison of ERA-5, ERA-interim and GPCP rainfall over the last 40 years: Process-based analysis of systematic and random differences. *Journal of Hydrology*, 583, 124632.
- Roque-Malo, S., & Kumar, P. (2017). Patterns of change in high frequency precipitation variability over North America. *Scientific reports*, 7(1), 1-12.
- Rui, L., & Yunfei, F. (2005). Tropical precipitation estimated by GPCP and TRMM PR observations. *Advances in atmospheric Sciences*, 22, 852-864
- Shiogama, H., Watanabe, M., Kim, H., & Hirota, N. (2022). Emergent constraints on future precipitation changes. *Nature*, 602(7898), 612-616.
- Zhang, W., Furtado, K., Wu, P., Zhou, T., Chadwick, R., Marzin, C., ... & Sexton, D. (2021). Increasing precipitation variability on daily-to-multiyear time scales in a warmer world. *Science advances*, 7(31), eabf8021.

Reviewer Reports on the First Revision:

Referees' comments:

Referee #2 (Remarks to the Author):

The authors have done a huge amount of work in responding to the reviewer comments. I'm happy with the revision, particularly the inclusion of multiple precipitation datasets and the discussion of the "warm get variable" paradigm, especially as relates to the local minimum of the SST anomalies associated with the IPO. Aside from a few minor comments below, my only major (and optional) comment is that the abstract could benefit from slightly clearer language. The important points are all there- but the impact of the paper outside the relatively narrow climate/D&A community could be increased by noting that while we can't (yet?) see the impact of climate change in long-term rainfall patterns, this paper clearly shows that climate change is already affecting the weather, defined as <10 day precipitation variability. This is a big deal, and should be stated very clearly.

As noted before, the distinction between this methods and other methods (stationary fingerprint-based D&A, ridge regression) is both helpful for clarity and makes a compelling case for the novelty of this method. I think this is one of the strongest aspects of the paper.

New (positive) changes

Thank you for including IMERG and GPCP- the results are now more robust. I particularly liked the new Fig 2b. I think it's very helpful to show the divergence in sign between observational datasets when linear trends are used to train the model along with the agreement in 10d precipitation.

Minor comments

L51: interference -> influence

L72: the detection -> detection, the natural internal variability -> internal variability

L73 the detection -> detection

L79 the non-anthropogenic weather noise -> non-anthropogenic weather noise

L110: radars -> radar

L111: not sure you need "Note" here

L 134 Is it emerged days or emergence days? Doesn't matter but please be consistent with Figure 1.

L178: The linear trends don't look negligible, they look opposite. Specify it's ≤ 1 yr timescales?

L180: degrees of increase is confusing. Suggest something clearer like "global warming trend"

L166 It's not negative in IMERG. Optional suggestion: I think it would be clearer to explicitly state this in a way a non-expert might understand. Suggested wording: in 3/4 datasets, when the model attempts to infer the global mean temperature change from long-term changes to precipitation, it finds not global warming, but cooling.

L233 was -> were (increases is plural)

241: the modest -> moderate

256: not sure "contaminated" is appropriate here- obscured?

L257 "is slightly increased" -> did increase slightly

L263: is or was? (Choose a tense and stick to it?)

L284: I really appreciate this rain gauge section, it's very helpful

L285: You've used "fingerprint" before to mean stationary fingerprints. For clarity, maybe rewrite this as "high-frequency precipitation variability in the tropics and mid-latitudes is associated with global mean temperature change" or similar

L294 "Note that..." I think I get what you're trying to say here, but I find it unclear and a bit confusing. Should this be combined with the prior paragraph? Why just the subtropical Atlantic, since the same patterns are also visible in the southeastern Pacific? A rewrite of this paragraph

could increase clarity

L514: experimentations... -> experiments to minimize the loss values for the validation dataset

Referee #3 (Remarks to the Author):

Summary:

Based on a large climate model ensemble, the authors have developed a DL model to predict annual global mean temperatures from daily precipitation maps. They applied this model to daily precipitation maps from a reanalysis and a satellite dataset and discovered an increasing trend, with roughly half of the days emerging from natural climate variability and thus indicating the detection of forced climate change in daily precipitation at a global scale. I find these results interesting, but I have some reservations about the observational datasets (reanalysis data are not observations!), the algorithms used for training and explaining the network, and the statistical treatment of uncertainty.

Originality and significance

I found the paper partly original in the use of standard DL methods to a relevant problem of detecting the signals of climate change in noisy daily precipitation anomalies. I also very much appreciated the attempt to introduce alternative approaches based on ML to the more traditional "fingerprinting" methods in the Detection and Attribution (D&A) literature. Their approach uses neural nets for prediction, and then explainable AI for understanding the rules encapsulated in the model is neat and traditionally effective. Nevertheless, other recent ML approaches to D&A should be cited, compared, and considered in the discussion.

Data & methodology

However, I have some methodological concerns I would like to raise, mainly related to the (ab)use of terminology and the stretching of conclusions, that I hope authors can address:

1- Using CNN+XAI to tackle a D&A question is sound and very welcome. However, the approach has many opportunities that the authors should have explored. First, one predicts a scalar global temperature value from a precip spatial snapshot, which is fine. However, one could infer the full global temperature map with the current ML instead. That would give their findings much more value and significance, especially in the context and discussion about regional-vs-global patterns and processes. Second, why only precipitation and not ingesting many other potentially explanatory forcings? The discussion would indeed be more complex, but at the same time, D&A would be more trustworthy (by mitigating the doubt about the impact of confounders in your "attribution")!

2- I am also concerned about some statements on interpretability-vs-explainability; authors rely on the traditional occlusion method to get insight into the DL model performance (looking directly at the learned filters says little indeed). There are several points here that I would like the authors to acknowledge, discuss and eventually do something about it.

The first issue, at a more conceptual level, is about the fact that it does matter what the XAI method used; the extracted patterns will not be actual causal fingerprints: the identified patterns are not real physical patterns (and thus cannot be discussed as causal attribution patterns as such), but they are patterns in the DL model world. Physical interpretation should be taken with a grain of salt. Downtoning some statements on "attribution" and discussing the approach's limitations would be good.

The second issue is more technical and related to the (big) limitations of occlusion (especially in your study setting). Although occlusion is a useful method for interpreting ML models, there are several main and critical problems associated with this method, such as interpretability bias (i.e. there may be other important features or interactions that the model is considering that are not captured); computational burden (training and evaluation should be done multiple times for each occlusion, which I suspect authors did not do it), difficulty in selecting occlusion parameters (e.g. the window and patch sizes here are critical and I do not understand why a CNN used with RF of 3x3 is then evaluated with 7x7 occlusion patches; this possibly compromises many results), and importantly occlusion does not work in nonstationary processes because of the distribution shift (i.e. your case!). Overall, occlusion is a powerful tool, but it should be used cautiously and with other interpretability methods to gain a more comprehensive understanding of ML models. I suggest trying other methods like 'neuron integrated gradients' or SHAP values, and of course, even simpler: visualize the weights of the regularized linear regression you did!

3- Novelty on ML-based attribution? There are several ML methods to tackle this problem. The authors should at least mention them and discuss them and even better compare their results, e.g. anchor regression in Sippel et al in <https://www.science.org/doi/10.1126/sciadv.abh4429>, physics-aware ML in Cortes et al <https://iopscience.iop.org/article/10.1088/1748-9326/ac6762>, or SNR identification by Barnes et al <https://agupubs.onlinelibrary.wiley.com/doi/full/10.1029/2019GL084944>.

4- Concerns about the data. The authors claim to use observational data, but I agree with my reviewer colleagues that ERA cannot be taken as such, and such statements should be rephrased. Nevertheless, I acknowledge the effort and liked the discussion on using gauge data, which is quite convincing. Perhaps the main problem comes from the short time series used for the "attribution" exercise, but I am afraid very little can be done (and even less if the first decades of ERA cannot be even considered!).

Stats

5- On the stats. Several issues to be treated carefully: i) I'm concerned about using some ad hoc definitions of counting statistics (e.g. the number of fractional EM days). I encourage the authors to discuss the limitations of such definitions better and strongly encourage using the Poisson distribution and EVT to characterize such exceeding fractions. ii) Add error bars to all figure bars (or even better, give the distributions!), as I suspect many error bars overlap and thus the attribution and statements could not hold; iii) unclear what the statistically significant tests are used in Fig 2.

Conclusions

This is a very good paper with an interesting and alternative approach to D&A based on AI methods (for both prediction and interpretation). The results are very interesting and are worth publishing. However, more work & comparisons are needed to probe the results, as I fear the methodological and data limitations compromise some of the conclusions.

Author Rebuttals to First Revision:

Referee #2 (Remarks to the Author):

Summary:

The authors have done a huge amount of work in responding to the reviewer comments. I'm happy with the revision, particularly the inclusion of multiple precipitation datasets and the discussion of the "warm get variable" paradigm, especially as relates to the local minimum of the SST anomalies associated with the IPO. Aside from a few minor comments below, my only major (and optional) comment is that the abstract could benefit from slightly clearer language. The important points are all there- but the impact of the paper outside the relatively narrow climate/D&A community could be increased by noting that while we can't (yet?) see the impact of climate change in long-term rainfall patterns, this paper clearly shows that climate change is already affecting the weather, defined as <10 day precipitation variability. This is a big deal, and should be stated very clearly.

: Thank you for the encouraging comments and the further suggestion. We fully agree that the abstract would benefit from a clear emphasis on our main message. We modified the last sentence of the abstract with much general terms which would be familiar to the readers outside the climate/D&A community as follows.

Line 48-51 : "Our results highlight that, although the long-term shifts in annual mean precipitation remain indiscernible from the natural background variability, the impact of global warming on daily hydrological fluctuations is already emerged."

As noted before, the distinction between this methods and other methods (stationary fingerprint-based D&A, ridge regression) is both helpful for clarity and makes a compelling case for the novelty of this method. I think this is one of the strongest aspects of the paper.

New (positive) changes : Thank you for including IMERG and GPCP- the results are now more robust. I particularly liked the new Fig 2b. I think it's very helpful to show the divergence in sign between observational datasets when linear trends are used to train the model along with the agreement in 10d precipitation.

: Thank you for the encouraging comments.

Minor comments:

L51: interference -> influence

: Revised as "impact of global warming".

L72: the detection -> detection, the natural internal variability -> internal variability

: Corrected.

L73 the detection -> detection

: Corrected.

L79 the non-anthropogenic weather noise -> non-anthropogenic weather noise

: Corrected.

L110: radars -> radar

: Corrected.

L111: not sure you need "Note" here

: As a reviewer suggested, "Note" is deleted.

L 134 Is it emerged days or emergence days? Doesn't matter but please be consistent with Figure 1.

: Sorry for the confusion. Throughout the manuscript, it is consistently changed to emergence days.

L178: The linear trends don't look negligible, they look opposite. Specify it's <= 1yr timescales?

: Thank you for pointing this out. We clarify this statement is for the variability shorter than 1 year period.

L180: degrees of increase is confusing. Suggest something clearer like "global warming trend"

: Thank you for the suggestion. It is corrected to "predicted global warming trends".

L166 It's not negative in IMERG. Optional suggestion: I think it would be clearer to explicitly state this in a way a non-expert might understand. Suggested wording: in 3/4 datasets, when the model attempts to infer the global mean temperature change from long-term changes to precipitation, it finds not global warming, but cooling.

: We agree that this sentence needs to be stated clearly. The corresponding sentence is modified as follows.

Line 166-167 : "The negative contribution of the linear trend component found in three of total four precipitation datasets may ..."

L233 was -> were (increases is plural)

: Corrected.

241: the modest -> moderate

: Corrected.

L256: not sure “contaminated” is appropriate here- obscured?

: Corrected as a reviewer suggested.

L257 “is slightly increased “-> did increase slightly

: Corrected.

L263: is or was? (Choose a tense and stick to it?)

: It is stick to the present tense throughout the paragraph.

L284: I really appreciate this rain gauge section, it’s very helpful

: Thank you for acknowledging our efforts.

L285: You’ve used “fingerprint” before to mean stationary fingerprints. For clarity, maybe rewrite this as “high-frequency precipitation variability in the tropics and mid-latitudes is associated with global mean temperature change” or similar

: Thank you for giving us an opportunity to clarify this point. The corresponding sentence is modified as follows.

Line 289-291: “Global warming has resulted in increased HF precipitation variability over the tropical and mid-latitude regions, while the subtropical Atlantic and southeastern Pacific show a predominance of climatological drying instead.”

L294 “Note that...” I think I get what you’re trying to say here, but I find it unclear and a bit confusing. Should this be combined with the prior paragraph? Why just the subtropical Atlantic, since the same patterns are also visible in the southeastern Pacific? A rewrite of this paragraph could increase clarity

: Thank you for pointing this out. The corresponding paragraph is shortened and rewritten to clearly demonstrate the cause of negative occlusion sensitivity over the subtropical Atlantic and southeastern Pacific as follows.

Line 297-301 : “Note that, the stronger positive AGMT response in bottom percentiles than the negative response in top percentiles results in a net negative AGMT response to the decreased HF variability over the subtropical Atlantic and southeastern Pacific (Fig. 4b). Consequently, this contributes to the negative occlusion sensitivity trend with HF precipitation input (Fig. 2c).”.

L514: experimentations... -> experiments to minimize the loss values for the validation dataset

: Corrected as a reviewer suggested.

Referee #3 (Remarks to the Author):

Summary:

Based on a large climate model ensemble, the authors have developed a DL model to predict annual global mean temperatures from daily precipitation maps. They applied this model to daily precipitation maps from a reanalysis and a satellite dataset and discovered an increasing trend, with roughly half of the days emerging from natural climate variability and thus indicating the detection of forced climate change in daily precipitation at a global scale. I find these results interesting, but I have some reservations about the observational datasets (reanalysis data are not observations!), the algorithms used for training and explaining the network, and the statistical treatment of uncertainty.

: Thank you for your insightful comments concerning our manuscript. All the comments from the reviewer are valuable and helpful for improving our manuscript. We have completed a revised version of our manuscript considering your reviews, with the clarification of the issues raised by the observational dataset we utilized. The revision responds elaborately to each of your comments through the requested additional analysis and experiments for the explaining the network, which we wish to address your concerns to your satisfaction. To help your better navigate and investigate the updates, we provide a detailed point-by-point response letter hereafter.

Originality and significance :

I found the paper partly original in the use of standard DL methods to a relevant problem of detecting the signals of climate change in noisy daily precipitation anomalies. I also very much appreciated the attempt to introduce alternative approaches based on ML to the more traditional "fingerprinting" methods in the Detection and Attribution (D&A) literature. Their approach uses neural nets for prediction, and then explainable AI for understanding the rules encapsulated in the model is neat and traditionally effective. Nevertheless, other recent ML approaches to D&A should be cited, compared, and considered in the discussion.

: We agree with the reviewer comment that the recent ML approaches to D&A should be properly recognized in the manuscript. Even though some of recent papers which utilized feed-forward neural networks (FFNNs) (e.g., Barnes et al., 2020, *ref.* 26 of the original manuscript) is already cited in the original manuscript, its difference is not properly stated. The convolutional neural networks (CNNs) used in this study would be beneficial compared to the FFNNs in two aspects. First, the convolutional process in CNNs is able to capture local features in the global domain, and is therefore suitable for detecting regional pattern changes associated with global warming. On the other hand, FFNNs seek the relationship between each input grid point and the output. Second, CNNs' partial translation invariant feature, which recognize the object in the image even though the shape and locations of the corresponding object is changed

(<https://noobest.medium.com/translational-invariance-and-translational-equivariance-in-cnn-25fb0f4f61e3>), allows CNNs to capture the non-stationary global warming signal in daily precipitation. It is not possible for the FFNNs which connects the input and output with the fixed weights. Therefore, as highlighted in the main text, none of recent ML methods have been successful to extract the global warming signal in daily precipitation.

The advantages of the CNNs compared to the recent ML approaches is clearly demonstrated by updating the paragraph in Line 100-107 of the revised manuscript.

Line 100-107 : “The convolutional process embedded in the CNN is able to capture local features in the global domain, making it suitable for detecting regional pattern changes associated with global warming. In addition, with its translation invariant feature, the DD model can extract common change patterns due to the global warming in both the model simulations and the observations despite their systematic differences. This feature contrasts with the existing D&A techniques, including the updated linear regression-based approaches (Sippel et al., 2021; Cortes-Andres et al., 2022) and the feed-forward neural networks (Barns et al., 2020), which detect climate change signals based on a global stationary fingerprint pattern.”

Data & methodology :

However, I have some methodological concerns I would like to raise, mainly related to the (ab)use of terminology and the stretching of conclusions, that I hope authors can address:

1- Using CNN+XAI to tackle a D&A question is sound and very welcome. However, the approach has many opportunities that the authors should have explored. First, one predicts a scalar global temperature value from a precip spatial snapshot, which is fine. However, one could infer the full global temperature map with the current ML instead. That would give their findings much more value and significance, especially in the context and discussion about regional-vs-global patterns and processes. Second, why only precipitation and not ingesting many other potentially explanatory forcings? The discussion would indeed be more complex, but at the same time, D&A would be more trustworthy (by mitigating the doubt about the impact of confounders in your "attribution")!

: Thank you for providing the opportunity to clarify this issue. It would be interesting to predict the full global temperature map, however, in this case, the task becomes a conventional prediction problem. It is not optimal for the D&A approach to predict the global temperature map, as it contains not only the global warming signal, but also other temporal-scales of climate variabilities (Mann et al., 2000; Power et al., 2021). That is, a globally-averaged temperature is often defined as a predictand for the D&A approaches as the spatially-inhomogeneous internal decadal variations are cancelled out by taking the spatial averages. Therefore, it has been used as the most direct proxy for the forced responses to the greenhouse gas increases (Sippel et al., 2020, 2021; Cortes-Andres et al., 2022, and many others). In contrast, the

variabilities associated with the climate oscillations will remain in the global temperature map, rather decreasing global warming signal detectability (e.g., Stott et al. 2010).

To demonstrate this point, we have performed the Empirical Orthogonal Functions (EOFs) of the annual-mean 2m-air temperature map (AMT). 1st EOF exhibited a global warming signal, which takes 40.03 % of total variance, and, 2nd, and 3rd EOFs showed the spatial patterns associated with the Interdecadal Pacific Oscillations (IPO), and Atlantic Multidecadal Oscillations (AMO), which takes 6.91 %, and 6.77 % of total variance, respectively (Figure B-1) (Chen and Tung, 2018; Yang et al., 2020).

For the variations of the globally-averaged annual-mean temperature (AGMT) (Figure B-2), the variability associated with IPO or AMO is systematically reduced; the AGMT variance regressed onto the 2nd, and 3rd EOF PC time-series of the AMT is only 1.18 %, and 0.4 % of total variance, which is significantly reduced from the explained variance of the AMT (i.e., 6.91 %, and 6.77 %), respectively. On the other hand, the variability associated with the global warming tremendously increased in AGMT; the AGMT variance regressed onto the 1st EOF PC time-series of the AMT is 98.01 %. This indicates that the deep learning model to predict full temperature map considers not only the impact of global warming, but also the impact of climate oscillations with other temporal time-scales. As a result, the choice of full temperature map as a predictand will be less effective to isolate the forced climate change signal from the internal climate variability than the use of AGMT.

Figure B-1. (a) 1st, (b) 2nd, and (c) 3rd Empirical Orthogonal Functions (EOFs) of the annual-mean temperature (AMT). The principal component time-series of the (d) 1st, (e) 2nd, and (f) 3rd EOFs are also shown.

Figure B-2. Time-series of the observed annual global mean 2m-air temperature (AGMT).

We understood the reviewer’s second comment as a suggestion to consider additional variables (e.g., humidity) to increase detectability of climate change signals. There are a few reasons why we use daily precipitation only. First, compared to the physical variables, such as temperature, humidity, sea-ice, or snow cover, daily precipitation has not been analyzed by any previous D&A studies even though it is a key variable for the global hydrological cycle which determines freshwater availability across the world and occurrence of hydrological weather extremes (i.e., floods).

Secondly, daily precipitation is one of the most challenging variables for climate change signal detection due to its large weather-related internal variability. As demonstrated in Extended Data Table 1 (also shown in Table B-1 in this rebuttal letter), the ratio of the long-term trend (i.e., signal) to total variability in the daily precipitation (i.e., signal + noise) is only 10 % of those in daily 2m temperature or specific humidity. This low signal-to-noise ratio in precipitation prevents the global warming signal from emerging beyond the internal variability range, particularly in the linear framework. As a result, the emergence of the climate change signal has not been detected in daily precipitation compared to temperature, or humidity (Extended Data Fig. 1 and Figure B-3 in this rebuttal letter). Combining other variables with daily precipitation may increase signal detectability but this is not the target of our study, i.e. identifying human influence on daily precipitation in isolation.

Thirdly, we note that the robust impact of the global warming signal on the daily precipitation occurs in its variance change, whereas temperature or humidity signal is known to be manifested in its long-term mean change. This discrepancy in time scales implies that

combining with other variables would possibly hinder the emergence of the anthropogenic signal embedded in the daily precipitation. For example, for daily precipitation, we demonstrated that the V-shape response function shows up clearly to capture the daily variance increase as global warming signal, and this kind of non-linear response pattern can be obscured by including specific humidity as its response function would exhibit a linear shape to detect a long-term trend. That is, differences in the global warming imprints between variables would prevent a proper understanding of associated physical mechanisms.

	Precipitation	2m temperature	2m Specific humidity
Ratio (Trend/Variability)	0.031	0.276	0.260

Table B-1. The ratio of the globally-averaged ratio of linear trend (unit: change per a decade) to the standard deviation of the daily 2m temperature, 2m specific humidity, and precipitation anomaly during 1950–2020 in CESM2 LE.

Figure B-3. (a) Time-series of the simulated annual global mean 2m-air temperature (AGMT) from 1850 to 2100 in the CESM2 LE (black line), and the annual average of the estimated daily AGMT by prescribing 2m temperature (T2m, green line), 2m specific humidity (SH2m, orange line), and precipitation (Pr, blue line) in the ridge regression model²³. Each dot denotes the estimated AGMT using daily input. The green, orange, and blue bar on the right side denotes the one standard deviation of estimated daily AGMT using the T2m, SH2m, and Pr during the historical period (i.e., 1850–1950), respectively. The black error bar denotes the 2.5th–97.5th percentile of the daily estimated AGMT in 1850–1950. (b) Time-series of the ratio of the annually-averaged AGMT to the AGMT of upper limit of test statistics (i.e., 97.5th percentile of the daily estimated AGMT in 1850–1950). The first year that the ratio exceeds 1 for each case is indicated as a number.

2- I am also concerned about some statements on interpretability-vs-explainability; authors rely on the traditional occlusion method to get insight into the DL model performance (looking directly at the learned filters says little indeed). There are several points here that I would like the authors to acknowledge, discuss and eventually do something about it.

The first issue, at a more conceptual level, is about the fact that it does matter what the XAI method used; the extracted patterns will not be actual causal fingerprints: the identified patterns are not real physical patterns (and thus cannot be discussed as causal attribution patterns as such), but they are patterns in the DL model world. Physical interpretation should be taken with a grain of salt. Downtoning some statements on "attribution" and discussing the approach's limitations would be good.

: Thank you for the comments. Differing from the concerns of the reviewer, the XAI method applied in this study can derive a physical pattern; the XAI visualizes the sensitivity of the AGMT (unit of °C) to the prescribed daily precipitation map (unit of mm/day), both of which are physical units. Therefore, the spatial pattern derived by the occlusion sensitivity also has a physical unit (unit of °C), which can quantify how much degree of warming is led by the input values in each grid point. This implies that hot spot regions highlighted in this study directly infer the areas which mostly contributes to the successful climate change detection. Therefore, XAI method demonstrates a relationship which is physically understandable, beyond showing patterns that apply only to the DL model world.

However, the occlusion sensitivity only visualizes a relationship between the input and the output variables, and its physical interpretation should be taken by analysts. In other words, occlusion sensitivity focuses on providing explanations for existing black box models, therefore, it is an explainable method (Rudin, 2019). To avoid the confusion, the occlusion sensitivity is described as an ‘explainable’ method rather than an ‘interpretable’ method throughout the manuscript.

Regarding the “attribution”, we now clarify that we do not use the XAI method to attribute the detected signal to a particular cause. That is, our XAI method only identifies a relatively important grids which contribute to the global warming signal emergence (i.e., detection process), and we do not further evaluate the emerged signal for attribution (i.e., quantifying the contributions of external forcings such as greenhouse gases, anthropogenic aerosols, or solar/volcanic activities). To tone down and clarify the role of XAI on the attribution, we have revised the related sentence as follows.

Line 197-199 : “the occlusion sensitivity was calculated for all days, and its linear trend during 1980–2020 was obtained to measure its contribution to the global warming signal detection.”

The second issue is more technical and related to the (big) limitations of occlusion (especially in your study setting). Although occlusion is a useful method for interpreting ML models, there are several main and critical problems associated with this method, such as interpretability bias (i.e. there may be other important features or interactions that the model is considering

that are not captured); computational burden (training and evaluation should be done multiple times for each occlusion, which I suspect authors did not do it), difficulty in selecting occlusion parameters (e.g. the window and patch sizes here are critical and I do not understand why a CNN used with RF of 3x3 is then evaluated with 7x7 occlusion patches; this possibly compromises many results), and importantly occlusion does not work in nonstationary processes because of the distribution shift (i.e. your case!). Overall, occlusion is a powerful tool, but it should be used cautiously and with other interpretability methods to gain a more comprehensive understanding of ML models. I suggest trying other methods like 'neuron integrated gradients' or SHAP values, and of course, even simpler: visualize the weights of the regularized linear regression you did!

: One does not need to train the model for multiple times to obtain an occlusion sensitivity (OS); the identical model trained with full dataset is utilized, and only the testing is done multiple times (Zeiler and Fergus 2014; Tang et al., 2019). Therefore, even though an OS for each day and each occlusion patch has to be calculated, the computational costs for OS calculation is affordable. In addition, the patch size for the OS is not necessarily matched to the convolutional filter size (Cheng et al., 2018; Islam et al., 2017; Tang et al., 2019). In general, studies that performed OS analysis set the size of the occlusion patch to be larger than the size of the convolutional filter. For example, Tang et al. (2019) utilized a 3×3 convolutional filter and set the size of the occlusion patch to 16×16 to explain their CNN-based model. Similarly, we set the slightly bigger size of the occlusion patch (i.e., 7×7) to examine the impact of the regional precipitation pattern.

To check whether the regional contribution to the global warming signal detection is sensitive to the choice of the XAI method, Figure B-4 shows the linear trend of the Shapley Additive exPlanations (SHAP), and integrated gradients (IG)' values (Lundberg and Lee 2007; Sundararajan et al., 2017). In addition, OS results with different size of the patch (i.e., 5 × 5 grid points) are also shown. Results using SHAP, IG, and OS with different patch size are largely similar to those using the original OS in the main text; the off-equatorial ITCZ, mid-latitude storm track regions, northern South America, and the equatorial Atlantic importantly contribute to the increased AGMT in time. The only difference is the smoothness in spatial distributions, which occurs because the SHAP and IG calculate a sensitivity of a single grid point while the OS exhibits a sensitivity of multiple grid points.

In short, sensitivity tests using the OS with a different patch size and other explainable methods such as SHAP and IG support our main findings. We have briefly added this point in the revised text (Line 204-208) with including Figure B-4 as an Extended Data Figure 6.

Line 204-208 : “These hot spot regions appear distinctly when using a different patch size for occlusion sensitivity (Extended Data Fig. 6a), or utilizing other explainable methods such as Shapley Additive exPlanations (SHAP) or Integrated Gradients method (Extended Data Fig. 6b and 6c).”

Figure B-4. Linear trend of the AGMT sensitivity (unit: $^{\circ}\text{C decade}^{-1}$) for 10 day high-pass filtered ERA5 and MSWEP precipitation anomalies during 1980-2020 measured by (a) occlusion sensitivity with patch size of 5×5 grid points and (b) Shapley Additive exPlanations (SHAP) and (c) Input \times Integrated Gradients methods. The SHAP value is estimated using a gradient explainer, where the explainer is approximated by 1,000 randomly selected samples from the training dataset. The input \times integrated gradients is estimated by multiplying the original input by each of the alpha values (i.e., 0, 0.2, 0.4, 0.6, 0.8, 1.0), computing the gradients of the CNN model, integrating the obtained gradients (the integral is approximated by a Riemann sum), and multiplying by the original input.

3- Novelty on ML-based attribution? There are several ML methods to tackle this problem. The authors should at least mention them and discuss them and even better compare their results, e.g. anchor regression in Sippel et al in <https://www.science.org/doi/10.1126/sciadv.abh4429>, physics-aware ML in Cortes et al <https://iopscience.iop.org/article/10.1088/1748-9326/ac6762>, or SNR identification by Barnes et al <https://agupubs.onlinelibrary.wiley.com/doi/full/10.1029/2019GL084944>.

: Thank you for letting us know the references. Those references are valuable to improve the quality of our manuscript. We have now cited them (Ref. 22, 25, and 26) in the revised manuscript as appropriate (Lines 81, 106, and 303).

4- Concerns about the data. The authors claim to use observational data, but I agree with my reviewer colleagues that ERA cannot be taken as such, and such statements should be rephrased. Nevertheless, I acknowledge the effort and liked the discussion on using gauge data, which is quite convincing. Perhaps the main problem comes from the short time series used for the "attribution" exercise, but I am afraid very little can be done (and even less if the first decades of ERA cannot be even considered!).

: We fully understand the reviewer's concern. During the previous review, same concerns were raised regarding the observational dataset. We believe that we have addressed these concerns properly by utilizing (1) multiple merged observations, which are MSWEP, IMERG, and GPCP, and (2) rain-gauge data.

In all four different precipitation datasets (MSWEP version 2.8, GPCP version 3.2, IMERG version 6, and ERA5 reanalysis), the predicted AGMT is above the 95 % confidence range of the natural variability (main Fig. 1a) and the number of the emergence day significantly increases in time (main Fig. 1b and 1c). In addition, our main findings through a series of sensitivity experiments hold consistently in all four precipitation datasets; the HF precipitation anomalies with periods shorter than 10 days were mostly responsible for the climate change detection (main Fig. 2a and 2b).

The results from the rain gauge data are consistent with those from the merged precipitation datasets. A robust increase in the magnitude of the HF precipitation variability during recent decades in the eastern U.S., which are in the hot spots; the difference map between [2001–2020] and [1960–1979] shows the robust variability increase after 2000s particularly over the northeast U.S (main Fig. 4e). On the contrary, over the western U.S., which is outside the hot spots, the HF precipitation variability does not show any trend in its magnitude. In addition, the HF variability change ratio is greater than the mean precipitation changes ratio only over eastern U.S. (Extended Data Fig. 9).

The short-observed time series is not an issue to measure the natural internal variability range since we estimate it from the long control simulations of our training dataset (i.e., CESM2 LE). The reviewer might refer a problem with a short-observed time series in terms of making the

detection much more challenging as the differences between the early and recent period (i.e., global warming signal) become smaller than the long-term time series. Yes, in general, longer-term observations would help to enhance signal detectability as noise becomes smaller, increasing signal-to-noise ratio (Eyring et al. 2021). This can be seen in the dotted black line of Figure 1c; an internally-generated ranges of EM day trends are smaller in the 40-year period than that in the 20-year period. By the lowered internal variability range, the detection becomes easier with a longer observed time period. Nevertheless, the observed trends in EM day fraction remain much larger than the internal variability level even when using short period datasets, which support robustness of our detection results as well as the superiority of our deep learning model.

Stats : 5- On the stats. Several issues to be treated carefully: i) I'm concerned about using some ad hoc definitions of counting statistics (e.g. the number of fractional EM days). I encourage the authors to discuss the limitations of such definitions better and strongly encourage using the Poisson distribution and EVT to characterize such exceeding fractions. ii) Add error bars to all figure bars (or even better, give the distributions!), as I suspect many error bars overlap and thus the attribution and statements could not hold; iii) unclear what the statistically significant tests are used in Fig 2.

: i) The definition of the counting statistics (i.e., number of EM days divided by total number of days per year) have been introduced by previous studies (e.g. Sippel et al., 2020). This fraction of days corresponds to annual probability of daily precipitation fields that have a significant impact of climate change. Many previous studies assessed signal emergence using similar measures of probability or counting statistics (e.g. Lehner and Stocker 2015, King et al. 2016, Seneviratne et al. 2021). Regarding the statistical distribution of emergence counts, we prefer to count EM days empirically since it is difficult to find a statistical distribution that provides a good fit with our counting values varying with time differently across cases. More specifically, even if we utilize Poisson distribution, this counting statistic is still required as Poisson distribution shows the probability of the number of event occurrences.

In addition, the main conclusion using a simple counting statistic (i.e., number of EM days) and its significance assessed based on the bootstrap method still hold when testing the significance using a Poisson distribution. For example, according to the Poisson distribution for a mean occurrence number (i.e., EM days) obtained from the long-term simulation of the CESM2 LE, the probability with fractional EM days > 5, 10, 15, and 20 % is 0.1088, 0.0002, 0.00000011, and 0, respectively. This indicates that 50 % of the observed fractional EM days after 2010s is clearly statistically significant.

Concerning extreme value theory (EVT), distribution based on EVT can be used to describe peak-over-threshold 'extreme' values (Coles et al. 2001), such as the Generalized Pareto Distribution (GPD). However, the GPD distribution describes the intensity of extreme events,

not counting statistics. Thus, fitting an EVT-based distribution to EM days seems inappropriate for the analysis of signal emergence.

ii), and iii) Thanks for the good point. We have now modified confidence ranges in all figures to consistently denote the natural variability obtained from a long-term simulation CESM2 LE as done in Figure 1. We first sampled 20-year segments from CESM2 ensemble simulations during 1850–1950 with a 10-year interval in the initial year of the segments. With 9 values per each ensemble member, a total of 720 (9 x 80) values were obtained. Similarly, total 960 samples of 41-year segments are obtained with a 5-year interval. An identical metric in each figure is calculated for all 20-year or 41-year segments of CESM2 LE; linear trend of the AGMT for each segment is calculated for Figure 2a and 2b, and, the linear trend of the precipitation variability is calculated for Figure 4a and 4b. The upper and lower 2.5 % value is defined as 95 % confidence range of a two-tailed significance test. We have revised our description of natural variability estimation in Methods accordingly (Line 478-496).

The confidence range based on the natural variability shows that the linear trend of the estimated AGMT using the HF precipitation variations is significant (Figure B-5; main Fig. 2a,b). Similarly, it is confirmed that the precipitation variability is significantly increased in eastern Pacific ITCZ and mid-latitude storm track regions (Figure B-6; main Fig. 4a, c).

Figure B-5. Linear trends of the estimated AGMT from DD model for (a) 1980–2020 (unit: °C decade⁻¹) using ERA5 reanalysis (red) or MSWEP (blue), and (b) 2001–2020 using IMERG (purple) or GPCP (orange). Each case shows results from using unfiltered precipitation anomalies (denoted as total), linear trends of the precipitation anomalies (trend), 10-day high-pass filtered (10d HP), 10–30 day band-pass filtered (10–30d BP), 30–90 day band-pass filtered (30–90d BP), 90 day–1 year band-pass filtered (90d–1y BP), and 1 year low-pass filtered (1y LP) precipitation. The dashed black horizontal lines denote an upper bound of a 95% confidence range of internal variability of the estimated AGMT linear trend, obtained from historical CESM2 LE simulations during 1850–1950.

Figure B-6. (a) Linear trend of the standard deviation (STD) of ERA5 (red) and MSWEP (blue) normalized HF precipitation anomalies during 1980–2020 (closed bars) or 2001–2020 (hatched bars) over the eastern Pacific ITCZ (black boxed area within 20°S–20°N) and mid-latitude storm track regions (black boxed area poleward of 30°S and 30°N). (b) Ratio of the linear trend of HF precipitation variability during 1980–2020 to the linear trend of precipitation climatology (each divided by 1980–1984 mean) in the eastern Pacific ITCZ, mid-latitude storm tracks, and subtropics using ERA5 (red) and MSWEP (blue). The error bars in panel a and b denote a 95% confidence interval of internal variability, obtained from historical CESM2 LE simulations during 1850–1950.

Conclusions : This is a very good paper with an interesting and alternative approach to D&A based on AI methods (for both prediction and interpretation). The results are very interesting and are worth publishing. However, more work & comparisons are needed to probe the results, as I fear the methodological and data limitations compromise some of the conclusions.

: We hope our responses regarding methodological and observational issues provided above address the reviewer’s concerns. Results from additional analysis using other explainable methods (SHAP and IG) exhibited a similar spatial distribution with results based on the occlusion sensitivity. We showed that, in spite of a short period of datasets, multiple observational datasets consistently support our main arguments that changes in HF precipitation variability contain rigorous climate change signal, far exceeding the range explained by natural internal variability. We have mentioned these points in the revised text (Line 112-113, Line 182-184, Line 309-310).

References

Barnes, E. A., Toms, B., Hurrell, J. W., Ebert-Uphoff, I., Anderson, C., & Anderson, D. (2020). Indicator patterns of forced change learned by an artificial neural network. *Journal of Advances in Modeling Earth Systems*, 12(9), e2020MS002195.

- Chen, X., & Tung, K. K. (2018). Global-mean surface temperature variability: Space–time perspective from rotated EOFs. *Climate Dynamics*, *51*(5-6), 1719-1732.
- Cheng, P. M., Tejura, T. K., Tran, K. N., & Whang, G. (2018). Detection of high-grade small bowel obstruction on conventional radiography with convolutional neural networks. *Abdominal Radiology*, *43*, 1120-1127.
- Coles, S., Bawa, J., Trenner, L., & Dorazio, P. (2001). *An introduction to statistical modeling of extreme values* (Vol. 208, p. 208). London: Springer.
- Cortés-Andrés, J., Camps-Valls, G., Sippel, S., Székely, E., Sejdinovic, D., Diaz, E., ... & Reichstein, M. (2022). Physics-aware nonparametric regression models for Earth data analysis. *Environmental Research Letters*, *17*(5), 054034.
- Eyring, V., Gillett, N. P., Achutarao, K., Barimalala, R., Barreiro Parrillo, M., Bellouin, N., ... & Sun, Y. (2021). Human Influence on the Climate System. In *Climate Change 2021: The Physical Science Basis. Contribution of Working Group I to the Sixth Assessment Report of the Intergovernmental Panel on Climate Change. IPCC Sixth Assessment Report*.
- IPCC, (2022) *Climate Change 2022: Impacts, Adaptation and Vulnerability. Contribution of Working Group II to the Sixth Assessment Report of the Intergovernmental Panel on Climate Change* [H.-O. Pörtner, D.C. Roberts, M. Tignor, E.S. Poloczanska, K. Mintenbeck, A. Alegría, M. Craig, S. Langsdorf, S. Löschke, V. Möller, A. Okem, B. Rama (eds.)]. Cambridge University Press. Cambridge University Press, Cambridge, UK and New York, NY, USA, 3056 pp., doi:10.1017/9781009325844.
- Islam, M. T., Aowal, M. A., Minhaz, A. T., & Ashraf, K. (2017). Abnormality detection and localization in chest x-rays using deep convolutional neural networks. arXiv preprint arXiv:1705.09850.
- King, A. D., Black, M. T., Min, S. K., Fischer, E. M., Mitchell, D. M., Harrington, L. J., & Perkins-Kirkpatrick, S. E. (2016). Emergence of heat extremes attributable to anthropogenic influences. *Geophysical Research Letters*, *43*(7), 3438-3443.
- Lehner, F., & Stocker, T. F. (2015). From local perception to global perspective. *Nature Climate Change*, *5*(8), 731-734.
- Lundberg, S. M., & Lee, S. I. (2017). A unified approach to interpreting model predictions. *Advances in neural information processing systems*, *30*.
- Mann, M. E., Gille, E., Overpeck, J., Gross, W., Bradley, R. S., Keimig, F. T., & Hughes, M. K. (2000). Global temperature patterns in past centuries: An interactive presentation. *Earth interactions*, *4*(4), 1-1.
- Power, S., Lengaigne, M., Capotondi, A., Khodri, M., Vialard, J., Jebri, B., ... & Henley, B. J. (2021). Decadal climate variability in the tropical Pacific: Characteristics, causes, predictability, and prospects. *Science*, *374*(6563), eaay9165.
- Rudin, C., “Stop explaining black box machine learning models for high stakes decisions and

- use interpretable models instead,” *Nature Machine Intelligence*, vol. 1, no. 5, pp. 206–215, 2019.
- Seneviratne, S. I., Zhang, X., Adnan, M., Badi, W., Dereczynski, C., Di Luca, A., ... & Zhou, B. (2021). 11 Chapter 11: Weather and climate extreme events in a changing climate.
- Sippel, S., Meinshausen, N., Fischer, E. M., Székely, E., & Knutti, R. (2020). Climate change now detectable from any single day of weather at global scale. *Nature climate change*, *10*(1), 35-41.
- Sippel, S., Meinshausen, N., Székely, E., Fischer, E., Pendergrass, A. G., Lehner, F., & Knutti, R. (2021). Robust detection of forced warming in the presence of potentially large climate variability. *Science Advances*, *7*(43), eabh4429.
- Stott, P. A., Gillett, N. P., Hegerl, G. C., Karoly, D. J., Stone, D. A., Zhang, X., & Zwiers, F. (2010). Detection and attribution of climate change: a regional perspective. *Wiley interdisciplinary reviews: climate change*, *1*(2), 192-211.
- Sundararajan, M., Taly, A., & Yan, Q. (2017, July). Axiomatic attribution for deep networks. In *International conference on machine learning* (pp. 3319-3328). PMLR.
- Tang, Z., Chuang, K. V., DeCarli, C., Jin, L. W., Beckett, L., Keiser, M. J., & Dugger, B. N. (2019). Interpretable classification of Alzheimer’s disease pathologies with a convolutional neural network pipeline. *Nature communications*, *10*(1), 2173.
- Yang, Y. M., An, S. I., Wang, B., & Park, J. H. (2020). A global-scale multidecadal variability driven by Atlantic multidecadal oscillation. *National Science Review*, *7*(7), 1190-1197.
- Zeiler, M. D., & Fergus, R. (2014). Visualizing and understanding convolutional networks. In *Computer Vision–ECCV 2014: 13th European Conference, Zurich, Switzerland, September 6-12, 2014, Proceedings, Part I 13* (pp. 818-833). Springer International Publishing.

Reviewer Reports on the Second Revision:

Referees' comments:

Referee #3 (Remarks to the Author):

I want to thank the authors for the thorough revision of the manuscript and for addressing all my concerns: 1) the use of other XAI methods (e.g. IG and SHAP values) showing similar patterns/results to OS, yet not that over-smoothed because of the larger windows used (I suggest to add a comment about it); 2) the smart use of several obs datasets confirms author's arguments on signal attribution, 3) adding error bars improves the robustness, reliability, and validity of the results, and 4) adding the suggested recent DA methods improves the contextualization of the proposed approach. Overall, I'm generally satisfied with the arguments and additional tests provided, and I'm happy to recommend acceptance of the manuscript.

Author Rebuttals to Second Revision:

Referee #3 (Remarks to the Author):

Summary:

I want to thank the authors for the thorough revision of the manuscript and for addressing all my concerns: 1) the use of other XAI methods (e.g. IG and SHAP values) showing similar patterns/results to OS, yet not that over-smoothed because of the larger windows used (I suggest to add a comment about it); 2) the smart use of several obs datasets confirms author's arguments on signal attribution, 3) adding error bars improves the robustness, reliability, and validity of the results, and 4) adding the suggested recent DA methods improves the contextualization of the proposed approach. Overall, I'm generally satisfied with the arguments and additional tests provided, and I'm happy to recommend acceptance of the manuscript.

: Thank you very much for the encouraging comments.